# BANKSY unifies cell typing and tissue domain segmentation for scalable spatial omics data analysis

Vipul Singhal [1,13], Nigel Chou [1,13], Joseph Lee [2], Yifei Yue[3], Jinyue Liu [1], Wan Kee Chock [1], Li Lin[4], Yun-Ching Chang[5], Erica Mei Ling Teo[5], Jonathan Aow [1], Hwee Kuan Lee[4,6,7,8,9,10], Kok Hao Chen [1] ✉ & Shyam Prabhakar [1,11,12] ✉

Spatial omics data are clustered to define both cell types and tissue domains. We present Building Aggregates with a Neighborhood Kernel and Spatial Yardstick (BANKSY), an algorithm that unifies these two spatial clustering problems by embedding cells in a product space of their own and the local neighborhood transcriptome, representing cell state and microenvironment, respectively. BANKSY's spatial feature augmentation strategy improved performance on both tasks when tested on diverse RNA (imaging, sequencing) and protein (imaging) datasets. BANKSY revealed unexpected niche-dependent cell states in the mouse brain and outperformed competing methods on domain segmentation and cell typing benchmarks. BANKSY can also be used for quality control of spatial transcriptomics data and for spatially aware batch effect correction. Importantly, it is substantially faster and more scalable than existing methods, enabling the processing of millions of cell datasets. In summary, BANKSY provides an accurate, biologically motivated, scalable and versatile framework for analyzing spatially resolved omics data.

A fundamental property of solid tissues is the arrangement of individual cell types in stereotypical spatial patterns. Spatial omics technologies now facilitate the study of tissue structure by revealing both the spatial locations and molecular profiles of cells. These technologies provide highly multiplexed transcriptomic, genomic or proteomic profiles at up to single-cell resolution, together with their locations (for example, multiplexed fluorescence in situ hybridization (FISH)[1–3], Slide-seq[4], Slide-DNA-seq[5], multiplexed ion beam imaging by time of flight[6], CosMx[7], CODEX[8]), and thus provide unprecedented insights into cellular states, functions and interactions within the tissue context.

[1]Spatial and Single Cell Systems Domain, Genome Institute of Singapore (GIS), Agency for Science, Technology and Research (A*STAR), Singapore, Republic of Singapore. [2]Faculty of Science, National University of Singapore, Singapore, Republic of Singapore. [3]Department of Chemical and Biomolecular Engineering, National University of Singapore, Singapore, Republic of Singapore. [4]Bioinformatics Institute (BII), Agency for Science, Technology and Research (A*STAR), Singapore, Republic of Singapore. [5]Veranome Biosystems, Mountain View, CA, USA. [6]School of Computing, National University of Singapore, Singapore, Republic of Singapore. [7]Singapore Eye Research Institute, Singapore, Republic of Singapore. [8]International Research Laboratory on Artificial Intelligence, Singapore, Republic of Singapore. [9]School of Biological Sciences, Nanyang Technological University, Singapore, Republic of Singapore. [10]Singapore Institute for Clinical Sciences, Agency for Science, Technology and Research, Singapore, Republic of Singapore. [11]Population and Global Health, Lee Kong Chian School of Medicine, Nanyang Technological University, Singapore, Republic of Singapore. [12]Cancer Science Institute of Singapore, National University of Singapore, Singapore, Republic of Singapore. [13]These authors contributed equally: Vipul Singhal, Nigel Chou. ✉e-mail: chenkh@gis.a-star.edu.sg; prabhakars@gis.a-star.edu.sg

One of the primary spatial omics data analysis tasks is to cluster cells into distinct cell types or subtypes. Because a cell's state is influenced by interactions with other cells, it would be informative to cluster cells using their own transcriptomes as well as their spatial relationships. However, previous spatial omics studies have mostly used clustering algorithms designed for single-cell RNA sequencing (scRNA-seq) data, which ignore spatial information[2,3,9]. It is thus important to develop formalisms for spatially informed cell-type clustering.

Recently, three spatially informed algorithms were proposed for cell-type clustering[10–12]. However, these tools were applied to datasets consisting of 10,000–100,000 cells; it is unclear if they will scale to larger datasets. Moreover, one of the methods[12] assumes that physically distant cells are less similar to each other, even though cells of the same type are often far apart, for example, in the case of intercalated immune cells resident in tissues[13], intermingled neuronal and glial cells in the brain[14] and elongated structures, such as epithelial layers or blood vessels[15]. Furthermore, a cell type can be found in repeating, but spatially separated, structures such as the two cerebral hemispheres[16,17] or neuroepithelial buds within brain organoids[14,18–20].

In contrast to cell-type clustering, multiple algorithms have been developed to identify tissue domains (for example, cortical layers in the mammalian brain[9,21]). This is a distinct algorithmic problem because each tissue domain could in principle include multiple cell types. Earlier domain segmentation methods encouraged physically proximal cells to have the same label (using Markov random fields (MRFs))[22,23]. This assumes that a cell's transcriptome resembles the average transcriptome of cells in its tissue domain, which is not always valid because diverse cell types are commonly intermingled within a single domain. Another family of methods uses deep neural networks[24–27], which provide flexibility in modeling but may be vulnerable to overfitting. Despite the variety of methods available for this task, it is not yet clear which paradigm is optimal for clustering transcriptomic data. Furthermore, as with cell-type clustering, these methods were largely demonstrated on small datasets. With a few exceptions[28], their performance on larger datasets has not been explored.

To address these limitations, we introduce a biologically motivated strategy for combining molecular and spatial information, named Building Aggregates with a Neighborhood Kernel and Spatial Yardstick (BANKSY). BANKSY leverages the fact that a cell's state can be more fully represented by considering both its own transcriptome and that of its local microenvironment. BANKSY uses a pair of spatial kernels to encode the transcriptomic texture of the microenvironment, one constructed using the weighted mean of gene expression in each cell's neighborhood and the other using an azimuthal Gabor filter (AGF), which is related to plane-wave Gabor filters[29–31]. The advantage of this strategy for cell typing is that it does not require cells of the same type to be physically proximal. Furthermore, this allows BANKSY to solve both cell typing and domain segmentation within a single machine-learning framework.

BANKSY labels spatially structured cell types and subtypes with high accuracy and is adept at distinguishing subtly different cell subtypes residing in distinct microenvironments. Moreover, by modifying a single hyperparameter, BANKSY can be used to accurately detect tissue domains rather than cell types. Importantly, BANKSY's feature augmentation strategy allows it to leverage highly scalable graph clustering algorithms that can accommodate millions of cells[32,33]. Finally, BANKSY is interoperable with the widely used bioinformatics pipelines Seurat (R)[34], SingleCellExperiment (R)[35] and SCANPY (Python)[36]. We anticipate that BANKSY's biologically inspired approach could be further extended, potentially forming the basis for additional algorithms. We note that a related strategy of using context to define the meaning of words has revolutionized the field of natural language processing[37].

## Results

One strategy for spatial clustering is to append cells' spatial coordinates to their gene expression vectors[9]. However, this causes spatially distant cells to fall into distinct clusters (Supplementary Section 1 and Supplementary Fig. 1), even when they have identical transcriptomes. A more intuitive approach is to instead append some representation of the cell's microenvironment. BANKSY uses the mean neighborhood expression and the AGF (Fig. 1a,b) to represent the transcriptomic microenvironment around each cell. Importantly, the AGF (Fig. 1b), which can be thought of as measuring the gradient of gene expression in each cell's neighborhood, is invariant to sample rotation. Next, these additional features are used to embed cells in a neighbor-augmented product space (Methods, 'The BANKSY algorithm', Fig. 1c and Extended Data Fig. 1a). After dimensionality reduction, followed by graph construction in the resulting embedding space, clustering can be performed using any graph partitioning algorithm. By default, BANKSY uses the Leiden community detection algorithm[33] for its speed and scalability, although other methods (Louvain, model-based clustering, and $k$-means) are also provided as options.

To control the relative contribution of a cell's own and neighbor features to cell–cell distances in the embedding, BANKSY uses a mixing parameter, $\lambda \in [0, 1]$, to weight the contributions of the cell-transcriptome matrix and the neighbor expression matrices (mean and AGF; Fig. 1 and Extended Data Fig. 1b). Smaller $\lambda$ settings emphasize cells' own transcriptomes and thus cause cells to cluster according to cell type. In the limit, when $\lambda = 0$, BANKSY reduces to conventional nonspatial clustering. BANKSY can be switched from cell-type clustering mode to domain segmentation mode by increasing $\lambda$, which increases the influence of the neighborhood signature and causes cells to cluster according to tissue domain.

We applied BANKSY to multiple datasets across diverse spatial transcriptomic technologies and evaluated performance both qualitatively and quantitatively. One qualitative evaluation strategy involves showing that the spatial algorithm outperforms nonspatial methods at finding expected spatial patterns corresponding to anatomical structures[9,10,23,25,38]. We adopted this strategy in Figs. 2, 3g–j and 5. Another qualitative strategy[11,12,23] is to show that the method discovers previously undescribed biological features not detected by competing methods. This strategy was used in Figs. 3a–f and 4, and in Extended Data Fig. 2. Finally, the quantitative approach[11,23,25] involves quantifying the performance of competing algorithms at recapitulating a set of externally defined cell type or tissue domain labels. We used this strategy in Fig. 5 and in Extended Data Fig. 3. For all analyses, we ran BANKSY using default settings (Methods, 'Default settings for BANKSY embedding and clustering').

### BANKSY improves the cell-type clustering of Slide-seq data

To evaluate cell-type clustering accuracy on spatially structured data, we applied BANKSY to mouse cerebellum data generated using Slide-seq v.1 and v.2 (refs. 4,39). Although both versions of Slide-seq have cellular resolution (10 μm), the spots do not coincide exactly with the cell locations. Consequently, we expected some degree of contamination from the transcriptomes of immediately adjacent cells. This degrades the performance of unsupervised clustering approaches and has motivated deconvolution techniques, such as robust cell type decomposition (RCTD)[40], which use reference scRNA-seq data to infer cell type composition within each spot. We reasoned that we could use BANKSY's neighbor-augmented embedding to more accurately cluster these Slide-seq datasets, even in the absence of a reference dataset.

On both datasets, BANKSY delineated the granular layer, the Purkinje neurons and the molecular layer interneurons (MLIs) more accurately than conventional nonspatial (BANKSY with $\lambda = 0$) clustering (Fig. 2a,b and Supplementary Figs. 2–7). It identified eight clusters in the Slide-seq v.2 dataset, corresponding to eight of the cell types identified using RCTD (Supplementary Figs. 2, 3, 8 and 9), in contrast to the six such clusters identified using nonspatial clustering. To quantify the contiguity of the spots within each layer-specific cluster, we defined a normalized cross-connectivity (NCC) score between each pair of cell types, where a higher score indicates greater intermingling (Fig. 2c).

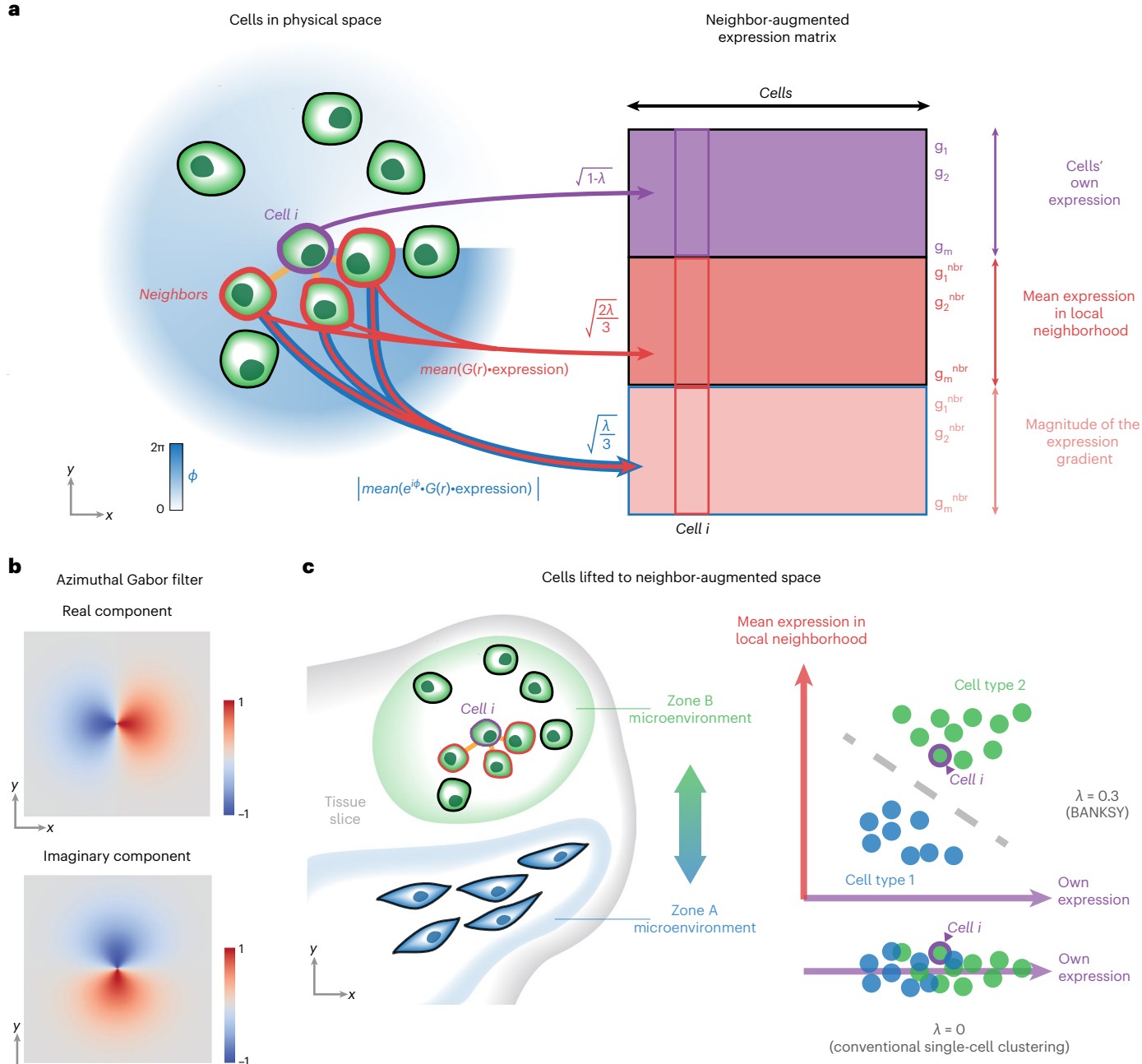

**Fig. 1 | BANKSY's neighborhood-based feature augmentation strategy for clustering. a**, The original gene–cell expression matrix (purple) was augmented with neighborhood-averaged expression matrices corresponding to the mean local expression (dark pink) and the AGF (light pink). Here, $\lambda$ is a mixing parameter that controls the importance of cells' own expression and neighborhood expression effects, $G(r)$ is a radially symmetric Gaussian kernel that decays from magnitude 1 at $r = 0$, 'expression' refers to each gene's expression level in each cell, the mean is taken over cells in the respective index cell's neighborhood and the $e^{i\phi}G(r)$ term confers gradient sensitivity to the AGF. **b**, Heatmap of the real and imaginary components of a gradient-sensitive AGF

kernel. The plots show an unnormalized AGF kernel: the real part ($\cos(\phi)$) senses the gradient along the $x$ axis and the imaginary part ($\sin(\phi)$) senses the gradient along the $y$ axis. **c**, Simplified schematic of two distinct cell types in the neighbor-augmented space. The neighbor expression features, representing the local microenvironment, help to separate two clusters that would be difficult to separate based on the cells' own expression alone. For simplicity, we show 'pure' microenvironments containing only a single cell type (cell type 1 in zone A and cell type 2 in zone B), although BANKSY is equally applicable to heterogeneous microenvironments containing mixtures of cell types (Extended Data Fig. 1).

BANKSY yielded a lower score than nonspatial clustering, confirming our visual assessment that its clusters were more contiguous. To quantify the accuracy of BANKSY, we used supervised RCTD[40] estimates as a reference. In both datasets, BANKSY clusters showed greater correspondence to RCTD inferences and consistency with the expression patterns of known markers (Fig. 2a and Supplementary Figs. 8 and 9). These results indicate that BANKSY enables accurate unsupervised clustering of Slide-seq data.

## BANKSY identifies unexpected cell subtypes in spatial data

Because cell types residing in two distinct microenvironments may have distinct transcriptomic states, we posited that BANKSY's neighbor-augmented approach might detect such subtly different states. We applied both nonspatial ($\lambda = 0$) clustering and BANKSY to a multiplexed error-robust FISH (MERFISH) mouse hypothalamus dataset[2], keeping all parameters besides $\lambda$ identical. Nonspatial clustering recapitulated the previously defined cell types, including mature

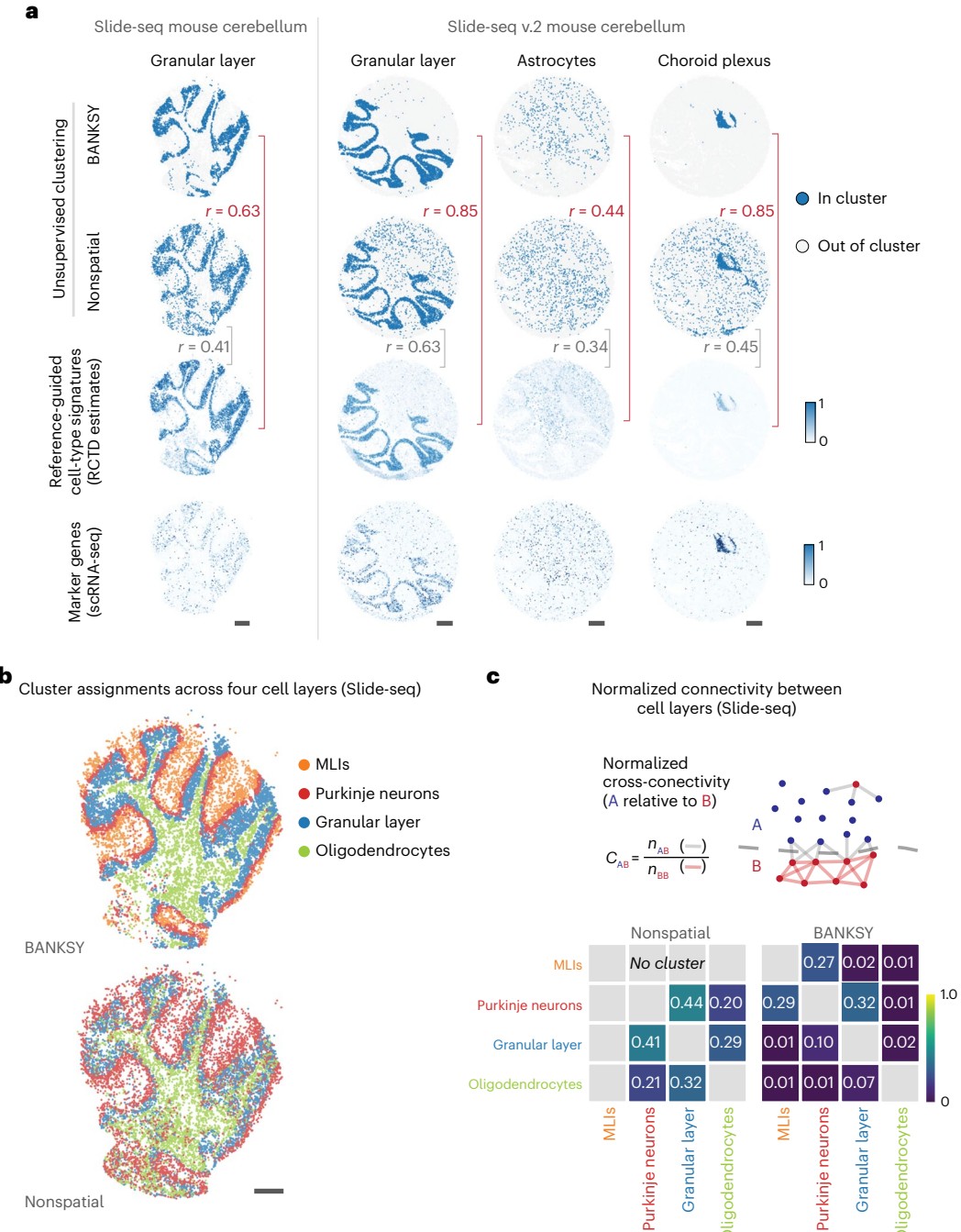

**Fig. 2 | BANKSY enables unsupervised cell-type clustering of mouse cerebellum Slide-seq data. a**, Spatial map of one of the four clusters in the Slide-seq data and three of the eight clusters in the Slide-seq v.2 data (for additional clusters, see Supplementary Figs. 2 and 3). Top row, BANKSY clustering. Second row, nonspatial clustering (BANKSY with $\lambda = 0$). Third row, RCTD[4] weights. The $r$ values indicate the point-biserial correlation of cluster assignments with RCTD weights. Bottom, Averaged expression of top marker genes from the scRNA-seq reference data. **b**, Spatial map of four cell type clusters in the Slide-seq

dataset (granular layer, Purkinje neurons, MLIs and the oligodendrocytes and polydendrocytes cluster). The MLI cluster is absent in nonspatial clustering ($\lambda = 0$). **c**, Top, Computation of normalized spatial cross-connectivity score between two clusters (A and B). Gray lines, spatial proximity relationships connecting cells from the two clusters; red lines, edges between cells from the same cluster. Bottom, NCC score matrix between clusters localized to distinct layers in the Slide-seq dataset. Higher scores indicate greater intermingling between layers.

oligodendrocytes (MODs) (Fig. 3a,b). In contrast, BANKSY separated MODs into two previously undescribed spatially separated subclusters, one of which corresponded to densely packed cells restricted to the anterior commissure of the hypothalamic preoptic region (which we termed white matter MODs (MOD-wm)), and the other to cells spread throughout the rest of the preoptic region (gray matter MODs (MOD-gm); Fig. 3c and Supplementary Figs. 10 and 11). Differential

expression analysis identified genes showing subtle, but coherent, expression differences between the two subclusters (Fig. 3d). *Mbp* and *Lpar1* were upregulated in the MOD-wm cluster (red); *Mlc1*, *Gad1*, *Cbln2* and *Syt4* were upregulated in the MOD-gm cluster (orange). The former set and their guilt-by-association neighbors (highly correlated genes in matched scRNA-seq data; Methods, 'Mouse hypothalamus MERFISH data') are involved in neuronal myelination[41,42], suggesting

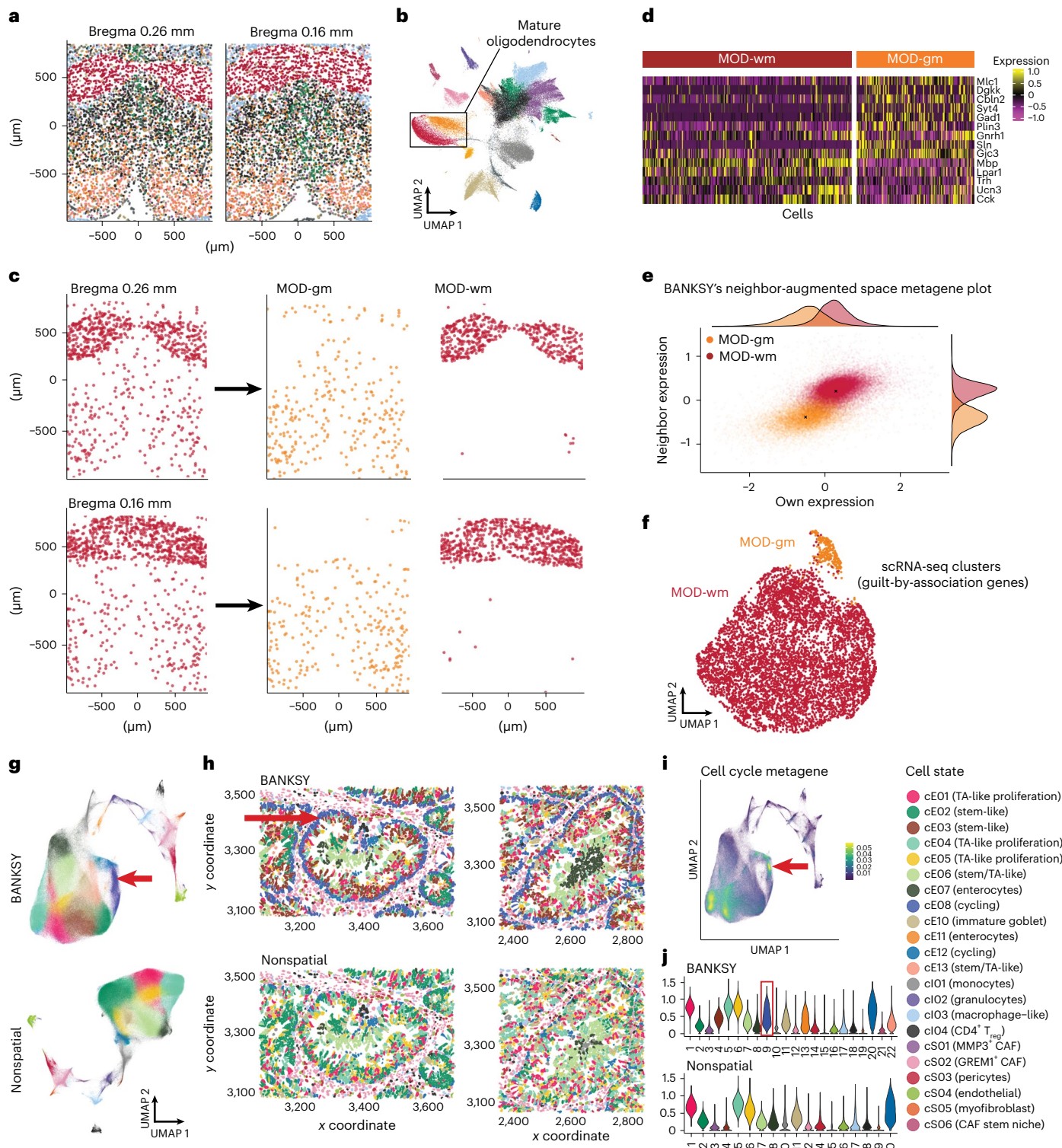

**Fig. 3 | BANKSY enables unsupervised cell-type clustering of mouse hypothalamus MERFISH and human colorectal cancer MERSCOPE data.**
**a**, Overview of the mouse hypothalamus dataset[2], colored using BANKSY cluster labels (Bregma 0.16 mm and 0.26 mm, as defined in the original study).
**b**, Corresponding UMAP representation of the BANKSY embedding, colored using BANKSY cluster labels. The orange and red clusters (boxed) correspond to the spatially separated mature oligodendrocyte subclusters. **c**, Spatial maps of all MODs (nonspatial clustering, λ = 0) and two MOD subtypes (BANKSY): two z-slices (Bregma 0.26 mm and 0.16 mm). **d**, Heatmap of genes differentially expressed between the two MOD subtypes, showing MERFISH expression values z-scaled across the mature oligodendrocytes. **e**, Average expression of MOD DEGs from **d**, plotted in the first two components of BANKSY's neighbor-augmented

product space (x axis, own expression; y axis, mean neighborhood expression).
**f**, Clustering of oligodendrocytes in scRNA-seq data from Moffitt et al.[2], using the DEGs in **d** and their top 25 scRNA-seq guilt-by-association (highly correlated) genes. **g**, UMAP visualization of cell clusters identified in MERSCOPE human colorectal cancer data. Cells are colored using BANKSY and nonspatial clustering (λ = 0) labels. Cluster colors were chosen to indicate correspondence between BANKSY and nonspatial clusters. **h**, Spatial maps for nonspatial (bottom) versus BANKSY (top) clustering, colored as in **g**. Red arrows, cycling epithelial cell cluster detected only by BANKSY. **i**, BANKSY UMAP plot colored according to the expression levels of the cell cycle metagene. **j**, Violin plot showing expression of the cell cycle metagene in BANKSY (top) and nonspatial (bottom) clusters. Red box, cycling cluster indicated in **g**–**i**. CAF, cancer-associated fibroblast.

possible differences in myelination between MODs in the anterior commissure and the rest of the preoptic area. The latter set and their top neighbors are associated with functional and signaling roles, such as maintenance of synapses and ion transport[43]. BANKSY's ability to distinguish these subpopulations arose from its use of neighborhood transcriptomic features, which facilitate better separation of the two clusters (Fig. 3e).

To corroborate the two MOD subpopulations, we clustered matched scRNA-seq data[2] using the DEGs and guilt-by-association genes described above (Supplementary Fig. 12). These genes separated the cells into two clusters (Fig. 3f), with the two groups of markers being upregulated in their respective subclusters (Supplementary Fig. 12). Thus, the MOD subtypes identified by BANKSY are supported by independently generated in vivo scRNA-seq data.

Additionally, BANKSY accurately reproduced cell types and subtypes from the original study (Supplementary Figs. 13 and 14). For example, BANKSY identified the ependymal monolayer in the caudal aspects of the third ventricle (Supplementary Fig. 14a and Supplementary Section 3) and the spatially localized excitatory and inhibitory neuronal subpopulations highlighted in Fig. 5a of the original study (Supplementary Fig. 13a).

To test BANKSY on more disordered tissue, such as neoplastic cancer, we analyzed a MERSCOPE human colon tumor dataset[44], which contained 677,451 cells and was thus too large (Fig. 6) to be processed by other spatial clustering algorithms. BANKSY successfully resolved intermingled mesenchymal and immune cell types and identified cycling epithelial cells that could not be identified using nonspatial clustering (Fig. 3g–j, red arrows and Supplementary Fig. 15).

We also applied BANKSY to a spatial RNA dataset from healthy human colon[7] (Methods, 'Human healthy colon CosMx data'). Uniquely, BANKSY identified a subpopulation of macrophages (Extended Data Fig. 2a, black arrow) that was spatially enriched in the colorectal submucosal layer (Extended Data Fig. 2b,c). Based on the upregulation of known markers[45,46], we annotated this population as M2 macrophages.

## BANKSY distinguishes anatomically distinct cell subtypes

Next, we tested BANKSY's cell clustering on a VeraFISH spatial transcriptomics dataset centered on the hippocampus, which included parts of the thalamus, somatosensory cortex (SSC) and fornix. The results were compared to nonspatial ($\lambda = 0$) clustering, MERINGUE[12], SpaGCN[25] and the hidden MRF (HMRF) methods from Giotto[38], BayesSpace[23], SpiceMix[11] and FICT[10] (Fig. 4). BANKSY identified region-specific cell types with subtly distinct transcriptomes (Fig. 4a–s and Supplementary Figs. 16–20), along with spatially intermingled and dispersed cell types (Supplementary Fig. 17b–d and Supplementary Section 3). For example, BANKSY separated hippocampal CA3 neurons from those in the SSC (Fig. 4a and Supplementary Fig. 18a). We reclustered published scRNA-seq data from the mouse brain[47] using differentially expressed genes (DEGs) between these two BANKSY clusters, along with their scRNA-seq guilt-by-association neighbor genes (Methods, 'Mouse hippocampus VeraFISH data'). This analysis separated the neurons into CA3 and L5/6 cortex (CTX) subclusters (Fig. 4t,u and Supplementary Fig. 18c)[47], confirming that BANKSY correctly identified these known neuronal subtypes.

In contrast to BANKSY, nonspatial clustering and MERINGUE merged the two neuronal subtypes into a single cluster (Fig. 4b,c). While Giotto identified both subtypes, it failed to identify several others (Fig. 4d and Extended Data Fig. 4). The remaining methods variously merged parts of the CA1 and dentate gyrus neurons with these two neuronal populations (blue, green, orange and brown populations in Fig. 4e–h). We reproduced these findings across multiple effective clustering resolutions (Extended Data Figs. 4 and 5).

BANKSY also separated MODs into subtypes localized to the fornix and thalamic nuclei (Fig. 4j), distinguished by the expression of *Mobp*, *Bcas1*, *Sparcl1* and *Nefl* (Fig. 4r and Supplementary Fig. 18b). To verify these subtypes, we reclustered the scRNA-seq dataset using the same guilt-by-association method. This analysis separated the oligodendrocyte cluster into subclusters whose markers matched those of the subtypes identified by BANKSY (Fig. 4v,w and Supplementary Fig. 18d).

Nonspatial clustering and MERINGUE merged the two oligodendrocyte subpopulations (Fig. 4k,l). Giotto failed to identify the thalamic oligodendrocytes; like FICT, it generated several patchy, spatially disconnected clusters (Extended Data Figs. 4 and 5). While BayesSpace and SpaGCN separated most of the cells in these subpopulations, they partially merged the thalamic and fornix oligodendrocytes (Fig. 4n,p, green cluster).

Overall, BANKSY outperformed nonspatial clustering and existing spatial clustering methods at resolving cell types in known anatomical structures in the vicinity of the mouse hippocampus.

## BANKSY accurately segments tissue domains

We quantified BANKSY's ability to segment tissue domains by clustering spots in a Visium human dorsolateral prefrontal cortex (DLPFC) dataset[9]. This dataset, which has previously been used as ground truth to benchmark domain segmentation algorithms, consists of 12 manually annotated brain sections from three individuals[23,25]. We benchmarked accuracy (Methods, 'Human DLPFC 10x Visium data') using the adjusted Rand index (ARI)[9], as well as the normalized mutual information[48] and Matthews correlation coefficient[49] metrics. By all three metrics, BANKSY outperformed Giotto[38], BayesSpace[23], SpaGCN[25] and SpiceMix[11] (Fig. 5a,b, Extended Data Fig. 6 and Supplementary Figs. 21–23). GraphST[26], BANKSY and STAGATE[27] were the top performers on this dataset in terms of median ARIs, with no significant difference between the three methods (Fig. 5b). Note that our results for SpiceMix differed from those reported in the original paper (Methods, 'Human DLPFC 10x Visium data').

Similarly, we benchmarked the domain segmentation accuracy of BANKSY against these algorithms on the STARmap 1,020 gene dataset[21] and found that BANKSY had the highest ARI (Fig. 5c and Extended Data Fig. 7c, Methods, 'STARmap data').

Deep-learning methods can be sensitive to random seed choice[50], which was indeed the case for the three deep-learning methods tested (SpaGCN, STAGATE and GraphST) (Extended Data Fig. 7b). To robustly estimate their performance, we calculated the median ARI across 11 commonly used seeds in both the DLPFC and STARmap analyses. Importantly, BANKSY was highly robust to random seed choice.

**Fig. 4 | BANKSY accurately identifies anatomically distinct cell subtypes around the mouse hippocampus. a**, BANKSY separates CA3 neurons from SSC neurons. **b–h**, Comparisons with nonspatial clustering ($\lambda = 0$) and several spatial clustering methods (**b**, nonspatial; **c**, MERINGUE; **d**, Giotto; **e**, BayesSpace; **f**, SpiceMix; **g**, SpaGCN; **h**, FICT). When an algorithm merged the CA3 or SSC neurons with those in the CA1 or dentate gyrus, the merged clusters are shown in blue (BayesSpace and FICT), light green and brown (SpiceMix) or light blue (SpaGCN). **i**, Example DEGs that distinguish CA3 neurons from SSC neurons. Red, high expression; white, medium expression; blue, low expression. **j**, BANKSY separates the fornix (green) and thalamic (purple) oligodendrocytes. **k–q**, Comparisons to other methods (as in **b–h**). **r**, Example DEGs distinguishing

fornix and thalamic oligodendrocytes. **s**, Schematic showing a coronal section of the hippocampal region shown in **a–r**. **t–w**, Corroboration using scRNA-seq data. **t**, UMAP showing CA3 and cortical neurons from the scRNA-seq data. **u**, Blue box, expression levels of DEGs determined from comparing the CA3 and SSC BANKSY clusters. Brown box, expression levels of the top two corresponding scRNA-seq guilt-by-association genes. **v**, UMAP showing fornix and thalamic oligodendrocytes in the scRNA-seq data. **w**, Blue box, expression levels of DEGs determined from comparing the fornix and thalamic oligodendrocyte BANSKY clusters. Brown box, expression levels of the top two corresponding scRNA-seq guilt-by-association genes.

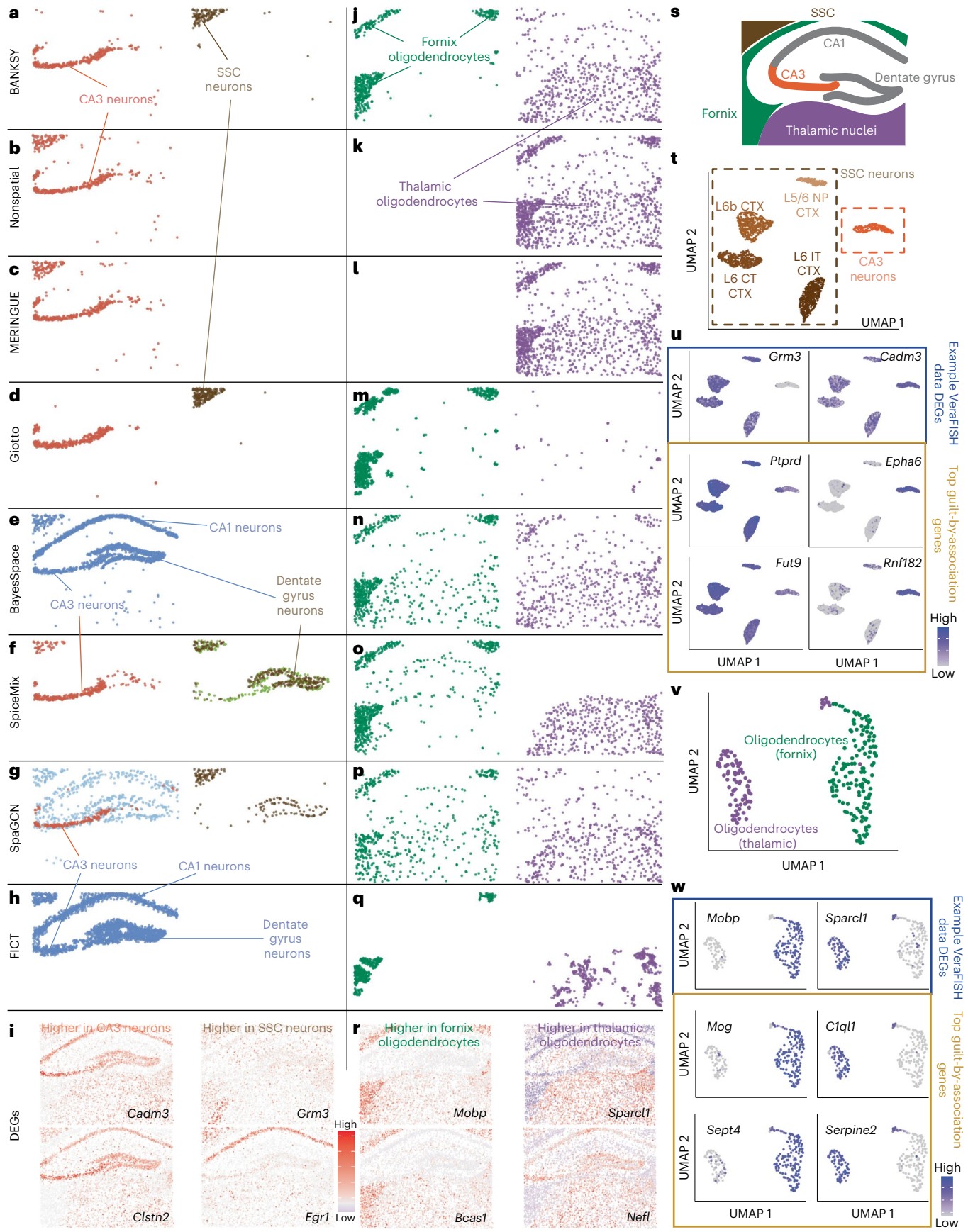

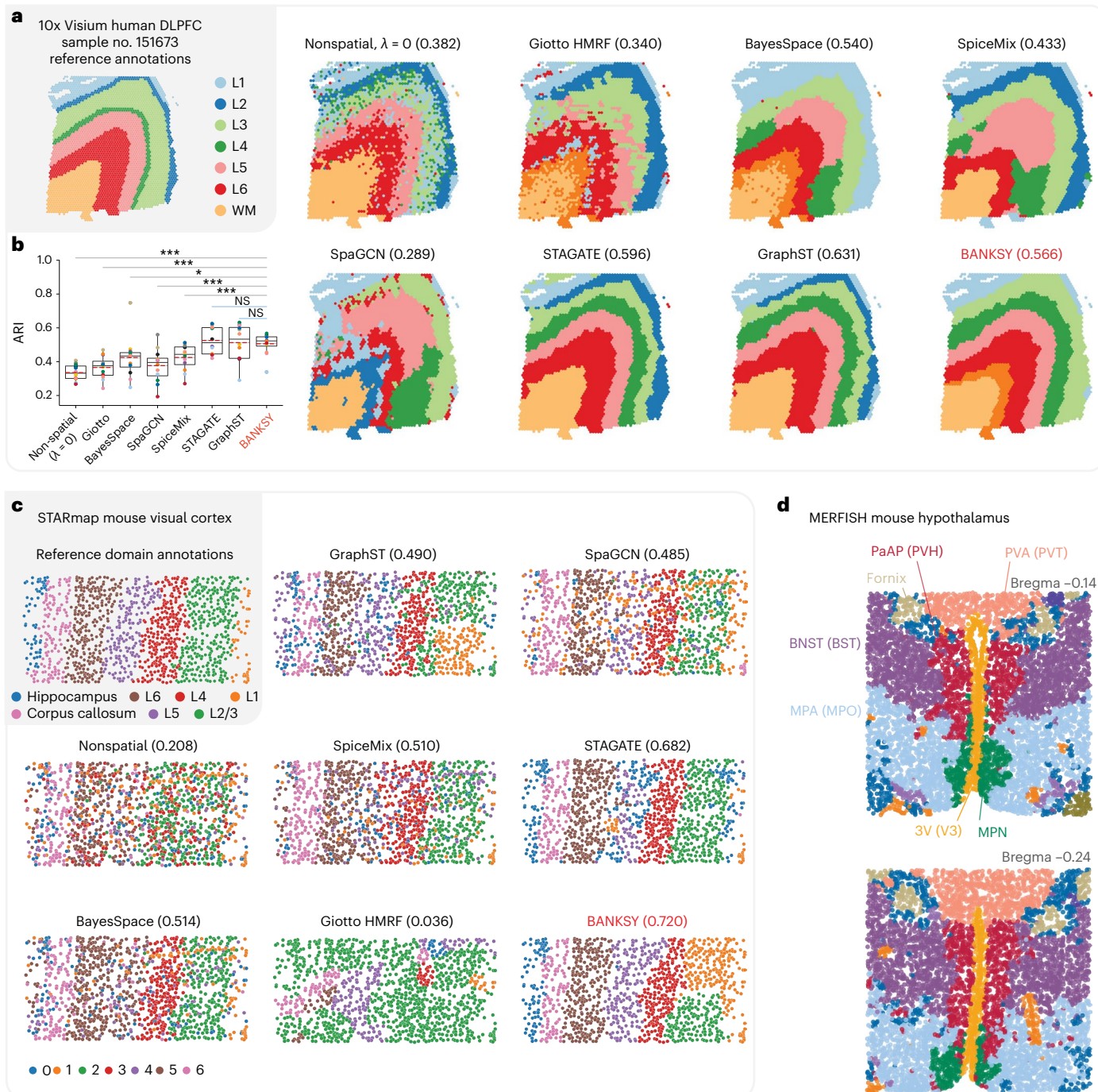

**Fig. 5 | BANKSY accurately segments tissue domains. a**, Left, Reference annotations for sample no. 151673 of the Visium human DLPFC dataset. Right, Spatial maps of DLPFC clusters from nonspatial clustering, Giotto's HMRF, BayesSpace, SpiceMix, SpaGCN, STAGATE, GraphST and BANKSY. Clusters are colored according to their closest matching reference annotation; the numbers in parentheses represent the ARI for each sample computed against the reference annotations. **b**, Box plot of domain segmentation accuracy on the 12 DLPFC datasets, quantified using the ARI metric. For the deep-learning methods (SpaGCN, STAGATE, GraphST) and BANKSY, each of the 12 data points represents the median across 11 random seeds. Center line, median of the 12 data points; red dotted line, mean of the 12 data points; height of the box, interquartile range

(IQR); whiskers, 1.5 × IQR. *$P < 0.05$, ***$P < 0.001$, NS $P > 0.05$ (paired one-sided Wilcoxon signed-rank test). Exact $P$ values from left to right: 0.00024, 0.00024, 0.010, 0.00049, 0.00024, 0.72, 0.48. **c**, Domain segmentation comparison on the STARmap 1,020-gene mouse visual cortex dataset (ARI in parentheses). L1–6, six neocortical layers. **d**, BANKSY domain segmentation identified known brain regions in MERFISH mouse hypothalamus data matching annotations from Moffitt et al.[2] (Allen Reference Atlas-Mouse Brain[52] annotations shown in parentheses when they differ). 3V (V3), third ventricle; BNST (BST), bed nucleus of the stria terminalis; MPA (MPO), medial preoptic area; MPN, medial preoptic nucleus; PaAP (PVH), paraventricular hypothalamic nucleus; PVA (PVT), paraventricular thalamic nucleus.

To demonstrate BANKSY's ability to cluster both cell types and tissue domains in the same dataset, we performed domain segmentation on the mouse hypothalamus MERFISH dataset that we previously analyzed using BANKSY's cell typing mode (Fig. 3). As there were no ground truth annotations for this dataset, we compared the results to brain regions annotated in the original study[2] and the Allen Mouse Brain

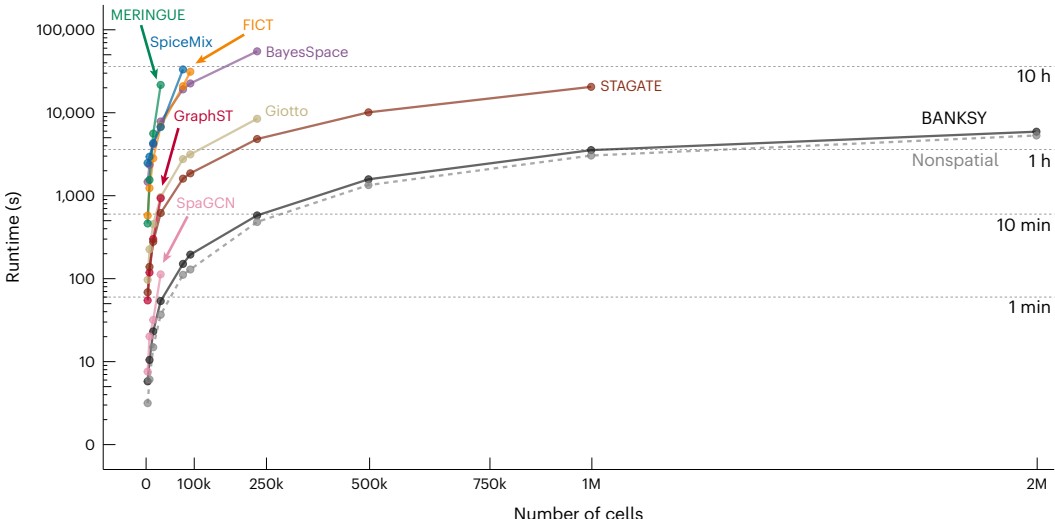

**Fig. 6 | BANKSY is scalable to large datasets and faster than existing spatial methods.** Runtimes of BayesSpace, FICT, Giotto's HMRF module, GraphST, MERINGUE's spatial clustering module, SpaGCN, SpiceMix, STAGATE, nonspatial clustering (Seurat) and BANKSY for increasing cell numbers, up to 2 million cells. All methods were benchmarked on a 16-CPU 128-GB machine. Runtimes are shown up to the maximum cell number accommodated by each method, with a cutoff of 16 h.

Reference Atlas[51,52], and found that the domains identified by BANKSY broadly matched known brain regions (Fig. 5d).

To test BANKSY on spatial proteomics data, we examined a human intestine CODEX dataset[8]. The authors proposed a hierarchical model of intestinal structure, in which cell types formed neighborhoods, which were organized into communities, which in turn formed tissue units. BANKSY's domain segmentation at low resolution (res = 0.06) accurately identified domains matching their tissue unit annotations (Extended Data Fig. 8a–c, Methods, 'Human healthy intestine CODEX data'). At a higher resolution (res = 0.35), BANKSY identified the boundary of the CD66+ mature epithelial community, which correlated with CD66 expression (Extended Data Fig. 8d,e). Additionally, BANKSY identified an α-smooth muscle actin-expressing community that was previously merged with the spatially distinct smooth muscle community.

### BANKSY enables spatially informed batch correction
In the DLPFC benchmarking analysis, we analyzed the data one sample at a time for consistency with benchmarking analyses in previous studies[9,23,25–27]. However, BANKSY can also be run in multisample mode (simple data concatenation) to jointly cluster cells or spots from multiple datasets. For example, concatenating BANKSY's z-transformed neighbor-augmented data matrices of four DLPFC datasets from one individual yielded tissue domain annotations more consistent with manual annotations (Supplementary Fig. 24). Simple concatenation may not always succeed in overcoming batch effects, necessitating the use of explicit batch correction techniques. For example, Huuki-Myers et al.[28] used nonspatial batch correction (Harmony[53]) to integrate the 12 Visium DLPFC datasets and then fed the resulting integrated object to BayesSpace[23] for spatial clustering (Harmony → BayesSpace). One potential limitation of this approach is that spatial information is ignored in the batch correction step.

To explore a spatially informed strategy, we applied Harmony to principal component scores computed using BANKSY's spatial neighbor-augmented embedding of Visium spots from the 12 datasets, and then clustered the spots in the resulting integrated space (BANKSY → Harmony). Joint BANKSY clustering of the 12 datasets in this manner outperformed nonspatial clustering, as well as BayesSpace with and without Harmony batch correction[28] (Extended Data Fig. 9).

### BANKSY scales to large datasets
We tested the scalability of BANKSY and the nine other methods tested in this study to large datasets (Fig. 6) using a mouse brain MERFISH dataset[44]. BANKSY was the only method that scaled to 2 million cells, while the other spatial algorithms hit memory or software limits, or failed to return a result within 16 h (Methods, 'Scalability analysis'). Moreover, BANKSY's runtime was only marginally higher than that of nonspatial clustering, while the other methods were 1–3 orders of magnitude slower. For graphics processing unit (GPU)-compatible methods (GraphST, SpiceMix, STAGATE and SpaGCN), we repeated the scalability benchmarking on a GPU server (Extended Data Fig. 10). BANKSY on a central processing unit (CPU) server was faster and eight times more scalable than these methods on the GPU. In summary, BANKSY is 10–1,000 times faster and 2–60 times more scalable than existing spatial clustering algorithms.

### BANKSY is robust to parameter variation
To further demonstrate the robustness of BANKSY to parameter selection, we performed systematic parameter sweeps on the VeraFISH, MERFISH, Visium and Slide-seq v.1 datasets. We varied $\lambda$, the number of embedding space neighbors ($k_{expr}$), the number of physical space neighbors ($k_{geom}$) and the number of principal components in a range centered on the default values. On all four datasets, the quality of the results was consistent across the range of parameter values, indicating that BANKSY's biologically motivated mathematical model confers stability and robustness to parameter variation (Methods, 'Effect of parameter variation' and Supplementary Figs. 25–40).

### BANKSY performs well on simulated benchmark data
Finally, we tested BANKSY on a simulated dataset, which offered the advantage of unambiguous ground truth labels for cell typing and tissue domain segmentation (Methods, 'Simulated data'). We trained a probabilistic model[54] of gene expression on STARmap expression data[21] and constructed spatial domains with the same cell-type proportions as those in the STARmap dataset. We then tested BANKSY and other methods on both cell typing and domain segmentation across a range of gene set sizes (400, 600, 800 and the full 1,020 gene set). On domain finding, BANKSY attained an ARI greater than 0.8 for all gene subsets, while the remaining methods had ARI values below 0.6 (Extended Data Fig. 3a–c). Across all gene set sizes, BANKSY

was also consistently the most accurate algorithm for cell typing (Extended Data Fig. 3d–f).

## Discussion

We have described BANKSY, a spatial clustering algorithm that unifies cell typing and tissue domain segmentation under a single, scalable machine-learning framework with minimal free parameters (Fig. 1). BANKSY uses the principle that a cell is known by the company it keeps, which suggests that cell states and tissue domains can be identified more accurately by appending spatial neighborhood descriptors to each cell's own gene expression vector. Specifically, BANKSY encodes the local transcriptomic neighborhood of each cell using the average neighborhood expression and the AGF. In the benchmarking analyses, BANKSY identified cell types (Figs. 2 and 4) and anatomical structures (Fig. 5) accurately. BANKSY also sensitively detected spatially segregated cell states, such as the MOD cell states (Fig. 3).

By increasing a single parameter ($\lambda$), BANKSY prioritizes the microenvironment transcriptome over cells' own transcriptomes, enabling detection of contiguous tissue domains defined by spatial neighborhood composition. This allowed BANKSY to identify cortical layers in human DLPFC (Visium) and mouse visual cortex (STARmap) data more accurately than existing domain segmentation tools (Fig. 5a,b,d). Similarly, BANKSY accurately identified anatomical structures in the mouse hypothalamus MERFISH data[2] and accurately segmented tissue domains in a CODEX spatial proteomics dataset of the human intestine[8]. Additionally, BANKSY's domain segmentation mode can be used for 'spatial quality control', that is, for identifying low-quality regions in spatial omics datasets (for example, necrotic zones in organoids and out-of-focus fields of view in imaging-based data).

BANKSY is orders of magnitude faster and 2–60 times more scalable than existing spatial clustering and domain segmentation algorithms (Fig. 6). The computational complexity of BANKSY's neighbor matrix calculation is linear ($\mathcal{O}(n)$) in the number of cells, making it far faster than the clustering step. Thus, BANKSY has, in effect, the same computational complexity as conventional nonspatial clustering. Speed and scalability are essential features of any modern spatial clustering tool, given that technologies are now available to profile over a million cells[55] in a single dataset.

Robustness to parameter variation is another key attribute of BANKSY. On three distinct datasets, BANKSY gave qualitatively similar results across a wide range of parameter settings (Supplementary Figs. 25–40). This essential feature facilitated our analysis of datasets at default parameter settings and engenders confidence in BANKSY's ability to generalize to diverse technologies and sample types with minimal parameter tuning. Indeed, we were able to apply BANKSY to a different single-cell omics modality (the CODEX multiplexed proteomics assay[8]) without changing any of the parameter settings. From a usability perspective, another benefit of BANKSY is that the code integrates seamlessly into widely used single-cell analysis workflows and data structures such as Seurat[56], SingleCellExperiment[35] and SCANPY[36].

Neighborhood-based feature augmentation is a biologically motivated and generalizable strategy for combining spatial and molecular information. We showed that it can be used to robustly and scalably infer cell types and tissue domains from diverse omics technologies. These inferences have been validated using established external benchmarks, that is, scRNA-seq data, known anatomical structures and simulated benchmarks. BANKSY has already been adopted in two recent studies. One sought best practices for analyzing Xenium data: they benchmarked domain segmentation methods on the mouse brain and identified BANKSY as the best performer[57]. Another evaluated BANKSY against five other domain segmentation methods on a human cerebellum Visium dataset and concluded that BANKSY correlated best to independently defined anatomical domains[58]. The generality of BANKSY has enabled its use on datasets completely unrelated to molecular profiling. For instance, it has been used to improve detection of laminar zones in whole-brain magnetic resonance imaging data[59].

Finally, we showed how BANKSY's spatial embedding can be used to aid data integration across samples by performing integrated analysis of the 12 DLPFC datasets[9]. It is likely that the BANKSY embedding could simplify other spatial omics tasks, such as inference of cell–cell signaling interactions and differentiation trajectories. In summary, BANKSY's biologically inspired approach offers an accurate, sensitive, versatile and scalable spatial clustering tool that unifies cell-type identification and domain segmentation.

## Online content

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

## Methods

All experiments were carried out in accordance with Agency for Science, Technology and Research (A*STAR) guidelines. In particular, the VeraFISH mouse data were collected in accordance with the approved Institutional Animal Care and Use Committee (IACUC) protocol no. 211580 obtained from the biomedical resource center.

### The BANKSY algorithm

In this section, we describe the BANSKY algorithm in the context of clustering a set of cells, based on the expression of a set of genes and the physical locations of the cells or capture spots (used in sequencing-based technologies). We captured gene expression variations in the spatial neighborhoods of cells using a weighted average of the expressions of neighboring cells and a kernel inspired by the Gabor filter. The Gabor filter is used in computer vision applications and involves the construction of the kernel by modulating a planar sinusoidal wave using a Gaussian function[29–31]. In this study, instead of modulating a planar wave, we modulated the azimuthal Fourier transform of the gene expression in a cell's neighborhood. The resulting kernel captures variations in gene expression in azimuthally anisotropic directions within each cell's neighborhood (that is, it measures gradients in gene expression in each neighborhood).

Before describing the algorithm in detail, we give a brief overview of its main mechanics. We began by constructing a neighborhood graph between cells in physical space, which can be done using, for instance, $k$-nearest neighbors or radius nearest neighbors. We used this to compute feature-wise $z$-scaled versions of two matrices: an average neighborhood expression matrix and an AGF matrix. These matrices are then scaled on the basis of a mixing parameter $\lambda$, which controls their relative weighting, and concatenated with the original gene–cell expression matrix to construct two neighbor-augmented matrices, one for cell typing and another for domain segmentation, which differ only in the value of $\lambda$ used in the combination step. Conceptually, this corresponds to lifting the embedding of the cells from the (nonspatial) gene expression space to a space constructed by taking the direct product of the original nonspatial expression space, the average neighborhood expression space and the AGF space. Both the cell typing and domain segmentation neighbor-augmented matrices are then processed similarly: dimension reduction (principal component analysis (PCA)), followed by graph-based clustering, resulting in the respective cell type and domain segmentation labels.

In the following sections, we describe this procedure formally.

**Spatial kernels.** Let a set of cells arranged in physical space be indexed by the set $\mathcal{U} = \{1, 2, \dots, N\}$, and have a set of spatial coordinates, along with matching cell indices, given by $\mathcal{X} = \{(u, x_u) \in \mathcal{U} \times \mathbb{R}^2\}$. For each cell, assume that the expression of the same set of $p \in \mathbb{Z}_{\geq 1}$ genes has been measured, so that the expression information can be expressed as a gene–cell expression matrix $\mathcal{C} = [c_1 \, c_2 \, \dots \, c_N] \in \mathbb{R}^{p \times N}$, where $c_u \in \mathbb{R}^p$ is the expression of the $p$ genes in cell $u$.

Next, we constructed spatial neighborhoods for all cells. In the present discussion, we restrict ourselves to $k$-nearest neighbors as our neighborhood construction policy, but we note that radius nearest neighbors or other policies may be handled analogously. We denote the set of $k_{\text{geom}}$ neighbors for cell $u$ with the following equation:

$$\eta^u_{k_{\text{geom}}} = \{v \in \mathcal{U} \setminus \{u\} | \text{ cell } v \text{ is within the } k_{\text{geom}}$$

$$\text{nearest neighbors of cell } u \text{ in } \mathcal{X}\}$$

**Weighted neighborhood mean.** For each gene $q$, we computed the mean expression in the neighborhood of cell $u \in \mathcal{U}$ as:

$$M_u^{(q)} = \sum_{v \in \eta^u_{k_{\text{geom}}}} g_v^{(q)} \Gamma_{uv}^{k_{\text{geom}}}$$

where $g_v^{(q)}$ is the expression of the $q$th gene in neighboring cell $v$ and the term $\Gamma_{uv}^{k_{\text{geom}}}$ is a Gaussian weighting envelope defined as:

$$\Gamma_{uv}^{k_{\text{geom}}} = \frac{\exp \frac{-r_{uv}^2}{r_{u\bar{v}}^2}}{\sum\limits_{w \in \eta^u_{k_{\text{geom}}}} \left( \exp \frac{-r_{uw}^2}{r_{u\bar{v}}^2} \right)}$$

with $\bar{v}$ defined as the $\lfloor (m \times k_{\text{geom}})/2 \rfloor$-th nearest neighbor (that is, for $k_{\text{geom}} = 15$, $\bar{v}$ corresponds to the 7-th nearest neighbor and $r_{u\bar{v}}$ is the distance between cells $u$ and $\bar{v}$). This effectively makes the Gaussian modulation envelope invariant to differences in cell densities in physical space. As a default, we set $k_{\text{geom}} = 15$ in all analyses in this study. Note that the results are insensitive to moderate variations around this set point (Supplementary Figs. 27 and 35).

**AGF.** Consider cells $u, v \in \mathcal{U}$. Let $(r_{uv}, \phi_{uv})$ be the position of cell $v$ in the local polar coordinate frame of cell $u$, with origin $(r_{uu}, \phi_{uu}) = (0, 0)$ and $\phi$ measured counterclockwise from the positive horizontal axis. For gene $q$, cell $u$, the AGF is defined as:

$$G_u^{(q)} = | \sum_{v \in \eta^u_{2k_{\text{geom}}}} g_v^{(q)} \Gamma_{uv}^{2k_{\text{geom}}} e^{i\phi_{uv}} | \tag{1}$$

where $\eta^u_{(\cdot)}$, $g_v^{(q)}$ and $\Gamma_{uv}^{(\cdot)}$ are as defined above. We used $2 \times k_{\text{geom}}$ as the number of spatial neighbors to compute the AGF because estimating a gradient requires more data points than estimating a mean. The azimuthally varying complex sinusoid $e^{i\phi_{uv}}$, together with the scale-invariant Gaussian-like wave modulation term $\Gamma_{uv}^{k_{\text{geom}}}$ in the Fourier transform (equation (1)) is in the spirit of the Gabor filter and motivates our terminology.

**Neighbor-augmented matrix.** Once the set of the individual terms $M_u^{(q)}$ and $G_u^{(q)}$ (over the genes and cells) have been calculated, we may collect them into two matrices, that is:

$$\mathcal{M} = \begin{bmatrix} M_1^{(1)} & M_2^{(1)} & \dots & M_N^{(1)} \\ \vdots & \vdots & \ddots & \vdots \\ & & & \end{bmatrix}$$

and

$$\mathcal{G} = \begin{bmatrix} G_1^{(1)} & G_2^{(1)} & \dots & G_N^{(1)} \\ \vdots & \vdots & \ddots & \vdots \\ G_1^{(p)} & G_2^{(p)} & \dots & G_N^{(p)} \end{bmatrix}$$

The rows (features) of these two matrices, along with those of the gene–cell matrix $\mathcal{C}$ are then zero-centered and scaled to have unit variance ($z$-scaled). The resulting scaled $\mathcal{C}$, $\mathcal{M}$ and $\mathcal{G}$ matrices are subsequently combined to form a neighbor-augmented matrix:

$$\mathcal{B} = \begin{bmatrix} \sqrt{1 - \lambda} \, \mathcal{C} \\ \sqrt{\lambda/\mu} \, \mathcal{M} \\ \sqrt{\lambda/(2\mu)} \, \mathcal{G} \end{bmatrix} \in \mathbb{R}^{3p \times N} \tag{2}$$

where $\mu = 1.5$ is a normalization factor to ensure the convexity of the linear combination of distance matrices (Supplementary Section 2) and $\lambda \in [0,1] \subset \mathbb{R}$ is a mixing parameter that controls the relative weights of the three component matrices. For cell typing, we used a default value of $\lambda = 0.2$ to construct the neighbor-augmented matrix $\mathcal{B}$; for domain segmentation, we used $\lambda = 0.8$ for the creation of the corresponding $\mathcal{B}$ matrix. In either case, the subsequent processing of this matrix was similar: PCA (20 principal components as a default) for

dimensionality reduction, followed by Leiden clustering for community detection. Finally, we note that at $\lambda = 0$, the algorithm only takes the cells' own expression into account and reduces to nonspatial (conventional) clustering.

## NCC score

We defined an NCC score to quantify intermingling across layers and used this to compare different clustering solutions (BANKSY versus nonspatial ($\lambda = 0$) in Fig. 2). In this section, we describe this score formally. Let $u$ and $\eta^u_{k_{geom}}$ be as defined in Methods, 'Weighted neighborhood mean'. A particular clustering solution is a partition of $u$; that is, if the set of cells has been clustered into $p$ clusters, $u$ can be written as $u = \cup_{q=1}^{p} s_q$, where for each $q$, $s_q \subset u$ is the set of indices of cells assigned to the $q$th cluster and may therefore be identified with that cluster. These sets are pairwise disjoint: $s_q \cap s_r = \varnothing$ for arbitrary $q, r \in \{1, 2, ..., p\}$.

Given $u$, $\eta^u_{k_{geom}}$ for each $u \in \mathcal{U}$ and a clustering solution $\{s_w\}_{w=1}^{p}$, we can define the NCC score of a given query cluster $s_q$ with regard to a reference cluster $s_r$ as:

$$\text{NCC}_{s_q, s_r} = \frac{\sum_{u \in s_r} \sum_{v \in \eta^u_{k_{geom}}} \mathbf{1}_{s_q}(v)}{\sum_{u \in s_r} \sum_{v \in \eta^u_{k_{geom}}} \mathbf{1}_{s_r}(v)} \tag{3}$$

where $\mathbf{1}_{s_q}(\cdot)$ is the indicator function:

$$\mathbf{1}_{s_q}(v) := \begin{cases} 1 & \text{if } v \in s_q, \\ 0 & \text{otherwise} \end{cases} \tag{4}$$

A higher NCC value indicates greater spatial intermingling between clusters, as this intermingling will result in a higher between-cluster connectivity (numerator).

## Metagene computation

Metagene expression captures the difference between two clusters that are to be compared and can be used to visualize how BANKSY helps to better separate out the cells in subclusters in the neighbor-augmented space, relative to the original own-expression space (as shown in Fig. 3b).

For the $i$-th cell, we defined the metagene expression of subcluster 2 relative to subcluster 1 as the difference between the average expressions of DEGs more highly expressed in subcluster 2 and those more highly expressed in subcluster 1. Explicitly, for the $i$-th cell, let $\{g_{11}^{(i)}, ..., g_{1p}^{(i)}\}$ be the expression values ($z$-scaled over all the cells in the two subclusters) of each of the $p$ DEGs upregulated in subcluster 1; similarly, let $\{g_{21}^{(i)}, ..., g_{2q}^{(i)}\}$ be the values of the $q$ genes upregulated in subcluster 2; thus, the metagene of the own expression for subcluster 2 relative to subcluster 1 is defined as:

$$m_{21}^{(i)} = \frac{\sum_{k=1}^{q} g_{2k}^{(i)}}{q} - \frac{\sum_{j=1}^{P} g_{1j}^{(i)}}{p} \tag{5}$$

The metagene expression for the expression of the neighbor is defined analogously using the corresponding $p$ and $q$ neighbor expression rows of these genes in the matrix $\mathcal{M}$. Once the metagene expressions are computed, each cell can be plotted in the product space of the neighboring and own-expression metagenes.

## Cluster consensus across runs

The comparison of BANKSY to nonspatial clustering ($\lambda = 0$) and other methods required matching cluster labels across different clustering solutions, such that clusters corresponding to the same or similar sets of cells were given the same numeric labels. This cluster matching or consensus enabled direct comparison of specific clusters to their closest equivalents in other methods. One example of this may be

seen in Fig. 4, where clusters corresponding to cell types arranged in anatomical structures were compared between BANKSY, nonspatial clustering and other existing methods, and in some cases across different sets of clustering resolutions or other parameters (for instance, in Fig. 4, Extended Data Figs. 4 and 5 or Supplementary Figs. 25–40). Another set of examples may be seen in Fig. 5a and the associated Supplementary Fig. 21, where domain segmentation solutions across different methods were compared. Cluster consensus is also required for computing some clustering accuracy metrics, for example, the Matthews correlation coefficient, one of the three metrics used to benchmark clustering performance on the human DLPFC dataset.

To perform the cluster consensus, we posed the cluster consensus problem as a linear sum assignment problem, which admits a strongly polynomial solution via the Hungarian algorithm[60]. Briefly, given two clustering partitions (defined in Methods, 'NCC score') on the same dataset, we set up a matrix such that the rows and columns corresponded to individual clusters from the two clustering partitions, respectively, and the entries were the number of cells that were common between the pair of clusters corresponding to that row–column pair. The goal was to find, for each row of this matrix, a unique matching column, so that the number of cells that were common between the matching clusters was maximized over all the clusters.

For the Python version of the code, we used the SciPy (v.1.6.2) function scipy.optimize.linear_sum_assignment[61]. For the R version of the code, we used the HungarianSolver function within the RcppHungarian[62] library on CRAN.

## Default settings for BANKSY embedding and clustering

For all analyses used in this study, we used the default parameters listed in this section.

The physical-space neighborhood graph was constructed using the spot, bead or cell locations, with a $k$-nearest neighbor approach. We used a setting of $k_{geom} = 15$, except in the Visium DLPFC data, where the spots were arranged in a unique hexagonal geometry. This motivated the use of $k_{geom} = 18$, which corresponded to taking up to the second-order neighbors of each spot.

In all cases, we $z$-scaled each row (centered the means and scaled the standard deviations of each feature to 1) of the $\mathcal{C}$, $\mathcal{M}$ and $\mathcal{G}$ matrices before computing the neighbor-augmented matrix. To compute the neighbor-augmented matrix, we used a setting of $\lambda = 0.2$ for cell typing and $\lambda = 0.8$ for domain segmentation, unless specified otherwise.

Next, we used PCA to reduce the dimensionality of the neighbor-augmented matrix, retaining the top 20 principal components. We then used the Leiden clustering[33] algorithm at default settings, with the number of neighbors in the expression space graph set to $k_{expr} = 50$, and ran it to convergence. We used the same parameters from the PCA step onward to perform nonspatial ($\lambda = 0$) clustering.

## Mouse cerebellum Slide-seq data

For the Slide-seq mouse cerebellum dataset, we filtered out cells with fewer than 20 and more than 1,000 gene counts. From Slide-seq v.2, we filtered out cells with fewer than 50 and more than 2,500 gene counts. For both datasets, we removed genes present in fewer than ten cells and retained the top 2,000 highly variable genes (HVGs). For Slide-seq v.2, we also filtered out bead locations beyond a fixed radius of 2,550 μm around an approximate center point of the puck to remove beads outside the puck.

We then normalized the total gene counts per cell to the median gene count and ran nonspatial and BANKSY clustering using default parameters ($\lambda \in \{0, 0.2\}$), with a clustering resolution parameter of 0.7.

We matched each cluster from both BANKSY and nonspatial clustering ($\lambda = 0$) to their closest scRNA-seq reference cluster using RCTD cell-type composition estimates and marker gene expression (the mean of top marker genes in the scRNA-seq reference cell type) as a guide.

We then quantitatively compared BANKSY and nonspatial clusters by computing the point-biserial correlation of the cluster assignments with the RCTD weights. We performed the same correlation analysis against the scRNA-seq marker gene expression levels.

## Mouse hippocampus VeraFISH data

One 6-week-old female mouse (C57BL/6NTac) was purchased from InVivos (https://www.invivos.com.sg/c57bl-6ntac/#tab-60391) and was euthanized and dissected for removal of the brain immediately on receipt. All animal procedures were done in accordance with the approved IACUC protocol (protocol no. 211580) obtained from the IACUC of the biomedical resource center. Mouse brains were embedded in optimal cutting temperature compound (Sakura), frozen and stored at −80 °C.

The Veranome Biosystems VeraFISH assay was used for data collection. Tissue samples were permeabilized using 70% ethanol overnight and incubated with VeraFISH sample prep wash buffer for 2 h at 47 °C. After overnight incubation in VeraFISH Target Probe and Hyb Buffer, samples were rinsed with VeraFISH sample prep wash buffer for 1 h at 47 °C and stored with the VeraFISH Cycling Buffer at 4 °C until imaging. Tissue samples were imaged in a fully automatic VSA-1 Imager from Veranome Biosystems that includes incubation with VeraFISH Barcode Probes and imaging through multiple hybridization processes. The multiplexed images were then processed using the VeraWorks software to reconstruct the spatial coordinates of the VeraFISH Barcode Probes. A Mask R-CNN-based deep-learning pipeline was used to segment 4′,6-diamidino-2-phenylindole-stained nuclei and generate single-cell expression matrix in .csv files (cellmatrix.csv).

Once we had the raw gene counts per cell, we removed cells with total counts less than the 5th percentile and greater than the 98th percentile. We also removed any genes with expression (at least one count) in less than 1% of cells. The total counts in each cell in the resulting gene–cell matrix were normalized to a value of 100, followed by BANKSY and nonspatial clustering using default parameters ($\lambda \in \{0, 0.2\}$), with a clustering resolution of 1.5.

To compute DEGs between the pair of neuronal subclusters shown in Fig. 4 and Supplementary Fig. 18a, we used the scran R package (v.1.18.7)[63]. We performed a Wilcoxon rank-sum test and $t$-test (two-sided, using default parameters) between each pair of clusters and took the union of the top ten markers from each of these comparisons. We repeated this procedure for the pair of oligodendrocyte subclusters.

For the scRNA-seq clustering analysis using the guilt-by-association genes (Fig. 4t–w and Supplementary Fig. 18), we used data from Yao et al.[47]. To isolate the L5/6 SSC and CA3 neurons, we used cells labeled 'L5/6 NP CTX', 'L6 CT CTX', 'L6 IT CTX', 'L6b CTX' and 'CA3' from the Smart-seq v.4 dataset. Our analysis was performed using the Seurat package, v.4.1.1 (ref. 34). We first normalized the expression count data using the NormalizeData function, with the normalization. method argument set to LogNormalize. That is, each cell's expression counts were divided by the total expression in that cell and multiplied by a scale factor of 10,000; finally, we log-transformed the data using log1p. Next, we used the ScaleData function to 0-center the mean of the expression of each gene and scaled its standard deviation to 1. Next, starting with the genes that were differentially expressed between the CA3 and SSC neurons in the VeraFISH data (as computed above), we first removed any genes not expressed in at least 1% of cells (using the raw counts data). Next, for each of the remaining genes, we computed the top 25 most highly correlated genes in the log-normalized and scaled data and once again removed any genes not expressed in at least 1% of cells. Finally, we clustered the cells using the remaining genes, repeating the subsetting, normalization and $z$-scaling steps on the raw count data for these genes, using five principal components, a resolution of 0.2 and all other parameters set to the Seurat defaults. We repeated this analysis for the oligodendrocytes by subsetting cells labeled 'Oligo' from the data in Yao et al.[47] (Smart-seq v.4 dataset).

For the MERINGUE (v.1.0) runs on this dataset, we used the standard pipeline from Miller et al.[12]. Briefly, MERINGUE computes a Delaunay triangulation graph between the physical locations of cells, with a maximum threshold distance within which cells are considered neighbors (given by the filterDists argument in the getSpatialNeighbors function; we chose a threshold of 750 pixels, such that most adjacent cells were defined to be neighbors). It then computes the shortest path length (geodesic) distance in this physical space graph between each pair of cells that are neighbors in the transcriptome space graph. These distances are used to weight the edges of the transcriptome space graph during graph-based clustering. We used 20 principal components and $k_{expr} = 10$ in MERINGUE's spatial clustering function, getSpatiallyInformedClusters. We then varied the parameter that controlled the effective clustering resolution ($k_{expr}$) to obtain clustering runs with 16–20 clusters.

Similarly, we ran BayesSpace (v.1.5.1)[23] on the first 20 principal components computed on the normalized gene–cell matrix. To define neighborhood structure for the Markov random field model, we created a spatial network with $k = 15$ nearest neighbors. For clustering, we used default settings ($t$-distributed error model with 50,000 Markov chain Monte Carlo iterations and a burn-in period of 1,000 iterations) and a gamma smoothing parameter of 3. The output number of clusters was varied from 16 to 20.

For Giotto (v.1.1.0), we first created a spatial network with $k = 10$ neighbors using default settings (Delaunay method and the maximum distance cutoff set to 'auto'). To identify genes with spatially coherent expression patterns, we ran BinSpect with default parameters and selected the top 50% of genes based on the BinSpect score. Based on the number of clusters obtained from nonspatial ($\lambda = 0$) and BANKSY clustering, we ran Giotto's HMRF for a fixed number of domains (16–20 clusters). Based on the authors' recommendations (https://cran.r-project.org/web/packages/smfishHmrf/smfishHmrf.pdf)[22], we tested $\beta$ (HMRF regularization parameter) from 0 to 50 in increments of 2. Finally, we selected the cluster labels obtained with $\beta = 12$ because the labels around this value of $\beta$ were relatively stable.

To run FICT (v.1.0.0), we had to modify the code for preparing the input data provided by the authors (https://github.com/haotianteng/FICT-SAMPLE)[10] because the sample code was only tailored to particular (fixed) datasets. After preparing the input data, we ran FICT on a neighborhood graph constructed with $k = 10$ nearest neighbors and a reduced dimension expression profile with $d = 20$. The number of cell types (clusters) was set to 16–20.

For SpiceMix[11], we first created a neighborhood graph from spatial coordinates using $k = 10$ nearest neighbors with the dbscan (v.1.1-10) package in R[64]. We then ran SpiceMix on the neighborhood graph and processed the expression matrix. We used the default setting for all other parameters according to the authors' tutorial notebook (https://github.com/ma-compbio/SpiceMix/blob/master/SpiceMix/main_Maynard2021.ipynb) from the latest version of SpiceMix (GitHub commit ID: aea69f8), which differs from the published version[11]. Next, we chose results from the iteration (20) that maximized the Q function returned by SpiceMix (log-joint probability of the parameters and data). To obtain cluster labels, we performed Leiden clustering[33] on the metagene latent space with the igraph package (v.1.2.11) in R[65], yielding 16–20 clusters by varying the clustering resolution parameter.

## Mouse hypothalamus MERFISH data

The data for the mouse hypothalamus atlas study[2] were downloaded from the link provided in the Data availability section. We used the data from all 11 naive animals (485,657 cells) and processed them with BANKSY's multisample mode, as described next. As done in the original study, the gene *Fos* was removed from the dataset because it contained not a number ('NaN') entries. This dataset consisted of gene–cell matrices ($c$) that were normalized by the imaged volume of each cell; therefore, it did not need to be count-normalized further. We ran BANKSY

and nonspatial clustering using default parameters ($\lambda \in \{0, 0.2\}$), with a clustering resolution of 0.5. The multisample mode involved concatenating the 11 neighbor-augmented matrices along the columns to form a larger multisample matrix. The resulting cluster labels were harmonized using our cluster consensus algorithm, as described in Methods, 'Cluster consensus across runs'. As in the original study, we removed all cells marked 'ambiguous' from the subsequent analysis. To compute the genes that were differentially expressed between the two oligodendrocyte subclusters, we used the scran package (v.1.18.7) as before, taking the union of the top ten DEGs from both the $t$-test and the Wilcoxon rank-sum test, with expression $z$-scaled across the oligodendrocytes.

We used Seurat v.4.1.1 to perform the clustering analysis using the guilt-by-association genes in the scRNA-seq data provided by Moffitt et al.[2]. The data matrix was filtered so that only cells with less than 20% mitochondrial RNA (pct.mito) and cells with more than 1,000 detected genes were kept (nFeature_RNA). Next, we subset the data to keep only cells labeled MODs by the authors of the original study[2] and divided the counts in each cell by the total counts in that cell, multiplied by 10,000 (Seurat default), log-transformed using log1p, and $z$-scaled the data (centered to 0 and the standard deviation scaled to 1). Next we identified the 25 most highly correlated genes to the DEGs found above. We removed genes that did not have a count of at least 1 in at least 1% of all MODs. The original data were subset to the resulting set of guilt-by-association genes and to the MODs, count-normalized to 10,000, log1p-normalized and $z$-scaled, as before. Next, we used Seurat's FindNeighbours with default parameters and the top five principal components to build a neighborhood graph in the expression space, followed by the FindClusters function with a resolution of 0.05. The uniform manifold approximation and projections (UMAPs) were generated with the top five principal components.

### Human colorectal cancer MERSCOPE data

Vizgen MERSCOPE data comprising the gene–cell matrix and cell centroids for the sample 'colon cancer 1' was obtained from https://info.vizgen.com/ffpe-showcase. The raw data consisted of 677,451 cells profiled across 500 genes. For the downstream analysis, we kept cells whose total transcript count was between the 5th and 98th percentile, yielding 629,164 cells. The total transcript count in each cell was normalized to the median transcript count and then natural log-transformed. We then performed BANKSY and nonspatial clustering using default settings ($\lambda \in \{0, 0.2\}$).

To obtain cluster annotations, we ran the reference-based single-cell annotation method SingleR (v.1.4.1)[66]. To construct a reference, we used scRNA-seq data from Pelka et al.[67], where 370,115 single cells were sequenced from colon normal and tumor samples obtained from 62 individuals. To reduce the influence of batch effects on cluster annotations, we first created pseudobulk expression profiles for each individual by taking the average expression of all cells from a single individual for each gene. Next, we created a pseudobulk for MERSCOPE data by taking the average expression across all cells. We then computed the Spearman correlation between the pseudobulk expression profile for each individual with the MERSCOPE pseudobulk and identified the individuals with the two highest correlations ($\rho = 0.67, 0.55$) for constructing the reference data. Raw counts and cell-type labels from these individuals were used as a reference in SingleR, with a Wilcoxon rank-sum test used to identify the top ten genes from each pairwise comparison between labels. Using pruned labels returned from SingleR, we annotated each cluster based on the majority label of all cells within a given cluster.

To identify cycling cells, markers were obtained from Table S2 from Smillie et al.[68] and Table S5 from Tirosh et al.[69]. Intersection of these markers with the genes present in the MERSCOPE dataset yielded a final gene set of 13 genes (*Aurkb*, *Birc5*, *Bub1*, *Ccnb1*, *Cdca7*, *E2f1*, *Foxm1*, *Mcm2*, *Mcm6*, *Mki67*, *Mybl2*, *Pcna* and *Plk1*). We used Nebulosa

(v.1.0.2)[70] to visualize the cycling metagene on the UMAP embedding of the BANKSY matrix.

### Human DLPFC 10x Visium data

The 10x Visium data of the DLPFC was obtained from the spatialLIBD project (http://spatial.libd.org/spatialLIBD)[71]. The data consist of 12 samples obtained from three individuals (four samples per individual). The layers of each sample were manually annotated[9], with each sample having either five or seven layers. Before the analysis, we removed all spots with ambiguous layer assignments. We ran BANKSY and nonspatial clustering ($\lambda \in \{0, 0.2\}$) on each sample separately, first normalizing the counts per spot to the median library size (total number of transcripts) across all spots of the sample. HVGs were then identified by modeling the mean variance relationship for each gene as implemented in Seurat[56]. Data were subset to the top 2,000 variable genes. BANKSY was run using default settings, with one exception: because of the unique hexagonal geometry of these data, we used $k_{geom} = 18$, which corresponded to taking all spots up to second-order neighbors of a given index spot. We adjusted the resolution parameter from 0.1 to 1.5 such that the number of clusters obtained matched the number of layers present in the manual annotation. Because more than one resolution parameter can yield the correct number of layers, for each dataset we report the median ARI across all parameter settings that had the correct layer number. Across all 12 datasets, we obtained a median ARI of 0.518. As a baseline, we ran the same pipeline for $\lambda = 0$ corresponding to nonspatial clustering.

For benchmarking, we ran Giotto's HMRF, BayesSpace, SpaGCN and SpiceMix on the same data. For Giotto, we used v.1.1.0 and followed the workflow on the package's site (https://rubd.github.io/Giotto_site/articles/tut11_giotto_hmrf.html). The workflow for all samples was the same. First, genes that were not detected in at least ten cells were filtered out. Next data were first count-normalized to a fixed scale factor and log-transformed, followed by gene-wise and cell-wise $z$-scaling. A spatial network was created using default settings (Delaunay method with the maximum distance cutoff parameter set to 'auto'). We identified genes with a spatially coherent expression pattern using the BinSpect method using default parameters and selected genes with an adjusted $P < 0.1$ and with BinSpect scores in the top one percentile, yielding approximately 160 genes per sample. HMRF was run on the spatial network and spatial genes were identified, with the number of domains set to the number of layers present in the manual annotation. Based on the author's recommendations (https://cran.r-project.org/web/packages/smfishHmrf/smfishHmrf.pdf), we tested $\beta$ values (HMRF regularization parameter) from 0 to 100 in increments of 2. We then selected $\beta = 16$, which gave the best median ARI across all samples.

For BayesSpace (v.1.5.1), we followed the authors' vignette (https://edward130603.github.io/BayesSpace/articles/maynard_DLPFC.html). The workflow for all samples was the same. HVGs were first identified by modeling the mean variance relationship for each gene as implemented in the scran package. PCA was then performed on the top 2,000 variable genes. We ran BayesSpace on the first 15 principal components, with the number of clusters set to the number of layers present in the manual annotation. Clustering was performed at default settings for Visium data ($t$-distributed error model with 50,000 Markov chain Monte Carlo iterations, a burn-in period of 1,000 iterations and a gamma smoothing parameter of 3).

The results for SpiceMix[11] differ from those reported in the original paper for three reasons: (1) we tested the latest version of their software, which the authors described as more accurate (https://github.com/ma-compbio/SpiceMix) than the previous version used in their paper; (2) the original SpiceMix study only benchmarked clustering performance on a small subset of the DLPFC dataset (4 of 12 samples). In contrast, we followed the common practice of testing on all 12 samples[23,25–27]; (3) in the original study, SpiceMix was trained jointly on the four selected samples, whereas we followed the precedent

of analyzing each sample independently[23,25–27]. In our benchmarking analysis of SpiceMix, we followed the authors' tutorial notebook (https://github.com/ma-compbio/SpiceMix/blob/master/SpiceMix/main_Maynard2021.ipynb) from the latest version of SpiceMix (GitHub commit ID: aea69f8).

Because of the stochasticity inherent in deep-learning methods, their performance can be sensitive to the choice of random seeds[50]. We therefore tested deep-learning methods (SpaGCN, STAGATE and GraphST), along with BANKSY, across 11 manual seeds and reported the median performance across seeds. We used eight seeds commonly used in machine learning, {0, 1, 2, 10, 42, 100, 123, 1234}, as well as three seeds used by the authors of the algorithms we tested, that is, {41, 1000, 2020}.

For SpaGCN (v.1.2.2), we followed the authors' tutorial notebook (https://github.com/jianhuupenn/SpaGCN/blob/master/tutorial/tutorial.ipynb), which demonstrated its analysis on sample no. 151673. We adapted the notebook for the 11 other samples, changing only the number of desired clusters (for samples with five cortical layers) but keeping all training parameters and model hyperparameters the same (200 epochs of training and a learning rate of 0.05). While the clustering results obtained were not identical to those reported in Hu et al.[25], comparisons of the ARIs obtained showed close correspondence to the reported values.

For STAGATE (v.1.0.1), we followed the authors' documentation (https://stagate.readthedocs.io/en/latest/T1_DLPFC.html). The data were preprocessed by filtering the top 3,000 HVGs, followed by normalization and log-transformation. To construct the spatial network, we used the recommended cutoff radius of 150 when testing on the 12 DLPFC samples. As recommended once again, we used mclust[72] in the clustering step when testing on the DLPFC dataset. Although STAGATE does not implement an option to refine the cluster labels (label smoothing), we applied the SpaGCN's refinement procedure. Similarly, we maintained the same training parameters and model hyperparameters across all tests (500 epochs and a learning rate of $10^{-4}$), and only changed the desired number of clusters.

For GraphST (v.1.1.1), we followed the authors' documentation (https://deepst-tutorials.readthedocs.io/en/latest/Tutorial%201_10X%20Visium.html), which demonstrated their workflow on sample no. 151673. In their preprocessing step, feature selection was conducted by filtering the top 3,000 HVGs, followed by normalization, log-transformation and scaling. We used the authors' default parameters in training the GraphST network. In the clustering step, the mclust algorithm was used as recommended by the authors. Although their code includes an optional refinement step, we used SpaGCN's refinement procedure for consistency across methods. Throughout our benchmarking, we applied the same procedure on all runs across the 11 seeds by maintaining GraphST's default training parameters and model hyperparameters (600 epochs of training, learning rate of 0.01) and fixed the number of clusters to be equal to that in the ground truth.

### STARmap data

The data for the mouse visual cortex 1,020 gene dataset was obtained as specified in Wang et al.[21]. To obtain the reference domain annotations, we started with the cell-type annotations provided by Wang et al.[21]. We performed three rounds of label smoothing, in each round relabeling a cell if more than eight of its 15 nearest neighbors belonged to a different cell type. After smoothing, we assigned cells with the oligodendrocyte, eL6-2, eL4 and eL2/3 cell-type labels to the corpus callosum, L6, L4 and L2/3 domains, respectively. Cells located to the left of the corpus callosum domains were labeled hippocampus, while cells positioned between the L6 and L4 domains were labeled L5. The L1 domain in this dataset only encompassed a small number of cells and was not amenable to label smoothing. Hence we manually separated the L1 domain from the L2/3 domain. The resulting domain labels closely

matched the positions of the domain boundaries that could be visually identified in Wang et al.[21] (see Extended Data Fig. 7a for a comparison of cell-type and domain labels).

For domain segmentation, we used the code provided by Wang et al.[21] for all normalization and scaling steps. We then performed BANKSY clustering using default settings, increasing the resolution in increments of 0.1 until seven clusters (corresponding to the number of spatial domains in the dataset) were obtained. This occurred at a clustering resolution of 0.9. Analyses using other methods, and the software versions used, are described in Methods, 'Human DLPFC 10x Visium data'. For SpaGCN, we used neighboring cells' contribution = 1 as recommended in Hu et al.[25] for this dataset.

For STAGATE, we adjusted the cutoff radius parameter to 500 to construct the spatial network, which yields an average of around ten neighbors per cell (mean = 10.42), as recommended in most tutorials in their documentation (https://stagate.readthedocs.io/en/latest/T3_Slide-seqV2.html). Note that STAGATE's documentation provides a tutorial on the STARmap dataset (https://stagate.readthedocs.io/en/latest/T9_STARmap.html) in which a different cutoff radius of 400 was set. However, this setting led to a lower performance (ARI = 0.544) compared to using a larger cutoff radius (ARI = 0.682); thus, we report STAGATE's performance using the latter setting.

As described in the previous section, we tested the deep-learning methods (SpaGCN, GraphST and STAGATE), along with BANKSY, on 11 commonly used random seeds and reported the median ARIs across these 11 seeds.

We note that both BANKSY and GraphST split the L2/3 region vertically. We performed additional analyses to investigate the possibility that this split was because of differential data quality between tiles in the upper and lower halves of the imaged region. First, we examined the spatial distribution of the number of detected genes (NODG) in each cell, a common quality control metric for data quality, but did not observe a tiling pattern in NODG that could explain the L2/3 split. We also did not find any statistically significant differences in NODG between the upper and lower clusters identified by BANKSY for any of the cell types (false discovery rate (FDR) $q > 0.2$; Benjamini–Hochberg adjustment for multiple testing). The NODG for each cell type with at least ten cells represented was compared using a two-tailed Wilcoxon rank-sum test. Thus, we did not find evidence that data quality differences drove the split in L2/3. Next, we asked if the split in L2/3 could be attributed to specific DEGs. We identified four genes with significant upregulation in the lower subcluster (*Trim32*, *Nr4a1*, *Nrgn* and *2900055J20Rik*; FDR $q$(Benjamini–Hochberg-adjusted) < 0.02, two-tailed Wilcoxon rank-sum test) and two genes with upregulation in the upper subcluster (*Hlf*, *Bcl6*; FDR $q$(Benjamini–Hochberg-adjusted) < 0.04, two-tailed Wilcoxon rank-sum test), which may explain the splitting of this tissue domain into two spatially distinct clusters by BANKSY and GraphST. Nevertheless, for quantitative benchmarking purposes, we treated the split in L2/3 as an erroneous prediction by BANKSY and GraphST, and it was therefore reflected in the ARI metric.

### Human healthy colon CosMx data

CosMX Spatial Molecular Imager data of the human healthy colon were downloaded from the NCBI Gene Expression Omnibus database (accession no. GSM7473683 (ref. 73). The data include 50,966 cells profiled across 960 genes. For the downstream analysis, we kept cells whose total transcript count was between the 5th and 98th percentile of the total transcript counts across all cells. These data had two sets of contiguous regions of interest (ROIs) (forming connected regions), along with some ROIs with scattered cells. We selected the ROIs corresponding to the larger of these connected regions (top part of the image; ROIs: 1–7, 13–16), yielding a total of 32,765 cells. Transcript counts per cell were normalized to the median total transcript count across all cells and were then natural log-transformed.

We then ran BANKSY and nonspatial clustering using default parameters ($\lambda \in \{0, 0.2\}$). We used Harmony (v.0.1.1) to perform batch correction across ROIs (run on the principal component scores computed from the BANKSY matrix, with ROI as the batch variable). Harmony-corrected PCA embeddings were then clustered.

To obtain cluster annotations, we ran the reference-based single-cell annotation method SingleR (v.2.2.0)[66]. We used scRNA-seq data from the Gut Cell Atlas[74] as a reference, using only cells from healthy adult donors. Processed counts and high-resolution cell-type labels were used as a reference in SingleR, with a Wilcoxon rank-sum test used to identify the top 15 DEGs from each pairwise comparison between labels. Using labels returned from SingleR, we annotated each cluster based on the majority label of all cells within a given cluster.

To identify markers of the macrophage subpopulation uniquely identified by BANKSY (Extended Data Fig. 2a, macrophage.2), we performed differential gene expression analysis with a Wilcoxon rank-sum test between the two macrophage subpopulations (macrophage.1 and macrophage.2; Extended Data Fig. 2c).

### Human healthy intestine CODEX data

Processed CODEX multiplexed imaging data of the healthy human intestine were downloaded from Hickey[75]. To identify tissue units, we used data from the ileum, right and transverse colon regions from donor B0012. We ran nonspatial clustering and BANKSY in domain segmentation mode using default parameters ($\lambda \in \{0, 0.8\}$). To jointly cluster the regions, we ran Harmony (v.0.1.1)[53] on the principal component scores computed from the BANKSY matrix, with region as the batch variable. Harmony-corrected PCA embeddings were then clustered, with the resolution chosen to yield three clusters based on tissue unit annotation (mucosa, muscle and submucosa) provided by Hickey et al.[8] (Extended Data Fig. 8a–c). To identify communities in the ascending colon region from donor B006, we followed the same procedure (without Harmony), increasing only the resolution to yield eight clusters to match the number of communities annotated in Hickey et al.[8] (Extended Data Fig. 8d,e).

### Simulated data

The simulated dataset was created using the scDesign2 tool[54] using default parameter settings (v.0.1.0). scDesign2 fits a probabilistic model to cell-type-specific RNA count distributions in real STARmap data (based on the cell type labels from Wang et al.[21]) and then generates simulated counts based on this model. In the simulation, we retained the layer structure, layer-specific cell-type composition and cell density observed in real STARmap data. We varied the size of the simulated gene set from 400 to 1,020 genes by selecting random subsets of the full 1,020-gene STARmap panel. We generated three replicate datasets for each condition using different random seeds and reported the median result across the seeds. We also benchmarked six existing cell typing or tissue domain segmentation algorithms on the same datasets.

The BANKSY runs were performed at the default parameters. The resolution parameter is typically tuned by the user based on the desired ontological resolution (cell lineage, type, subtype or state) or the desired number of clusters. To obtain the desired number of cell types or tissue domains, we varied the resolution from 1.5 to 2.5 for cell typing and 0.5 to 1.5 for domain segmentation. All clustering runs matching the desired number of clusters were used to compute the median ARI.

Most parameters for the other methods were also the defaults, as in the 'Mouse Hippocampus VeraFISH data' and 'Human DLPFC 10X Visium data' sections of this study. We ran BayesSpace (v.1.5.1) on the top 20 principal components computed on the normalized gene–cell matrix and a $k$-nearest neighbor graph with $k = 15$. For clustering, the $t$-distributed error model was used with default parameters. For Giotto (v.1.1.0), we created a spatial network with $k = 15$ nearest

neighbors using the default settings (Delaunay method with the maximum distance cutoff set to 'auto'). We tested $\beta$ (HMRF regularization parameter) from 0 to 50 in increments of 2 and selected all runs that yielded the correct number of clusters. For FICT, we used the same modified code with $k = 10$ nearest neighbors because $k = 15$ resulted in underclustering. Interestingly, while FICT allows the user to specify the number of output clusters, the actual number of clusters returned does not exactly match the specified number, which instead serves as an upper bound. For SpiceMix, we once again used the latest version of their package (GitHub commit ID: aea69f8). We first created a neighborhood graph with $k = 15$ neighbors. We used the defaults for the regularization parameter $\lambda_{\Sigma} = 1 \times 10^{-6}$, the number of metagenes $k = 15$ and the number of iterations (200). We followed the authors' tutorial notebook (https://github.com/ma-compbio/SpiceMix/blob/master/SpiceMix/main_Maynard2021.ipynb) for clustering. For domain finding, we applied smoothing to the metagene matrix and labels as implemented by the authors. The smoothing radius was set at the 90th percentile of the 15th nearest neighbor distance. For cell typing, smoothing was not performed.

On all datasets in this simulated benchmark, BANKSY substantially outperformed all competing methods at domain segmentation ($\lambda = 0.8$; Extended Data Fig. 3a–c). BANKSY also outperformed all other methods on all datasets at cell typing ($\lambda = 0.2$; Extended Data Fig. 3d–f). This result further supports the generality and high accuracy of BANKSY, and demonstrates the algorithm's ability to perform both functions merely by varying a single hyperparameter.

### Scalability analysis

To benchmark the scalability of BANKSY, we measured the elapsed time and peak memory of BANKSY and other methods on datasets of up to 2 million cells. All methods were benchmarked on a virtual machine with 128 GB and 16 threads (Ubuntu v.18.04.5 LTS).

We used a Vizgen MERSCOPE mouse brain dataset consisting of nine coronal slices from three biological replicates[44]. The size of the slices ranged from 70k to 88k cells, with 483 genes profiled per slice. To form smaller datasets, we cropped one slice (slice 2, replicate 1) to 4k, 8k, 16k, 33k and 83k cells (5% to 100% of the data, logarithmically spaced). To form larger datasets, we concatenated multiple slices to form datasets of sizes 100k, 250k, 500k, 1 million and 2 million cells. As the total number of cells from all nine slices was less than a million, we duplicated slices whenever necessary. In concatenating the spatial coordinates for different slices, we ensured that the $k$-nearest spatial neighbors for each cell in a slice belonged to the same slice.

We benchmarked BANKSY (v.0.1.3) against conventional nonspatial clustering (Seurat v.4.0.5), FICT (v.1.0.0), MERINGUE (v.1.0), SpiceMix (without GPU, GitHub commit ID: aea69f8, latest version of the code), Giotto's HMRF (v.1.1.0), SpaGCN (v.1.2.2), STAGATE (v.1.0.1) and GraphST (v.1.1.1). The analysis pipeline for each method is similar to what is described in the 'Mouse Hippocampus VeraFISH data' and 'Human DLPFC 10X Visium data' sections. The elapsed time and peak memory were measured using the peakRAM package (v.1.0.2) for BANKSY, conventional clustering, MERINGUE and Giotto HMRF and the /usr/bin/time command for FICT and SpiceMix. We terminated any run after 16 h of elapsed time.

Of the six methods benchmarked, only BANKSY and conventional nonspatial clustering scaled to over a million cells. MERINGUE failed on datasets equal to or larger than 33k cells because of R's indexing limits ('long vector not supported'). FICT failed on datasets equal to or larger than 250k cells because there was insufficient memory (NumPy 'ArrayMemoryError'). Giotto's HMRF failed on datasets equal to or larger than 500k cells due to a recursion error ('maximum recursion depth exceeded in comparison'), despite setting the stack size to unlimited (ulimit -s) and increasing the program's default recursion limit by tenfold to 500,000 (https://github.com/RubD/Giotto/blob/master/inst/python/reader2.py#L137).

In the scalability analysis performed on the GPU, we benchmarked the runtimes of algorithms compatible with GPU on an NVIDIA Tesla T4 with 16 GB of random access memory (CUDA v.12.0) using the same datasets as in the CPU benchmarks (Extended Data Fig. 10). We ran all methods on increasingly large datasets until an out-of-memory error was encountered.

### Effect of parameter variation

We assessed the effect of parameter variation on the output of BANKSY across four datasets obtained using distinct spatial technologies: the 10x Visium Human DLPFC dataset (Supplementary Fig. 25); the MERFISH mouse hypothalamus dataset (Supplementary Fig. 26); the VeraFISH dataset (Supplementary Figs. 27–34), and the Slide-seq v.1 dataset (Supplementary Figs. 35–40). For each dataset, we varied the $k_{geom}$, $\lambda$, number of principal components and $k_{expr}$ parameters around the default settings.

Because there is a known ground truth for the 10x Visium dataset, we quantitatively assessed the effect of parameter variation by computing the median ARI across all 12 samples for a given set of parameters. In particular, we varied the number of principal components from 15 to 30 in increments of 5, $k_{expr}$ from 30 to 70 in increments of 10 and the number of highly variable features used from 1,000 to 3,000 in increments of 500. $\lambda$ was fixed at 0.2 while $k_{geom}$ was fixed at 18 corresponding to all spots up to the second-order neighbors of the index spot. We compared each of the parameters in a pairwise fashion, yielding 65 different parameter combinations (Supplementary Fig. 25). Across all combinations, the median ARI remained high (median of medians = 0.512) and exhibited little variation (IQR = 0.015), suggesting that BANKSY is robust to variation in the input parameters (Supplementary Fig. 25).

### Statistics and reproducibility

Most datasets shown in this article are from publicly available datasets (Data availability section). For these datasets and the VeraFISH dataset, no statistical method was used to predetermine sample size and no data were excluded from the analyses. Full sample sizes are as follows: Slide-seq and Slide-seq v.2 datasets: 25,551 and 39,496 cells, respectively; MERFISH mouse hypothalamus data: all 11 naive animals (485,657 cells); scRNA-seq study of MODs in the mouse hypothalamus, all cells labeled MODs by the authors of the original study (6,611 cells); VeraFISH mouse hippocampus dataset: all 10,994 cells; corresponding scRNA-seq analysis: all cells in neuronal clusters (as labeled by the authors, 2,386 cells) and all cells labeled 'Oligo' (231 cells); MERSCOPE colorectal cancer data: all 677,451 cells; DLPFC Visium dataset: all 12 samples; STARmap dataset: all 1,207 cells annotated by the authors; CODEX data: all 33,958 cells from the ileum, all 25,403 cells from the right colon region and all 27,784 cells from the transverse colon region from donor B0012 for tissue domain annotation. For community annotation: 38,371 cells from donor B006 in the ascending colon region; simulated data: we created three samples for each gene set condition (400, 600, 800 and 1,020 genes), including 4,996 cells each.

The only data exclusions were as follows. In the mouse hypothalamus MERFISH data, we removed cells marked 'ambiguous' (as per the original study) and the gene *Fos*, which contained 'NaN' entries. In the DLPFC Visium data, we removed spots marked 'ambiguous' by the authors. In the CosMx data, there were two major connected regions of field of view, along with some field of views with scattered cells. We used the larger of these connected regions for analysis.

The DLPFC data include four samples (two pairs of 'spatial replicates'[9]) from each of three patients, resulting in 12 datasets. Following the usual practice in the field, we reported median statistics and box plots over these 12 samples. For the simulated data, we generated three replicates for each gene count condition (400, 600, 800, all 1,020) and reported the median values of ARIs over these, replicated for all tested

methods. In all other datasets, the entire dataset was clustered and analyzed as a single dataset, after appropriate quality control.

Randomization was used in the simulated dataset: the genes to subset were picked randomly to generate 400, 600 and 800 gene sets of the full 1,020 gene set. No other randomization was used in this study (deterministic subsetting methods like HVGs were used for gene selection, and quality control metrics like NODG were used for cell subsetting).

No blinding was applicable in this study because no sample group allocation was performed.

### Reporting summary

Further information on research design is available in the Nature Portfolio Reporting Summary linked to this article.

## Data availability

For the SlideSeq mouse cerebellum dataset, data were obtained from https://github.com/RubD/spatial-datasets/blob/master/data/2019_slideseq_cerebellum/raw_data/slideseq_cerebellum_urls.txt. We used the BeadLocationsForR and MappedDGEForR.csv files. For Slide-seq v.2, we obtained the data from the Broad Institute Single Cell Portal at https://singlecell.broadinstitute.org/single_cell/study/SCP948. The MERFISH mouse hypothalamus data were obtained from https://doi.org/10.5061/dryad.8t8s248. The Vizgen MERSCOPE data for the colon cancer 1 sample were obtained from https://info.vizgen.com/ffpe-showcase. The CosMX SMI data of the human healthy colon was obtained from the NCBI Gene Expression Omnibus database (accession no. GSM7473683). The processed CODEX multiplexed imaging data of the healthy human intestine were downloaded from Hickey[75]. The mouse hippocampus data were collected using the VeraFISH assay (Veranome Biosystems) as described in the Methods, 'Mouse hippocampus VeraFISH data'. Data are available by running the command data(hippocampus) in the BANKSY package or directly from https://github.com/prabhakarlab/Banksy/blob/bioc/data/hippocampus.rda. The 10x Visium data of the DLPFC were obtained from the spatialLIBD project (http://spatial.libd.org/spatialLIBD)[71]. The STARmap mouse visual cortex data were obtained from http://clarityresourcecenter.org.

## Code availability

The Banksy R package can be obtained from https://github.com/prabhakarlab/Banksy; the scripts to reproduce the R analyses can be found at https://github.com/jleechung/banksy-zenodo and require R v.3.5 or higher to function. The Python version is available from https://github.com/prabhakarlab/Banksy_py, while the IPython notebooks to reproduce our analysis on Slide-seq v.1, Slide-seq v.2 and STARmap are available on the Banksy-manuscript branch of the BANKSY_py GitHub repository. The R and Python scripts to reproduce the analysis can also be found in the Zenodo archive at https://doi.org/10.5281/zenodo.10258795 (ref. 76).

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

## Acknowledgements

V.S. and N.C. received funds from the A*STAR Graduate Academy and A*STAR Career Development Award Project no. 202D800010. J. Liu was supported by the Singapore Ministry of Health's National Medical Research Council under its Open Fund - Young Individual Research Grant Project no. MOH-000239-00 and A*STAR. K.H.C. was funded by an A*STAR grant no. I1801E0029, NMRC grant no. OFIRG20nov-0056 and a Singapore National Research Foundation Competitive Research Programme grant no. NRF-CRP25-2020-0001. S.P., L.L. and W.K.C. were funded by an Industry Alignment Fund-Pre-positioning Programme grant no. H18/01/a0/020 from A*STAR; S.P. was additionally funded by an NMRC grant no. OFIRG21jun-0090. The funders had no role in study design, data collection and analysis, decision to publish or preparation of the manuscript. We thank B. Ranjan, G. Yeo, X. Zhou and R. Baskar for useful discussions, and to J. Yi Wong for help with Linux administration and scripting.

## Author contributions

BANKSY was conceived by V.S. and S.P., with help from H.K.L. and K.H.C., and subsequent conceptual contributions and benchmarking by N.C., J. Lee and Y.Y. J. Lee and V.S. designed and wrote the R package. N.C. and Y.Y. developed the Python implementation. J. Lee, Y.Y., V.S. and N.C. performed the benchmarking of the existing methods, with helpful insights from J. Liu, W.K.C. and L.L. J.A. helped with the interpretation of the results. The VeraFISH data were provided by Y.-C.C. and E.M.L.T. V.S., N.C., J. Lee, Y.Y., S.P. and K.H.C. wrote the manuscript. All authors edited the manuscript.

## Competing interests

E.M.L.T. and Y.-C.C. are employees of Veranome Biosystems. The other authors declare no competing interests.

## Additional information

**Extended data** is available for this paper at https://doi.org/10.1038/s41588-024-01664-3.

**Correspondence and requests for materials** should be addressed to Kok Hao Chen or Shyam Prabhakar.

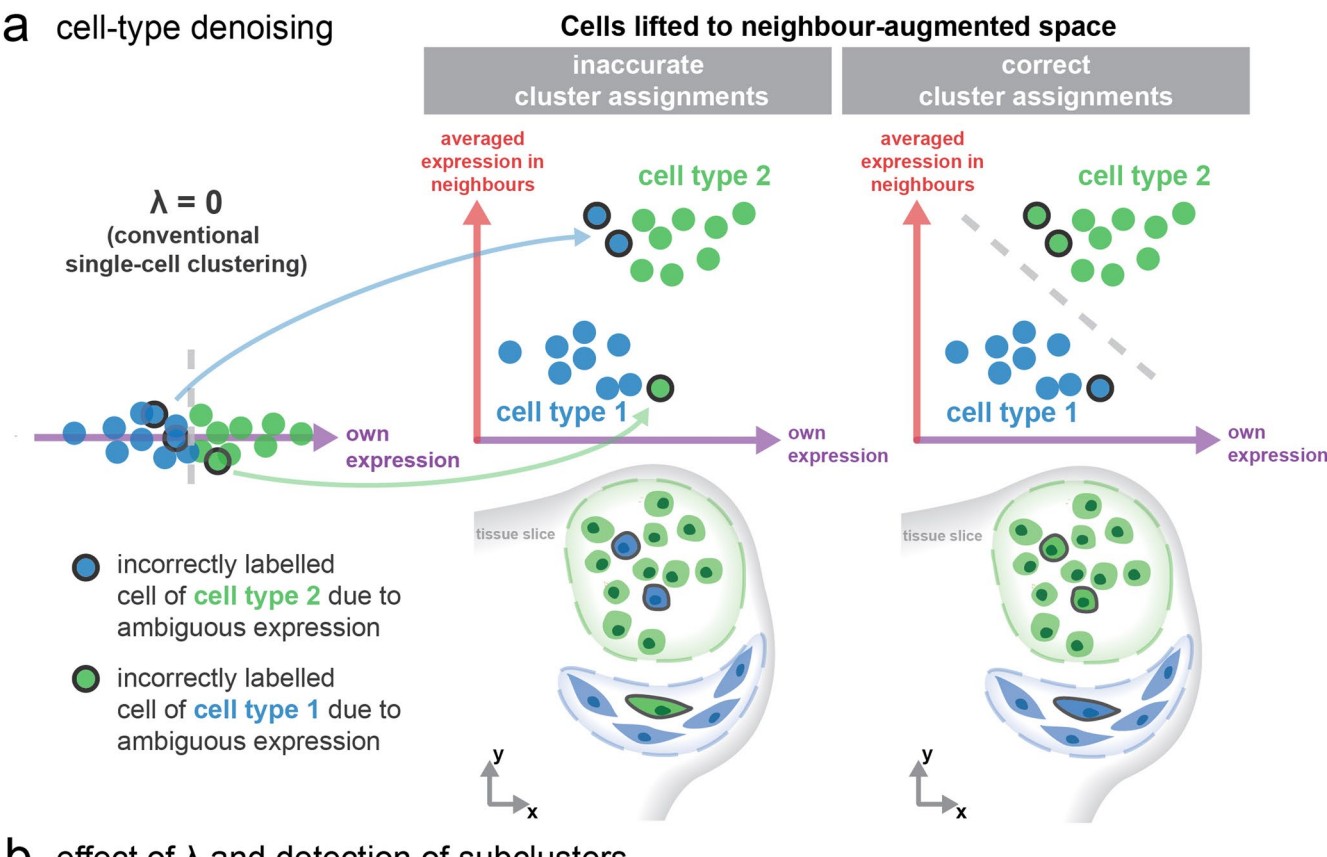

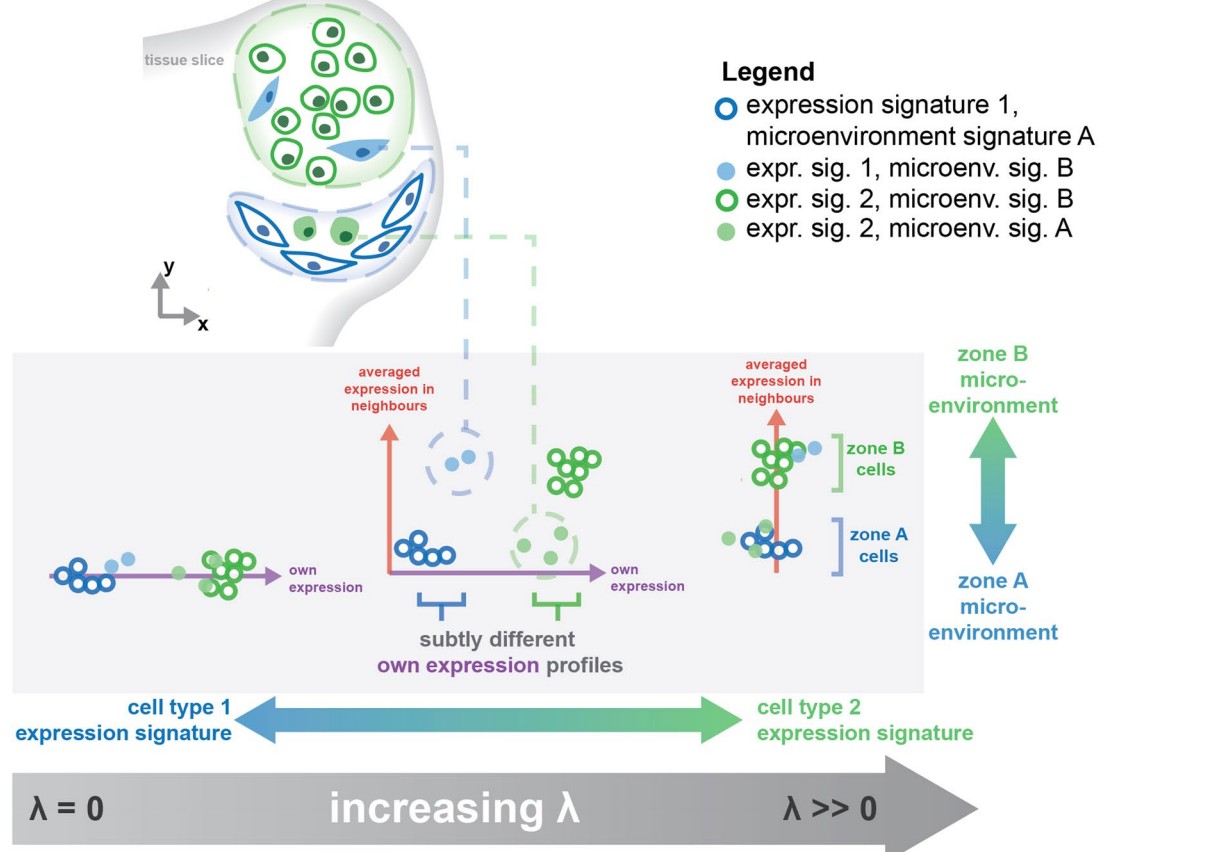

**Extended Data Fig. 1 | See next page for caption.**

**Extended Data Fig. 1 | Detailed schematic to aid conceptual understanding of BANKSY.** How BANKSY may lead to more accurate cluster assignments relative to non-spatial clustering ($\lambda=0$) by accounting for local microenvironmental information. (a) Cells near the decision boundary in own expression space alone get better separated in the neighbor-augmented space. We have used only the mean neighborhood expression for illustration purposes, but the same arguments apply to the AGF feature space. (b) Top – Schematic of tissue slice shows physical locations of two cell types (1 and 2) in two zones (A and B). Zone A contains mainly cell type 1 but some of cell type 2. Zone B contains mainly cell type 2 but some of cell type A. Bottom – Effect of increasing $\lambda$: at $\lambda=0$ (bottom left), only the cell's transcriptome is used, making it identical to conventional unsupervised clustering analysis. As $\lambda$ increases (bottom middle), subsets of cells in different environments with subtly different transcriptomes become easier to tease out. For instance, the blue circles – filled and empty – correspond to the same cell type in two different neighborhoods, possessing subtle transcriptomic differences. A similar effect is also shown for the green cell type. At higher lambdas (bottom right), a zone or spatial-domain segmentation effect occurs where zones representing different microenvironments are clustered separately and can comprise multiple cell types per zone.

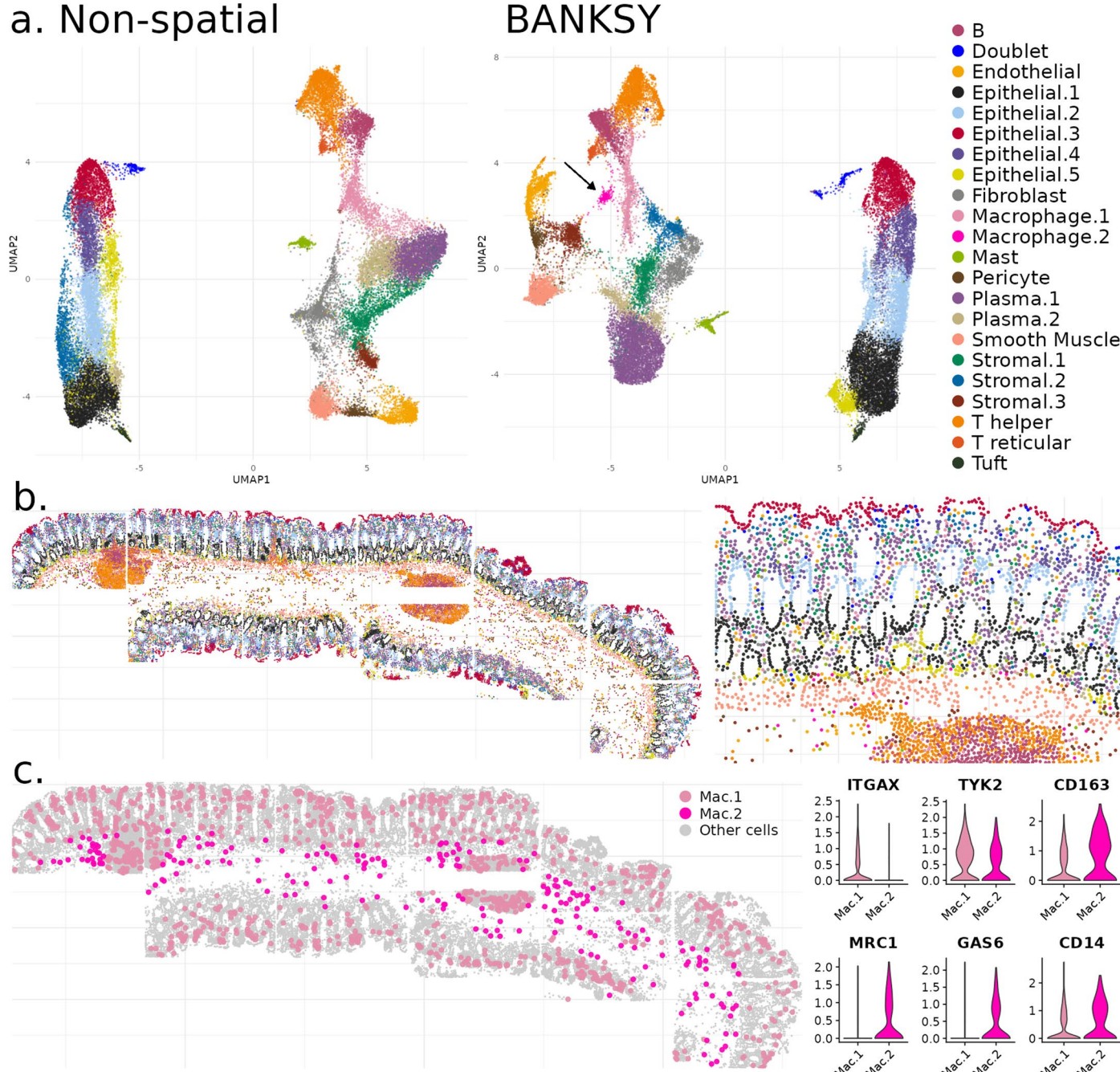

**Extended Data Fig. 2 | BANKSY cell typing on a CosMX colon tissue dataset.**
a) Non-spatial and BANKSY UMAPs colored by their respective clusters, with cell type labels for BANKSY clusters. b) Spatial plots for the full dataset and a single ROI colored by BANKSY clusters. c) Spatial distribution of macrophages, with the Mac.2 cluster enriched in the submucosa. Violin plots show DE genes between the two macrophage subpopulations. Mac.1 n = 1177, Mac.2 n = 214.

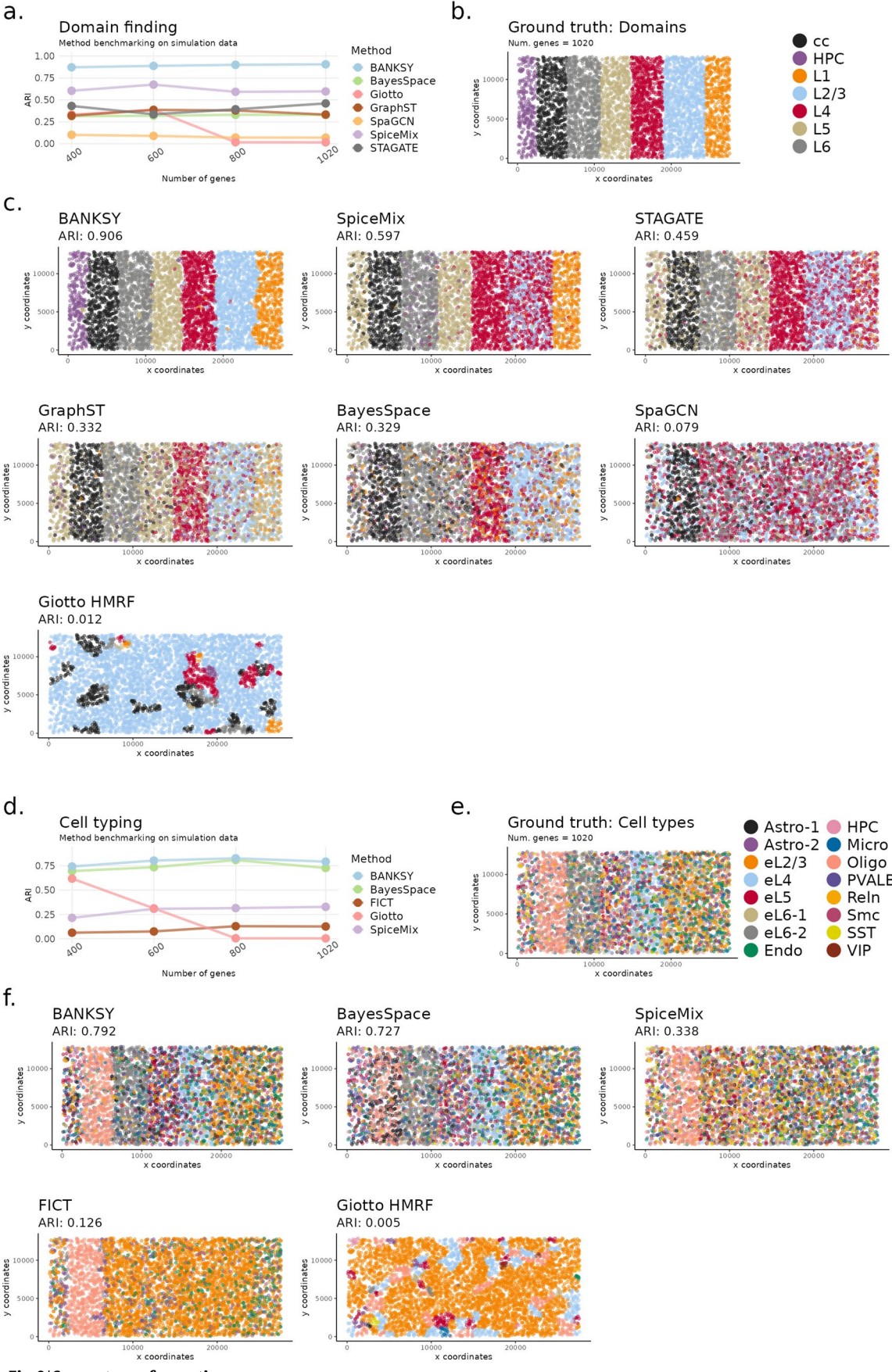

**Extended Data Fig. 3 | See next page for caption.**

**Extended Data Fig. 3 | Benchmarking of domain segmentation (a-c) and cell type clustering (d-f) on data simulated from the STARMAP 1020 gene mouse visual cortex dataset.** Data were simulated using scDesign2, with gene expression characteristics modeled after the 16 cell types in the STARmap data. Cell type compositions across the 7 domains (vertical strips) were modeled after cell type compositions in each domain in the STARmap data. (a) ARIs (median of 3 pseudorandom number generator seeds) of each method for domain segmentation as a function of gene number. The methods were used to generate domain labels by setting the number of requested clusters to 7. (b) Ground truth annotations for domain segmentation. (c) Qualitative comparisons across methods for a representative 1020 gene dataset. (d) ARIs (median of 3 seeds) of each method for cell type clustering as a function of gene number. The methods were used to generate cell type labels by setting the number of requested clusters to 16. (e) Ground truth annotations for cell type clustering. (f) Qualitative comparisons across methods for a representative 1020 gene dataset.

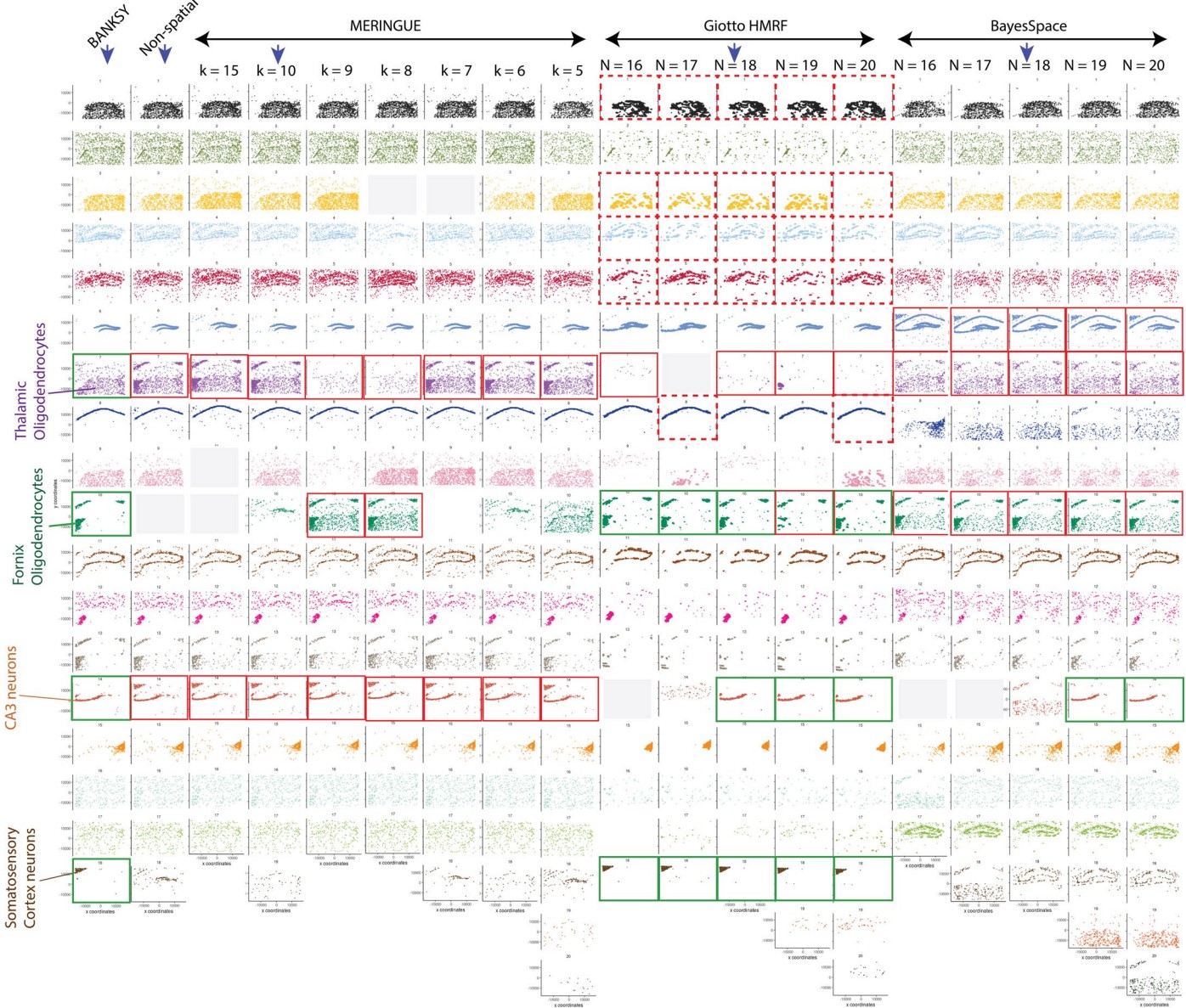

**Extended Data Fig. 4 | Comparing BANKSY and non-spatial clustering results on the VeraFISH data to MERINGUE, Giotto (HMRF) and BayesSpace (HMRF).** The number of clusters was varied by adjusting parameters related to clustering resolution (number of nearest neighbors, k, in the graph for MERINGUE, and the number of clusters, N, for SpaGCN and FICT). Each column represents one clustering run for the indicated method at a given effective resolution. Blue arrows indicate the runs compared in Fig. 4. Green and red boxes indicate success and failure to identify the known neuronal or oligodendrocyte subclusters (Fig. 4; Supplementary Fig. 18). Conventional non-spatial clustering and MERINGUE (at any number of clusters) failed to separate both the oligodendrocyte subclusters and the neuronal subclusters. While Giotto separated the CA3 neurons (orange) from the somatosensory cortex neurons (brown), and at some resolutions was

able to identify the fornix oligodendrocytes (dark green), it failed to identify the thalamic oligodendrocytes (purple) at all resolutions. Importantly, Giotto tended to cause clusters to form spatially disconnected groups of cells ('patches') that do not correspond to any known anatomical structures in the thalamus or hippocampal formation (dashed red boxes). BayesSpace merged several neuronal clusters: at N = 16 - 18, it merged the somatosensory cortex neurons with the CA3, CA1 and dentate gyrus neurons (blue clusters; all anatomical regions labeled in Fig. 4 in the main text), and at N = 19 - 20, it separated out the CA3 neurons, from the somatosensory cortex neurons, but merged the CA1 and dentate gyrus neurons. It also failed to separate out the thalamic oligodendrocytes from the fornix oligodendrocytes (red boxes around the dark green cluster).

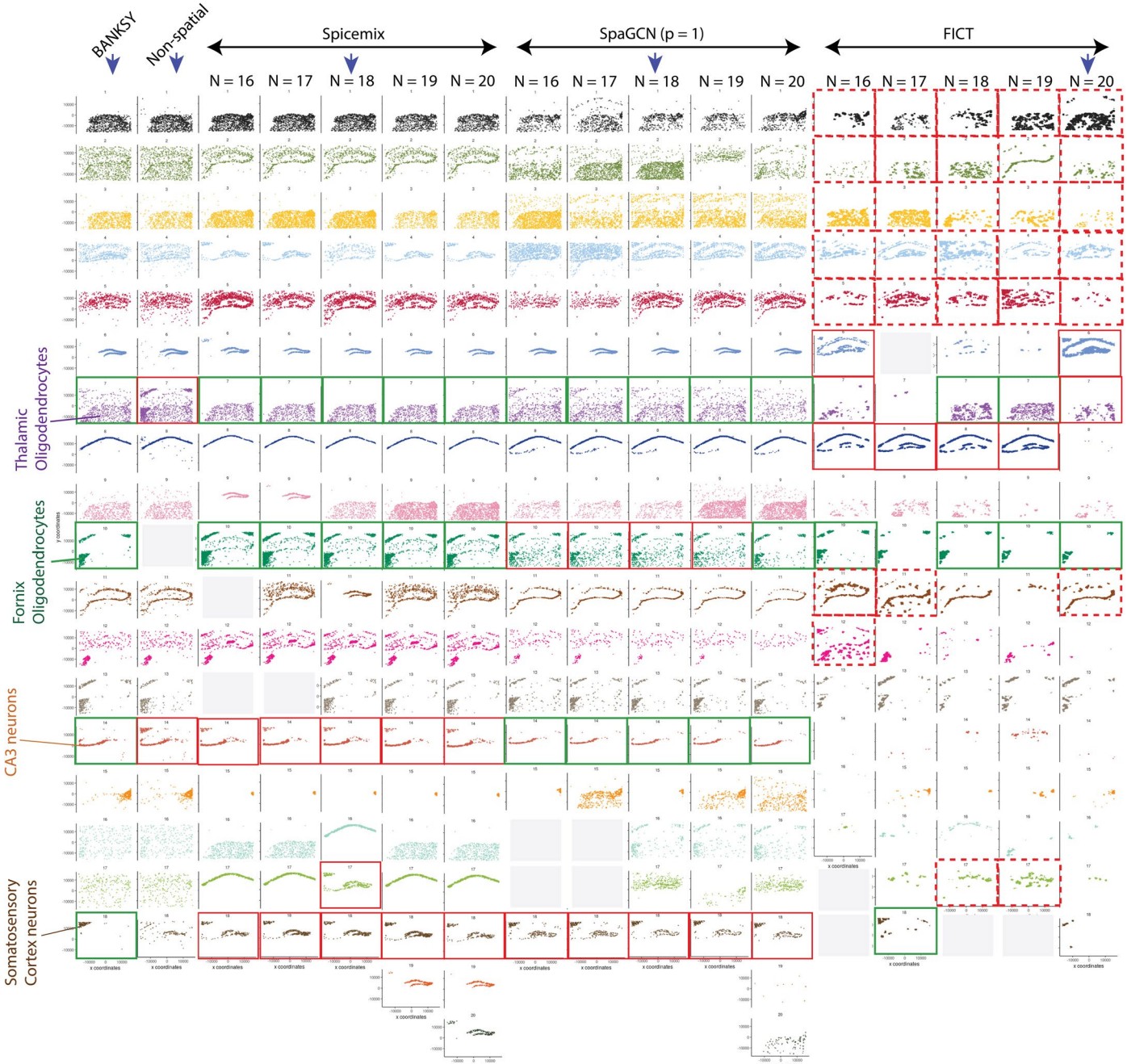

**Extended Data Fig. 5 | Comparing BANKSY and non-spatial clustering results on the VeraFISH data to SpiceMix (HMRF), SpaGCN and FICT (HMRF).** Similar to Extended Data Fig. 4. As before, the number of clusters was varied by adjusting parameters related to clustering resolution. Each column represents one clustering run for the indicated method at a given resolution (controlled by the number of clusters input parameter, N). Blue arrows indicate the runs compared in Fig. 4. Green and red boxes indicate success and failure (respectively) to identify the known neuronal or oligodendrocyte subclusters (Fig. 4; Supplementary Fig. 18). SpiceMix correctly separated the thalamic and fornix oligodendrocytes, merged the somatosensory cortex neurons with either

the CA3 neuron (orange clusters) or the dentate gyrus neurons (brown and light green clusters). SpaGCN was able to identify the CA3 neurons, but merged the somatosensory cortex neurons with cells in the dentate gyrus (brown cluster), and was unable to separate the thalamic and fornix oligodendrocytes (red boxed green clusters). FICT, like Giotto in Extended Data Fig. 4, tended to group cells into disconnected spatial patches that do not conform to known anatomical structures (dashed red boxes), and tended to merge the dentate gyrus, CA1, CA3 and somatosensory cortex neurons into one or two clusters (light and dark blue clusters).

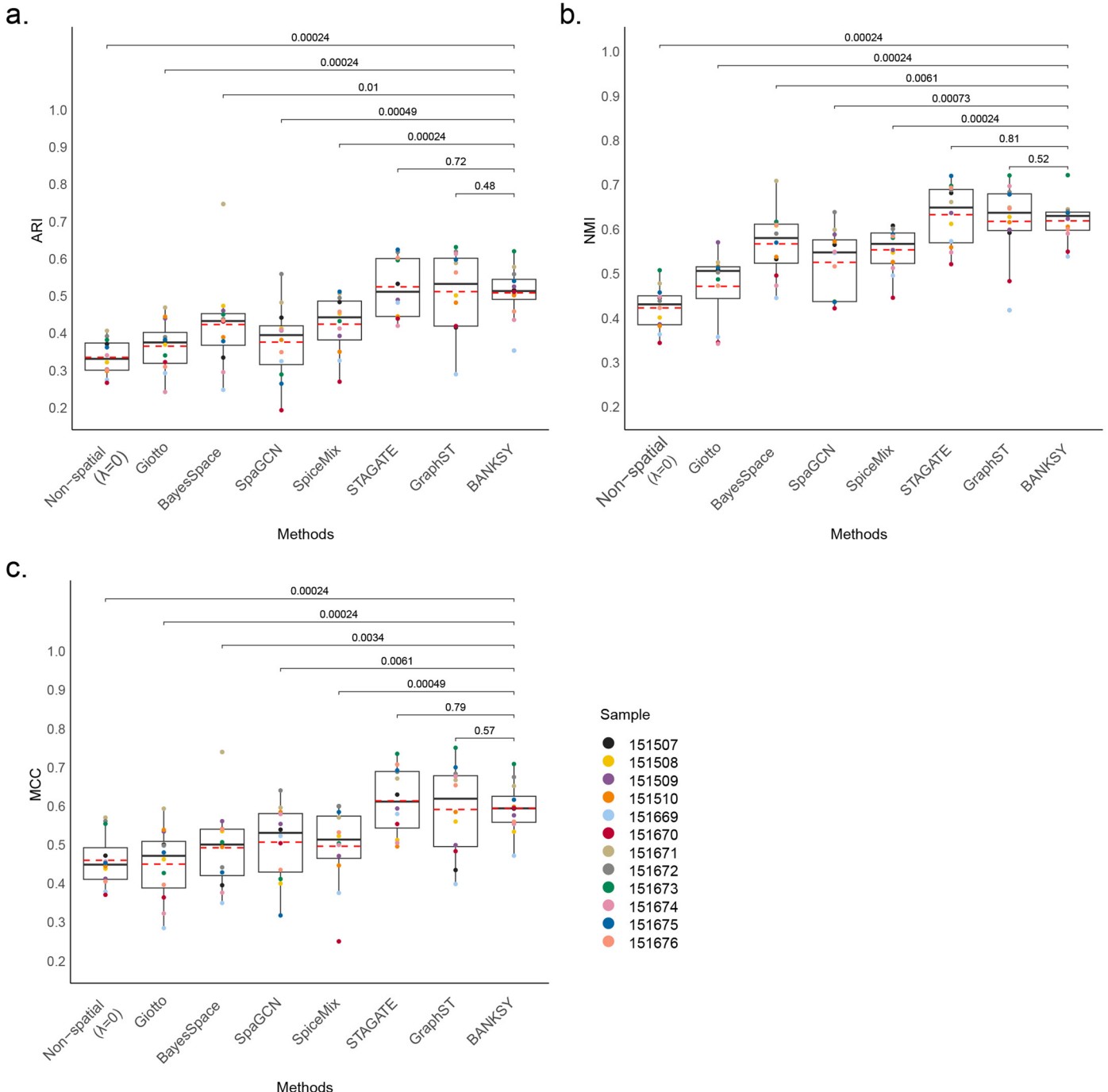

**Extended Data Fig. 6 | Clustering performance of each method on all 12 DLPFC datasets, quantified using 3 metrics.** (a) Adjusted Rand index (ARI); (b) normalized mutual information (NMI); and (c) Matthews correlation coefficient (MCC). Significance is computed using paired one-sided Wilcoxon signed rank test (*: p≤0.05, **: p≤0.01, ***: p≤0.001). Exact p-values from left to right for (a)

0.00024, 0.00024, 0.010, 0.00049, 0.00024, 0.72, 0.48, (b) 0.00024, 0.00024, 0.0061, 0.00073, 0.00024, 0.81, 0.52, (c) 0.00024, 0.00024, 0.0034, 0.0061, 0.00049, 0.79, 0.57. Center line: median, dotted line: mean, height of box: interquartile range (IQR), whiskers: 1.5 x IQR. In all boxes, n = 12 samples.

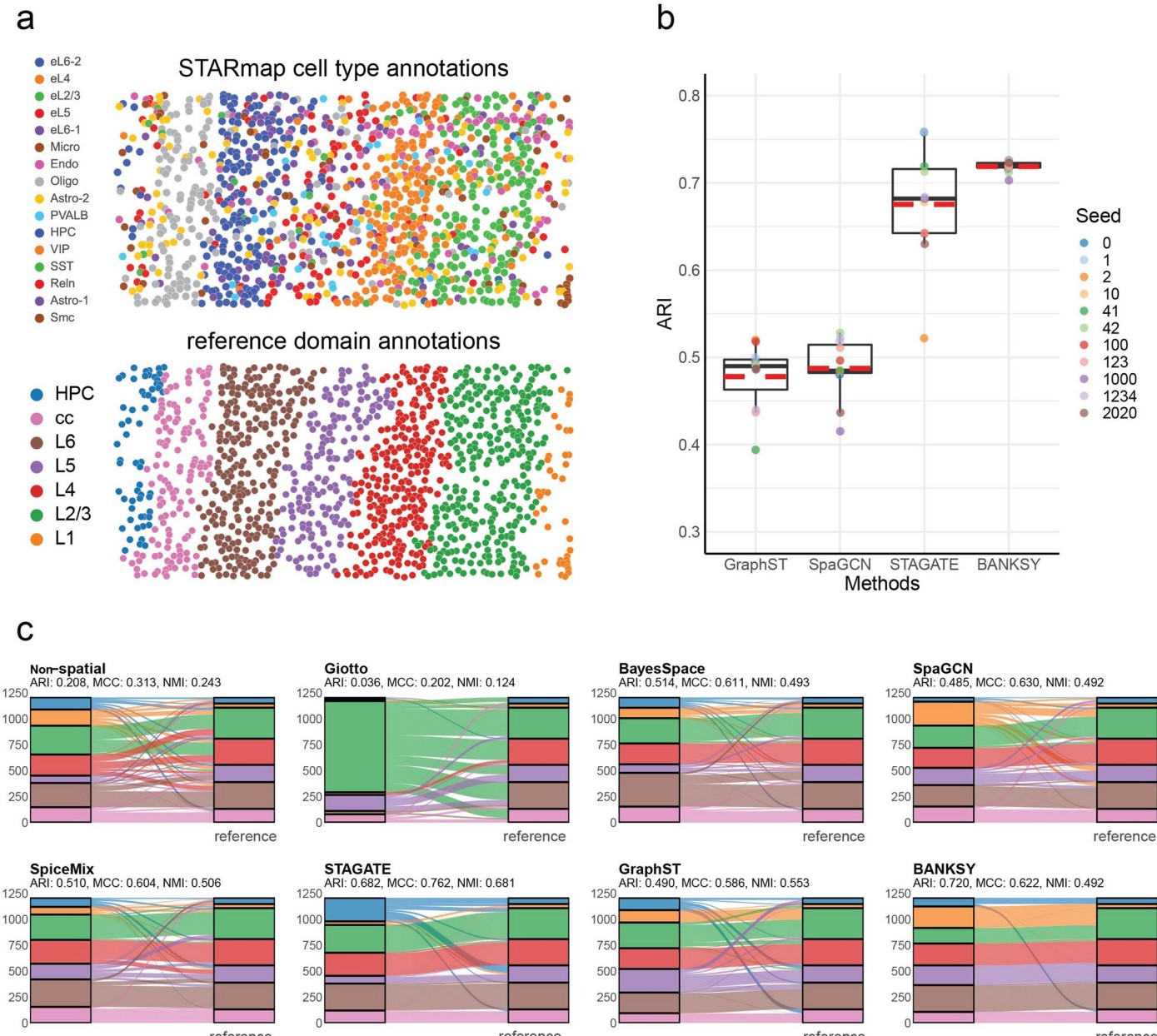

**Extended Data Fig. 7 | STARmap mouse visual cortex 1020 gene dataset.**
(a) Top: cell-type annotations; Bottom: implied domain annotations. (b) Domain segmentation comparison (boxplot) on the STARmap 1,020-gene mouse visual cortex dataset, showing performance variability across random seeds. GraphST, STAGATE, SpaGCN and BANKSY were run with n = 11 different random seeds and

ARI relative to the reference domain annotation was calculated in each case. (c) STARmap benchmark: alluvial plot showing correspondence between cluster labels assigned by spatial algorithms (left) and manual domain annotations (right). Correspondence is quantified using ARI, MCC and NMI.

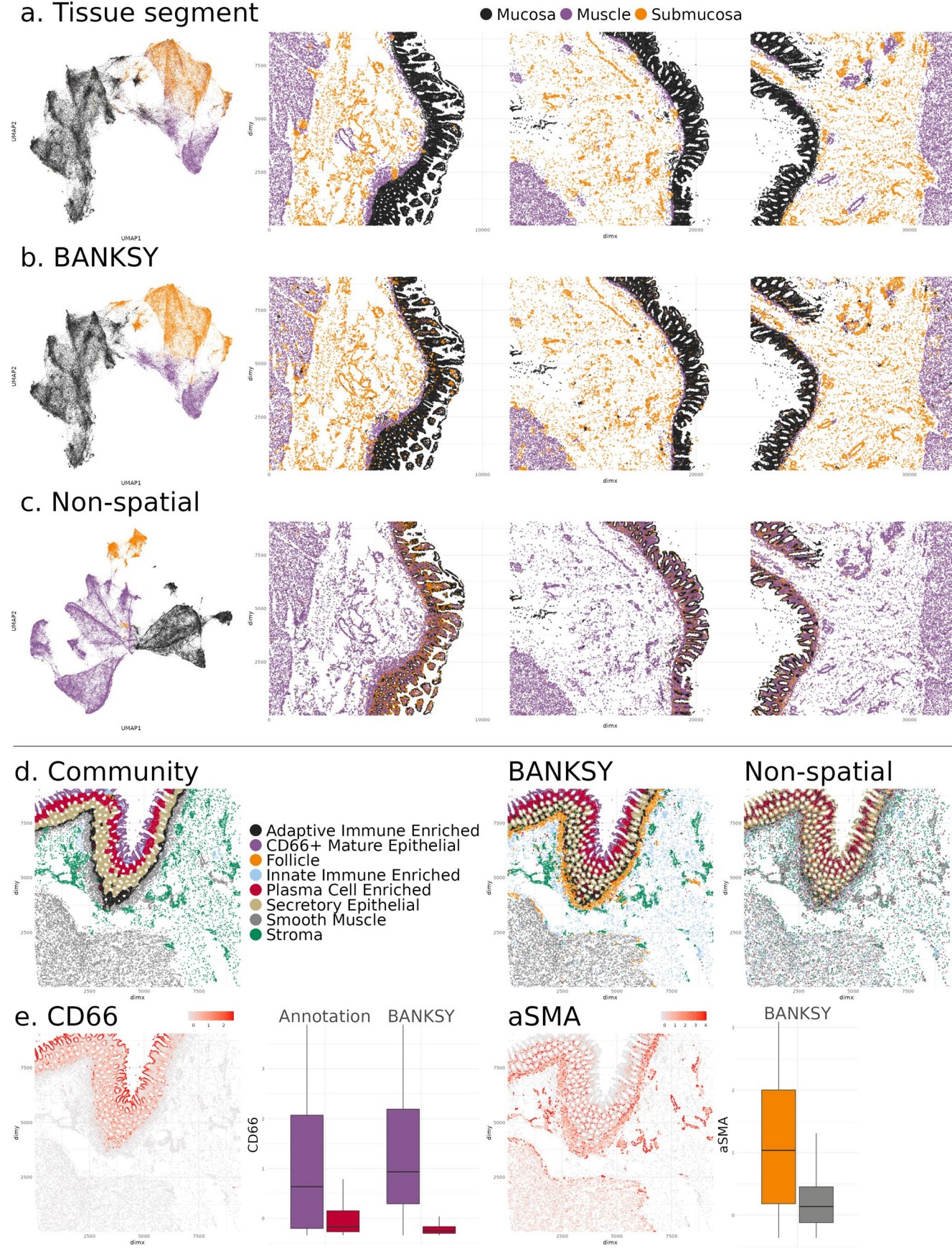

**Extended Data Fig. 8 | See next page for caption.**

**Extended Data Fig. 8 | BANKSY tissue segmentation and community identification on CODEX multiplexed imaging data of the intestine.**
a). BANKSY UMAP and spatial plots colored by tissue segment annotations.
b). BANKSY UMAP and spatial plots colored by BANKSY clusters. c). Non-spatial UMAP and spatial plots colored by non-spatial clusters. d). Comparison of BANKSY and non-spatial clusters with community domain annotations.
e). Marker genes for smooth muscle and mature epithelial communities identified by BANKSY. Boxplots are colored as in (d). Center line: median, dotted line: mean, height of box: interquartile range (IQR), whiskers: 1.5 x IQR. CD66 Annotation: n = 3179 (mean = 1.47, median = 0.636) vs 6189 (mean = 0.107, median = -0.167) cells, BANKSY: n = 3435 (mean = 1.70, median = 0.935) vs 5232 (mean = -0.184, median = -0.249), aSMA BANKSY: n = 3222 (mean = 1.26, median = 1.04) vs 4826 (mean = 0.245, median = 0.138). All p-values < 2.2e-16.

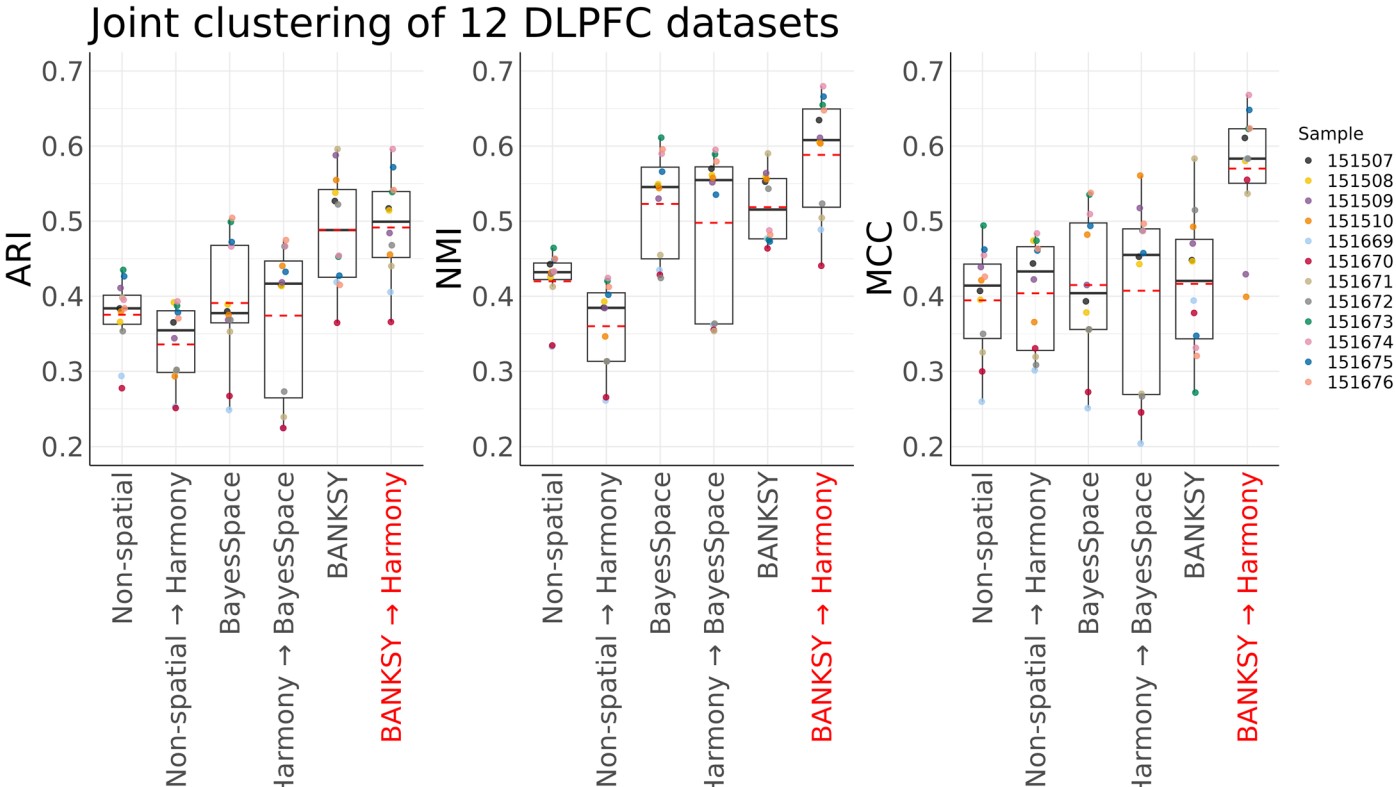

**Extended Data Fig. 9 | Benchmarking different integration methodologies.** Performance of non-spatial clustering, BayesSpace and BANKSY with and without batch correction on joint clustering of spots from n = 12 DLPFC Visium datasets.

ARI: adjusted Rand index. NMI: normalized mutual information. MCC: Matthew's correlation coefficient. Center line: median, red dotted line: mean, height of box: interquartile range (IQR), whiskers: 1.5 x IQR.

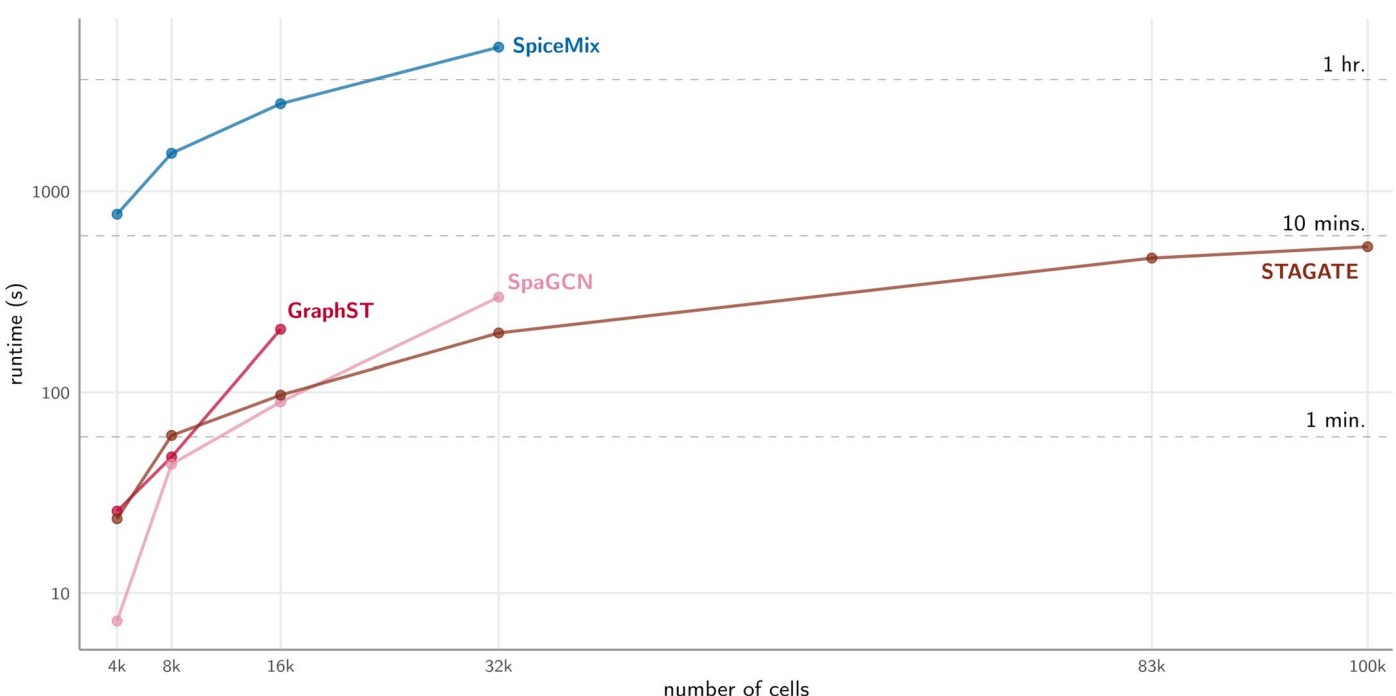

**Extended Data Fig. 10 | GPU runtimes of GraphST, SpaGCN, SpiceMix and STAGATE.** All methods were benchmarked using a NVIDIA T4 GPU with 16GB of memory. Runtimes are shown up to the maximum cell number accommodated by each method.

# Reporting Summary

## Statistics

For all statistical analyses, confirm that the following items are present in the figure legend, table legend, main text, or Methods section.

| n/a | Confirmed | |
|---|---|---|
| ☐ | ☒ | The exact sample size (*n*) for each experimental group/condition, given as a discrete number and unit of measurement |
| ☒ | ☐ | A statement on whether measurements were taken from distinct samples or whether the same sample was measured repeatedly |
| ☐ | ☒ | The statistical test(s) used AND whether they are one- or two-sided<br>*Only common tests should be described solely by name; describe more complex techniques in the Methods section.* |
| ☒ | ☐ | A description of all covariates tested |
| ☒ | ☐ | A description of any assumptions or corrections, such as tests of normality and adjustment for multiple comparisons |
| ☐ | ☒ | A full description of the statistical parameters including central tendency (e.g. means) or other basic estimates (e.g. regression coefficient) AND variation (e.g. standard deviation) or associated estimates of uncertainty (e.g. confidence intervals) |
| ☐ | ☒ | For null hypothesis testing, the test statistic (e.g. *F*, *t*, *r*) with confidence intervals, effect sizes, degrees of freedom and *P* value noted<br>*Give P values as exact values whenever suitable.* |
| ☐ | ☒ | For Bayesian analysis, information on the choice of priors and Markov chain Monte Carlo settings |
| ☒ | ☐ | For hierarchical and complex designs, identification of the appropriate level for tests and full reporting of outcomes |
| ☐ | ☒ | Estimates of effect sizes (e.g. Cohen's *d*, Pearson's *r*), indicating how they were calculated |

*Our web collection on statistics for biologists contains articles on many of the points above.*

## Software and code

Policy information about availability of computer code

| Data collection | The VeraFISH data was collected using the VSA-1 Imager from Veranome Biosystems, and the collected images were processed throught the VeraWorks software (version 1.0_006) to reconstruct the locations of the barcode probes. Segmentation was performed using a Mask-RCNN neural network, also part of the same software. |
|---|---|
| Data analysis | All BANKSY analysis was done with the Python (Python version 3.8, BANKSY version 1.1.0) or R (R version 4 (version >3.5), BANKSY version 0.1.5). For solving the linear sum assignment problem via the Hungarian algorithm, we used scipy (version 1.6.2) in the Python version of BANKSY, and the RcppHungarian (version 0.3) in the R version. The Seurat analysis of the VeraFISH and MERFISH data was done with Seurat version 4.1.1. Other software versions were as follows: BayesSpace version 1.5.1, Spicemix latest version (Git commit id: aea69f8), Leiden clustering from igraph version 1.2.11, scran version 1.18.7, SingleR version 1.4.1, Nebulosa version 1.0.2, SpaGCN version 1.2.2, Giotto version 1.1.0, BayesSpace version 1.5.1, STAGATE version 1.0.1, GraphST version 1.1.1, Harmony version 0.1.1, scDesign2 version 0.1.0, FICT version 1.0.0, MERINGUE version 1.0, peakRAM version 1.0.2 and CUDA version 12.0.<br><br>The R package, as well as scripts to reproduce analyses, can be obtained from https://github.com/prabhakarlab/Banksy, while the python version is available from https://github.com/prabhakarlab/Banksy_py. The IPython notebooks to reproduce our analysis on Slide-seq v1, Slide-seq v2 and STARmap are available on the Banksy-mansucript branch of the BANKSY_py Github repository. The scripts to reproduce the R analysis are available via https://github.com/jleechung/banksy-zenodo and via the Zenodo file repository: 10.5281/zenodo.1025879 |

For manuscripts utilizing custom algorithms or software that are central to the research but not yet described in published literature, software must be made available to editors and reviewers. We strongly encourage code deposition in a community repository (e.g. GitHub). See the Nature Portfolio guidelines for submitting code & software for further information.

## Data

Policy information about availability of data

All manuscripts must include a data availability statement. This statement should provide the following information, where applicable:
- Accession codes, unique identifiers, or web links for publicly available datasets
- A description of any restrictions on data availability
- For clinical datasets or third party data, please ensure that the statement adheres to our policy

The SlideSeq mouse cerebellum data were obtained from https://github.com/RubD/spatial-datasets/blob/ master/data/2019_slideseq_cerebellum/raw_data/ slideseq_cerebellum_urls.txt. We used the BeadLocationsForR and MappedDGEForR .csv files. For Slide-seq V2, we obtained the data from the Broad Institute Single Cell Portal at  https://singlecell.broadinstitute.org/single_cell/study/SCP948.The MERFISH mouse hypothalamus data were obtained from https:// doi.org/10.5061/dryad.8t8s248. The Vizgen MERSCOPE data for sample `Colon cancer 1' was obtained from https://info.vizgen.com/ffpe-showcase. The CosMX SMI data of the human healthy colon was obtained from NCBI GEO database (accession GSM7473683). The processed CODEX multiplexed imaging data of the healthy human intestine were downloaded from https://datadryad.org/stash/dataset/doi:10.5061/dryad.pk0p2ngrf. The mouse hippocampus data were collected using the VeraFISH assay (Veranome Biosystems, LLC, Mountain View, CA, USA) as described in Section 8.7. The data are available by running the command data(hippocampus) in the R version of the BANKSY package or directly from https://github.com/prabhakarlab/Banksy/blob/bioc/data/hippocampus.rda. The 10x Visium data of the dorsolateral prefrontal cortex (DLPFC) were obtained from the spatialLIBD project (http://spatial.libd.org/spatialLIBD) (Pardo B., et al., 2022). The STARmap mouse visual cortex data were obtained from http://clarityresourcecenter.org.

## Research involving human participants, their data, or biological material

Policy information about studies with human participants or human data. See also policy information about sex, gender (identity/presentation), and sexual orientation and race, ethnicity and racism.

| Reporting on sex and gender | No human data were collected as part of the study. |
|---|---|
| Reporting on race, ethnicity, or other socially relevant groupings | No human data were collected as part of the study. |
| Population characteristics | No human data were collected as part of the study. |
| Recruitment | No human data were collected as part of the study. |
| Ethics oversight | No human data were collected as part of the study. |

Note that full information on the approval of the study protocol must also be provided in the manuscript.

# Field-specific reporting

Please select the one below that is the best fit for your research. If you are not sure, read the appropriate sections before making your selection.

☒ Life sciences ☐ Behavioural & social sciences ☐ Ecological, evolutionary & environmental sciences

For a reference copy of the document with all sections, see nature.com/documents/nr-reporting-summary-flat.pdf

# Life sciences study design

All studies must disclose on these points even when the disclosure is negative.

| Sample size | In the Slide-seq and Slide-seq v2 datasets, we used all cells (25,551 and 39496 cells respectively). In the MERFISH mouse hypothalamus data (Fig. 3a-e), all 11 Naive animals were used for analysis (485,657 cells). In the scRNA-seq study of mature oligodendrocytes in the mouse hypothalamus (Fig. 3f), all cells labeled 'Mature Oligodendrocytes' by the authors of the original study were used after standard QC cutoffs (cells with >20% mitochondrial genes or <1000 NODG, see Methods), resulting in 6611 cells. In the VeraFISH mouse hippocampus dataset (Fig. 4a-r), we used all 10,994 cells in the dataset. For the corresponding scRNA-seq analysis, all cells with neuronal clusters corresponding to the relevant cortical and hippocampad CA3 regions (as labeled by the authors of the original study) were used, resulting in 2386 cells. Similarly, all cells labeled 'Oligo' were used for the oligodendrocyte analysis, resulting in 231 cells. In the MERSCOPE CRC data (Fig. 3f-h), we used all 677,451 cells in the dataset. In the DLPFC Visium dataset we used all 12 samples in the dataset, as is standard in the benchmarking performed in the field. For the STARMAP dataset, we used all 1207 cells annotated by the authors. In the colon CosMx data, there were two major connected (contiguous) regions of fields of view (FOVs), along with some FOVs with scattered cells. We used the larger of these connected regions for analysis, resulting in 32,765 cells. In the CODEX data, we used all 33,958 cells from the ileum, all 25,403 cells from the right color region, and all 27,784 cells from the transverse colon region for tissue domain annotation from Donor B0012 and 38371 cells from from donor B006 in the ascending colon region for community annotation. In the simulated data, we generated datasets modeled of real STARMAP data (see Methods Section 8.15), and created 3 samples simulated for each gene-set condition (400, 600, 800 and 1020 genes), comprising 4996 cells each. |
|---|---|
| Data exclusions | In the mouse hypothalamus MERFISH data, we followed the authors of the original study to remove cells marked 'Ambiguous' by them and the gene Fos, which contained 'NaN' entries. In the DLPFC Visium data, we removed spots marked 'ambiguous' by the authors. In the CosMx |

| Replication | The DLPFC data comprises 4 samples from each of 3 patients, resulting in 12 total datasets. following the usual practice in the field, we reported median statistics and boxplots over these 12 samples. For the simulated data, we generated 3 replicates for each gene count condition (400, 600, 800, all 1020) and reported median values of ARIs over these replicated for all tested methods. In all other datasets, the entire dataset was clustered and analyzed as a single dataset. |
| --- | --- |
| Randomization | In the simulated dataset, the genes to subset were picked randomly to generate 400, 600, 800 gene sets of the full 1020 gene set. No other randomization was used in this study (deterministic subsetting methods like highly variable genes were used for gene selection, and quality control metrics like number of detected genes were used for cell subsetting). |
| Blinding | Blinding was not applicable to this study because no sample group allocation was performed. |

data, there two major connected regions of FOVs, along with some FOVs with scattered cells. We used the larger of these connected regions for analysis, discarding the smaller region and the FOVs with scattered cells. In all other analyses, we used all cells, and applied standard QC cutoffs.

# Reporting for specific materials, systems and methods

We require information from authors about some types of materials, experimental systems and methods used in many studies. Here, indicate whether each material, system or method listed is relevant to your study. If you are not sure if a list item applies to your research, read the appropriate section before selecting a response.

## Materials & experimental systems

| n/a | Involved in the study |
| --- | --- |
| ☒ | Antibodies |
| ☒ | Eukaryotic cell lines |
| ☒ | Palaeontology and archaeology |
| ☐ | ☒ Animals and other organisms |
| ☒ | Clinical data |
| ☒ | Dual use research of concern |
| ☒ | Plants |

## Methods

| n/a | Involved in the study |
| --- | --- |
| ☒ | ChIP-seq |
| ☒ | Flow cytometry |
| ☒ | MRI-based neuroimaging |

## Animals and other research organisms

Policy information about studies involving animals; ARRIVE guidelines recommended for reporting animal research, and Sex and Gender in Research

| Laboratory animals | One six weeks old female mouse (C57BL/6NTac) was purchased from InVivos, Singapore, (https://www.invivos.com.sg/c57bl-6ntac/#tab-60391), and was euthanized and dissected for removal of the brain immediately upon receipt (the mice were not housed and housing conditions do not apply). |
| --- | --- |
| Wild animals | No wild animals were used in the study. |
| Reporting on sex | Sex differences were not the aim of this study. Mouse brain data was only used to compare different spatial clustering algorithms. |
| Field-collected samples | No field collected samples were used for this study. |
| Ethics oversight | All animal procedures were done in accordance with the approved Institutional Animal Care and Use Committee (IACUC) protocol (Protocol #211580) obtained from the IACUC of the biomedical resource center. |

Note that full information on the approval of the study protocol must also be provided in the manuscript.

## Plants

| Seed stocks | No plant data was collected or analyzed as part of this study. |
| --- | --- |
| Novel plant genotypes | No plant data was collected or analyzed as part of this study. |
| Authentication | No plant data was collected or analyzed as part of this study. |

