## [Peer Review File · Nature Genetics]

Peer Review Information

Manuscript Title: BANKSY unifies cell typing and tissue domain segmentation for scalable spatial omics data analysis

Corresponding author name(s): Dr Shyam Prabhakar, Dr Kok Hao Chen

Reviewer Comments & Decisions:

Decision Letter, initial version:
--

13th Apr 2023

Dear Dr Prabhakar,

Thank you for submitting your manuscript entitled "BANKSY: A Spatial Clustering Algorithm that Unifies Cell Typing and Tissue Domain Segmentation", for consideration. I regret that we are unable to publish it in Nature Genetics.

As you may know, we decline a substantial proportion of manuscripts without sending them to referees, so that they may be sent elsewhere without delay. Our editorial judgments are based on such considerations as the degree of advance provided, the breadth of potential interest to researchers and timeliness.

Here, we appreciated the potential broad audience for BANKSY, and the technical novelty of incorporating the AGF into the model of neighbourhood gene expression. However, we concluded that the overall performance gain over current methods remained unclear; for example, the ARIs of the various human DLPFC FOVs differ between the main and supplementary figures across methods, and furthermore the ARIs and segmentation for SpiceMix are inconsistent with that publication's benchmarking (see Fig. 6 of <https://www.nature.com/articles/s41588-022-01256-z>), while the calculated ARI for BANKSY in Fig. 5d here appears high given the orange/green segmentation appears clearly incorrect. If you are able to clearly demonstrate consistent performance improvement across a variety of benchmarks, as the SpiceMix publication presented, we would be interested in considering an appeal of this decision; please get in touch if you would like to discuss this further.

In this case, your manuscript has not matched our criteria for further consideration at Nature Genetics, and we think it would find a more suitable outlet in another journal. You might want to consider our sister journal *Nature Communications* as a potential venue for the publication of these results. *Nature Communications* publishes high quality and influential research and across the full spectrum of the natural sciences. More information on the journal, the potential benefits of transfer and a link to transfer your paper, can be found at the bottom of this email. Please

note that the editorial team at Nature Communications will consider your manuscript independently of our suggestion to transfer.

Please be assured that this editorial decision does not represent a criticism of the quality of your work, nor are we questioning its value to others working in this area. We hope that you will rapidly receive a more favorable response elsewhere.

With all best wishes,

Michael Fletcher, PhD
Senior Editor, Nature Genetics

Decision Letter, Appeal:

16th May 2023

Dear Shyam,

Thank you for your message of 16th May 2023, asking us to reconsider our decision on your manuscript "BANKSY: A Spatial Clustering Algorithm that Unifies Cell Typing and Tissue Domain Segmentation". I have now discussed the points of your letter with my colleagues, and we think that you have some valid points. We therefore invite you to revise your manuscript along the lines that you propose.

When preparing a revision, please ensure that it fully complies with our editorial requirements for format and style; details can be found in the Guide to Authors on our website (<http://www.nature.com/ng/>).

Please be sure that your manuscript is accompanied by a separate letter detailing the changes you have made and your response to the points raised. At this stage we will need you to upload:

1) a copy of the manuscript in MS Word .docx format.

2) The Editorial Policy Checklist:

<https://www.nature.com/documents/nr-editorial-policy-checklist.pdf>

3) The Reporting Summary:

(Here you can read about the role of the Reporting Summary in reproducible science:

<https://www.nature.com/news/announcement-towards-greater-reproducibility-for-life-sciences-research-in-nature-1.22062>)

Please use the link below to be taken directly to the site and view and revise your manuscript:

[redacted]

With kind wishes,

Michael Fletcher, PhD
Senior Editor, Nature Genetics
ORCID: 0000-0003-1589-7087

Decision Letter, first revision:

7th Jul 2023

Dear Shyam,

Your Article, "BANKSY: A Spatial Clustering Algorithm that Unifies Cell Typing and Tissue Domain Segmentation" has now been seen by 3 referees. You will see from their comments below that while they find your work of interest, some important points are raised. We are interested in the possibility of publishing your study in Nature Genetics, but would like to consider your response to these concerns in the form of a revised manuscript before we make a final decision on publication.

In brief, the 3 reviewers all sound positive and supportive of a potential publication, but vary in their enthusiasm and the amount of further work required before that final decision.

Reviewer #1 is the most skeptical, but they do sound open to the idea that BANKSY is indeed an improvement over current methods. They provide a number of specific suggestions that would help demonstrate such.

Referee #2 is more positive; their suggestions are also thoughtful and, we feel, would help to expand the potential user group by demonstrating the broad applicability of BANKSY to upcoming technological developments.

Reviewer #3 is the most strikingly positive of the lot. Their comments are focused on improving the usability of the R/python releases of BANKSY.

In our reading of these reviews, there is a clear path to an eventual publication, and none of the suggested additional work seems unduly difficult or excessive - so we hope you and your co-authors will be able to comprehensively respond to each request. We highlight that it will be especially important to persuade Referee #1 of BANKSY's improved performance, and would suggest focusing on this. We also appreciate Reviewer #3's suggestions to make the BANKSY codebase more stable/reliable by submitting it to bioC - we would certainly recommend doing so!

To guide the scope of the revisions, the editors discuss the referee reports in detail within the team, including with the chief editor, with a view to identifying key priorities that should be addressed in revision and sometimes overruling referee requests that are deemed beyond the scope of the current study. We hope that you will find the prioritized set of referee points to be useful when revising your study. Please do not hesitate to get in touch if you would like to discuss these issues further.

We therefore invite you to revise your manuscript taking into account all reviewer and editor comments. Please highlight all changes in the manuscript text file. At this stage we will need you to upload a copy of the manuscript in MS Word .docx or similar editable format.

*2) If you have not done so already please begin to revise your manuscript so that it conforms to our Article format instructions, available [here](http://www.nature.com/ng/authors/article_types/index.html). Refer also to any guidelines provided in this letter.

[redacted]

Sincerely,

Michael Fletcher, PhD
Senior Editor, Nature Genetics

ORCID: 0000-0003-1589-7087

Referee expertise: all 3 reviewers are in the field of spatial/single-cell genomics methods development.

Reviewers' Comments:

Reviewer #1:

Remarks to the Author:

Vipul Singhal, Nigel Chou, and colleagues resent their tool BANKSY, which aims to combine the cell type annotation and domain segmentation steps. They apply the tool to a number of healthy/diseased tissue datasets from various spatial transcriptomics methods and proceed to present some examples of improved cell-state annotation in niches and domain identification performance over existing tools. In particular, they also demonstrate that their algorithm is computationally efficient based on processing time and cellular throughput.

The authors present a very good piece of work, that is well laid out and easy to follow. The increased computational performance is also very good.

However, a key point about the novelty of existing similar methods is raised by the authors in the introduction. They point out that other tools also adopt this concept, so a demonstration of significantly increased performance compared to existing tools for cell typing and spatial clustering is required. On this point, the authors had also submitted a letter to the editor discussing quantitative performance, with particular attention paid to ARI scores and SpiceMix.

At this current stage, I believe are a number of limitations that should be addressed before further consideration.

Major points

- 1) One major result of the original MERFISH hypothalamus publication by Moffitt et al (<https://doi.org/10.1126/science.aau5324>) was that they "identified ~70 neuronal populations characterized by distinct neuromodulatory signatures and spatial organizations". The authors of Banksy do not provide any major comparison of the cell typing and spatial localization for these neuronal types, which were previously identified using non-spatial methods. Looking at the UMAP presented in Figure 3b suggests that the authors were not able to capture this. While comparing all of the types presented by Moffitt et al, the authors should aim to characterize 5-10 examples of spatially localized neuronal subtypes as presented in the supplementary material of Moffitt et al.
- 2) The original Moffitt study identified 6 Mature OD clusters in Moffitt et al Figure S22, using a non-spatial cell tying method. This is more than the Banksy results. The authors should compare their results to Moffitt et al, or identify other examples where Banksy performs better than non-spatial

methods for cell-typing including improved performance over the original analysis of the data.

3) In Suppl Fig 12 (comparing non-spatial and Banksy UMAPs in the MERFISH hypothalamus), visual evaluation of UMAPs to evaluate clustering performance is controversial. To really demonstrate the superiority of Banksy over non-spatial clustering, the authors should show that using non-spatial clustering that the Mature OD cluster cannot be split across a range of reasonable parameters that result in up to e.g. 30 clusters from the non-spatial algorithm.

4) Is the identification of bad-quality regions a feature or a bug? The authors demonstrate that data quality affects Banksy's clustering, identifying the necrotic core of brain organoids. However, in many imaging-based spatial profiling methods, the quality of each tile is as good as neighboring tiles. Could this be why Banksy incorrectly splits layer L2/3 in the STARmap data? The authors should investigate the tiling in the STARmap image, and evaluate whether the splitting of L2/3 is due to it.

5) For the colorectal cancer MERFISH data analysis I feel that comparing 22 Banksy clusters to 20 non-spatial clusters is an unfair comparison. The authors should instead demonstrate that increasing the number of non-spatial clusters to a higher number of clusters (e.g. 30) does not identify those clusters unique to Banksy.

6) In the domain segmentation comparisons, the authors compare to a range of widely known domain segmentation algorithms, however, they miss some of the newer algorithms based on graph-based modeling methods, e.g. STAGATE (<https://doi.org/10.1038/s41467-022-29439-6>), GraphST (<https://doi.org/10.1038/s41467-023-36796-3>), which also demonstrates very good performance (with self-reported ARI's higher than Banksy). I would suggest the authors add at least GraphST and STAGATE to their comparisons.

Minor

- I find the example presented in Suppl Fig 1 to be very crude. As far as I am aware, there are little to no commonly used tools for spatial clustering that are based on that simple model. I suggest both the figure and associated text be removed, at the author's discretion.
- Given the incorrect splitting of L2/3 (despite the higher ARI), the authors should clearly state that subjecting evaluation of L2/3 shows that Banksy performed worse than some of the other algorithms such as BayesSpace, Spice Mix etc despite the higher metrics, suggesting a need for use of newer metric to evaluate domain contiguity etc.
- The authors do not describe in detail how the algorithm would deal with rare cell types.
- SpiceMix is now published, so the bioRxiv reference should be updated
- Fig S19 could be arranged as a matrix with all rows showing comparable regions.
- Fig S29, 31,33 panels do not look aligned

Reviewer #2:

Remarks to the Author:

The authors developed Building Aggregates with a Neighborhood Kernel and Spatial Yardstick (BANKSY) that unifies clustering and domain segmentation analysis. BANKSY uses two spatial kernels to encode the transcriptomic signatures of a microenvironment around a cell, including a weighted mean expression (a Gaussian weighting envelope based on the radial physical distance between

neighbouring cells) and an azimuthal Gabor Filter (AGF, for analysing the spatial texture of neighbourhood gene expression). By concatenating the non-spatial gene expression matrix with the average neighbourhood expression matrix and an AGF matrix, BANKSY constructs a neighbour-augmented matrix. AGF, an effective method for extracting frequency and spatial location information in a polar coordinate system, is an innovative approach that the authors introduced to analyse spatial transcriptomics data. The speed and scalability gain derives from the fast and linear construction of the neighbour augmented matrix. BANKSY is faster and more scalable to millions of cells, as opposed to other tools, especially those dependent on probabilistic optimisation of parameters like BayesSpace, Giotto HMRF and SpiceMix or those models using neural network models like SpaGCN.

The manuscript is well written. The code is available in R and Python, and the documentation is detailed and helpful, although it can be polished further (minor comments below). The authors have performed comprehensive benchmarking and parameter sweep analyses and convincingly show that this approach improves the domain mapping task.

My main suggestions are below:

- BANKSY is capable of processing million-cell datasets, making it suitable for the latest versions of the new generation of spatial single-cell resolution technologies. It would be useful to provide examples of how BANKSY works with common data in the field, like Xenium and CosMX. Would BANKSY work for high plex protein data like Akoya FusionCycler data?
- The spatial information used in BANKSY is for cells in one tissue section. How can BANKSY be applied to integrate data from multiple tissue sections or multiple regions of interest (ROIs) from the same sample?
- In the section 4.1 BANKSY algorithm, the authors describe clearly how the Neighbor augmented matrix can be computed. However, it is recommended to add more information and discussion on how this matrix is used for downstream analysis. For example, PCA is applied to this matrix, followed by Leiden community detection. Is this matrix used for other types of analyses and whether multiple matrices are generated for cell type identification and domain mapping?
- Using the augmented matrix described above, the final results highly depend on lambda that determines the relative weights/importance of the original gene-by-cell matrix vs. the weighted mean matrix vs. the AGF matrix (equation 2, justified in Supp Note 2). The authors have assessed the effect of varying lambda values from 0 to 0.3 (Supp Fig 31). With a lambda at 0.3, the weights for matrix C, M and G are 0.83, 0.45, and 0.32, respectively, giving more weight to the original gene expression matrix. It would be helpful also to see the effect of lambda values ≥ 0.8 where BANKSY prioritises more spatial domain information. For example, with a lambda at 0.8, the corresponding weights for C, M, and G are 0.44, 0.73 and 0.52.
- Could the authors add more justification on why matrix G is assigned a lower weight than matrix M?
- It is well demonstrated that the BANKSY algorithm improves tissue domain mapping, but could the authors explain how BANKSY improves cell type identification?

Minor comments:

- Fig 1b not cited in the main text

- In Fig 3h, are the number of clusters from non-spatial and BANKSY the same for comparing the cell type annotation?
- For deep learning based methods like SpaGCN, could be authors make use of GPUs as this is often needed for these methods.
- For the Python version, would be helpful for installation to make it available through Anaconda or PyPI
- The R installation went very smoothly (Macs users might need to install the additional bitops package)
- The default R tutorials are well written and the workflow is understandable, but some commands wouldn't work as they are in the current tutorial: <https://prabhakarlab.github.io/Banksy/> (for example the SubsetBanksy as an error: nCount not found; a very small error that can be fixed easily)

Reviewer #3:

Remarks to the Author:

The authors of this manuscript present a novel and very interesting method for clustering called BANKSY. It has a key parameter, lambda, that controls whether to cluster cells (or spots for spot-based spatially-resolved transcriptomics methods such as Visium) with information borrowed from neighbor spatial cells or not. The degree to which you borrow information from neighbor cells will influence the resulting spatial shape of the resulting clusters. The authors showed with basically as many mainstream technologies you can think of how their method compares to other competitor methods, and if there was some orthogonal measurement ("ground truth"), then they evaluated their performance against it. BANKSY looks very promising based on these comparisons as well as in terms of scalability. It is available as both an R package as well as a Python software, though I question further below whether this will be the case moving forward. The R package is very well documented and has multiple vignettes at <https://prabhakarlab.github.io/Banksy> showing how to use the software with multiple data types. The R package has been tested in multiple operating systems and in theory users can install it (I did so with R 4.3.1 – aka the latest release version – without any issues). Some of my major comments below are mostly focused on the software rather than the manuscript, since I think that the software can be improved a little bit to have a much larger impact on users. For example, by increasing the interoperability of Banksy with other common data containers. More happy users will lead to more citations ;). In other words, I do think that this extra work will be worth doing as I think that many people will want to use BANKSY to analyze their spatially-resolved data. I think the manuscript is very well written (with some missing method citations, which can be fixed) and the major points are well sustained. I do think that readers will benefit from improving the explanation behind Figure 1b as this novel feature role is easily overlooked. As a future BANKSY user, I'm excited to read this work and hope that your software will be maintained for years to come. Though I understand that maintaining open source software is challenging in this funding climate.

Major comments

* To me, Supplementary Figure 1 is too important to be included in the supplement. I recommend adding it after Figure 1C, maybe in a vertical format like | Sup Fig 1A | Sup Fig 1B | etc, or whichever way you think it fits best. I am aware that given a large influence to spatial coordinates does result in something similar to what we see in Supplementary Figure 1D. BANKSY's contributions lie in the land of $\lambda > 0$ and < 1 , which is why I anticipate that many people will use it. Yes, the rest of the main figures are important to convince users across different technologies and applications of the validity of

the results as well as the performance of BANKSY. But this figure is the one that exemplifies the best the extremes in the values of lambda as well as what happens in the interesting part of $\lambda > 0$ and < 1 .

* Deposit your code on Zenodo or some more permanent location than GitHub. Otherwise, you could decide to delete your GitHub repository at any moment. See

<https://docs.github.com/en/repositories/archiving-a-github-repository/referencing-and-citing-content> for more details. Then on lines 1529 and 1530 you can cite the DOI for your two GitHub repositories.

* I think that authors should include in their discussion the fact that BANKSY was run one sample at a time (like in the case of the Maynard et al, 2021 Visium data). This can be a major limitation if the resulting clusters across samples are not guaranteed to be interpreted as the same unit. That is, cluster 1 from sample 1 might not mean the same thing as cluster 1 from sample 2. See

https://edward130603.github.io/BayesSpace/articles/joint_clustering.html for an example of a competitor method that does perform joint clustering. I would also highly recommend discussing how batch effects impact BANKSY when analyzing multiple samples. See also <https://doi.org/10.1101/2023.02.15.528722> supplementary Figure 9 for the effect of batch correction on joint clustering with BayesSpace.

* Where is the code for reproducing your analyses? Please include it on section 5.4 Code availability. Also deposit it on Zenodo or some similar host (Figshare, etc).

* I think that you will have many people interested in using your software. As an R user, I mostly looked at <https://github.com/prabhakarlab/Banksy>. I think that it would be best to submit your R package to Bioconductor <http://bioconductor.org/> with the Software and Spatial biocView terms http://bioconductor.org/packages/release/BiocViews.html#___Spatial. This will help you and your users by making sure that your software is tested on Windows, Linux, and macOS daily. Yes, I see that you did test it on these OSES at GitHub Actions

<https://github.com/prabhakarlab/Banksy/blob/6d76280cad5169be77af8c4b055bcf7a2399c3d3/.github/workflows/check-standard.yml#L21-L23>, but that's only at the time of pushing a commit. A cron job on GitHub Actions could help, but I do think that submitting your software to Bioconductor would be the best.

* Related to your R package, I'm not sure why you decided to make new container classes instead of re-using classes like SpatialExperiment or SingleCellExperiment. Different classes mean that users will have to re-cast their data and as package authors, you'll have to keep a close eye to make sure users can easily transform their data from those other common containers into your objects. That is, you'll need to keep your `asBanksyObject()` function that I see at

<https://prabhakarlab.github.io/Banksy/articles/single-cell-exp.html#running-banksy> updated. Overall, if you submit your R package to Bioconductor, the reviewers from Bioconductor might have more specific suggested changes with users in mind.

* In contrast to the R version, the Python version has very sparse documentation at https://github.com/prabhakarlab/Banksy_py. It also only has 3 commits versus 203 on the R version. Do you intend to maintain both versions? If you keep only the R version, Python users could use

<https://bioconductor.org/packages/zellkonverter/> to transform the AnnData objects into SpatialExperiment ones, then run Banksy in R, and export the clustering results into a format that they can read in Python. Also, if you are not aware, basilisk at

<https://www.bioconductor.org/packages/basilisk/> can be used to incorporate Python dependencies on R packages. For example, if there's some computation that for some reason you prefer to run through Python rather than R. So this could be a way to merge your two projects into one.

Minor comments

- * Page 3, Lines 110 to 116 would benefit from specific citations.
- * Page 3, Lines 117 to 119. It would be best to cite the examples on mammalian brain you are referring to here.
- * Page 3 lines 131 to 133. <https://doi.org/10.1101/2023.02.15.528722> did use BayesSpace on 113,927 Visium spots and <https://doi.org/10.1038/s41586-022-05060-x> has 91,517 spots (though they didn't use a spatial clustering method). Of course, it's not millions, but it is larger than some of those initial Visium studies. Also, it feels weird to highlight millions on spots as a limitation of BayesSpace since you ran BANKSY one sample at a time, and did not perform joint clustering like it can be done with BayesSpace https://edward130603.github.io/BayesSpace/articles/joint_clustering.html.
- * Page 4 lines 158 to 160. Please cite Seurat, SingleCellExperiment, Scanpy and other methods.
- * Figure 1 is quoted on the main text overall, without quoting Fig 1a, Fig 1b, etc. Given that, the caption for Figure 1b is very sparse and doesn't really explain the "real component" versus the "imaginary component" shown, nor the -1 to 1 values shown. Parts a and c are much more explained both in the figures themselves as well as the caption.
- * Figure 2a: by "non-spatial", do you mean BANKSY with $\lambda = 0$? Or some other non-spatial clustering method? Ideally, it would be $\lambda = 0$ to make sure that we are not comparing Leiden vs some other type of non-spatial clustering method (say Louvain). If you are indeed comparing $\lambda = 0$ vs some other λ value, I recommend specifying this on the caption of figure 2a.
- * Page 22 line 1007: corresopnding -> corresponding
- * Page 23, lines 1022 to 1025. Please formally cite the methods you mentioned. For example, with R you can check how to cite a package by using `citation("RcppHungarian")`. A lot of work is put into software development (just like you have done with Banksy) and citations is how such work is formally credited. Related to this, `citation("Banksy")` doesn't link to the pre-print version of Banksy <https://www.biorxiv.org/content/10.1101/2022.04.14.488259v1>. This can be done by adding an `inst/CITATION` file at <https://github.com/prabhakarlab/Banksy>.
- * Page 23 line 1057. Related to one of my earlier points, I think that it would be best if you changed phrases like "We then performed BANKSY and non-spatial clustering" to something like "We then performed BANKSY ($\lambda = 0.2$) and non-spatial clustering ($\lambda = 0$)" if indeed you used BANKSY with $\lambda = 0$ for the non-spatial clustering. In any case, being explicit about the λ values is best, despite your common λ values being stated in lines 1034 to 1036. I see that you did explicitly the λ values sometimes, like in line 1239. I encourage you to do this all the time.
- * I'm not familiar with VeraFISH data, but on page 24 line 1093 it caught my attention that you are normalizing cell counts to a single common value (100 in this case). Is this commonly done when analyzing this type of data? If so, you could cite some external source where these processing steps have been used.
- * Page 24 line 1102. Also line 1205. Please cite scran (it is spelled with a lower case s). Check the output of `citation("scran")` in R.
- * Line 1111. Cite Seurat.
- * Line 1120: why are you now using 2% instead of 1% as in line 1091? You also used 2% at line 1218.
- * Sometimes you spell out the word "version" like in line 1111 and sometimes you use "v" like in line 1142. I suggest that you use one option to be consistent.
- * Lines 1143 to 1148 (and others) are confusingly written. You make it sound like you are setting a lot of non-default parameters. I strongly recommend being explicit to highlight which are the default values and which are not. For example, the 50,000 MCMC iterations and the burn-in period of 1,000 iterations are the defaults at

<https://github.com/edward130603/BayesSpace/blob/8e9af8f2fa8e93518cf9ecee1ded9ab93e88fffd/R/spatialCluster.R#L123> (gamma is set to NULL but defaults to 3 when platform is equal to "Visium" from the documentation at <https://edward130603.github.io/BayesSpace/reference/spatialCluster.html> and <https://github.com/edward130603/BayesSpace/blob/8e9af8f2fa8e93518cf9ecee1ded9ab93e88fffd/R/spatialCluster.R#L162-L164>) whereas k (or N as you used in your manuscript) = 16 to 20 is not the default. As a reader, I have a hard time distinguishing default parameters vs non-default parameters that you used for other software that I'm not as familiar with. Later on, like in lines 1221 to 1222 you are clearer by saying which function you used and which non-default parameters you used (if any).

* Lines 1157 to 1158, cite the `smfishHmrf` package.

* Line 1163. The link is not hyperlinked. Also, hm... I don't think that FICT has a formal citation from what I see at <https://github.com/haotianteng/FICT-SAMPLE>. But please check. You could ask the authors at <https://github.com/haotianteng/FICT-SAMPLE/issues> about how to cite them formally. They could simply deposit their code to Zenodo <https://docs.github.com/en/repositories/archiving-a-github-repository/referencing-and-citing-content>, which would generate a DOI that you could then formally cite.

* Lines 1169 to 1171, cite `SpiceMix` and `dbscan`. Yes, I know that you later cite `SpiceMix` in line 1175, but since you mentioned it at line 1169, it's best to cite it there too.

* Line 1175: I really like how you mentioned the specific Git commit ID!

* Line 1178: cite Leiden. See the output of `?igraph::cluster_leiden` which points to "Traag, V. A., Waltman, L., & van Eck, N. J. (2019). From Louvain to Leiden: guaranteeing well-connected communities. *Scientific reports*, 9(1), 5233. doi: 10.1038/s41598-019-41695-z, arXiv:1810.08473v3 [cs.SI]"

* Line 1179: cite `igraph`.

* Line 1210 to 1211 doesn't make sense to me. Maybe there's a typo? Like a gene is either from the mitochondria or it isn't. Did you mean "cells" instead of "genes"?

* Line 1230. If there is no explicit version number, specify the date you downloaded the data, in case the name is kept the same ("colon cancer 1"), but the underlying data changes at some future point.

* Lines 1267. Cite `spatialLIBD`; check citation("spatialLIBD"). Same in line 1522.

* Line 1272 "normalizing the data to the median library size for each sample" could be a little clearer. I think here you meant to say that for each sample, you normalized the counts per spot to the median library size across all spots of the sample.

* Related to an earlier point, lines 1315 and 1316 say "Default values were used for all other parameters."

* Line 1409, change "We" to "we".

* Cite `scDesign2`.

* Supplementary Figure 28. Can you highlight with some box color annotation the values you used for the main figures? Do a similar thing for Supplementary Figure 29 and other related figures. It can be hard to remember all the parameters mentioned in the main text when looking at these supplementary figures.

* Fix the citation in supplementary page 45. It says citation 9, but it's 7 on the main paper. Actually, I see that you duplicated references in the supplementary material. Hm... I'd check with the editors if that's ok if they prefer to have unique main text references vs supplementary text references (so no overlap between them).

* Some of your supplementary citations are missing information. I recommend including the URL for some of them like reference 1, 4, 6, and 7.

* * I'm aware that `spatialLIBD::fetch_data()` can be used to download the Maynard et al 2021 data. So at <https://prabhakarlab.github.io/Banksy/articles/dlpc-analysis.html> I was a bit disappointed to not see how you transformed the `SpatialExperiment` objects from that study into a `BanksyObject`. This

information is not present at <https://prabhakarlab.github.io/Banksy/reference/dlpfc151673.html> or similar files either.

Author Rebuttal, first revision:

Reviewer #1:

Remarks to the Author:

Vipul Singhal, Nigel Chou, and colleagues present their tool BANKSY, which aims to combine the cell type annotation and domain segmentation steps. They apply the tool to a number of healthy/diseased tissue datasets from various spatial transcriptomics methods and proceed to present some examples of improved cell-state annotation in niches and domain identification performance over existing tools. In particular, they also demonstrate that their algorithm is computationally efficient based on processing time and cellular throughput.

The authors present a very good piece of work, that is well laid out and easy to follow. The increased computational performance is also very good.

However, a key point about the novelty of existing similar methods is raised by the authors in the introduction. They point out that other tools also adopt this concept, so a demonstration of significantly increased performance compared to existing tools for cell typing and spatial clustering is required. On this point, the authors had also submitted a letter to the editor discussing quantitative performance, with particular attention paid to ARI scores and SpiceMix.

At this current stage, I believe are a number of limitations that should be addressed before further consideration.

We thank the reviewer for their valuable feedback, and for their positive remarks on the clarity of presentation and BANKSY's improved computational performance.

We appreciate the need to demonstrate significant performance improvement compared to existing tools for cell-typing and domain segmentation. BANKSY's performance has now been benchmarked against 8 other algorithms in total (see point-by-point responses below for details). At both cell-typing and domain segmentation, BANKSY is the highest overall performer. Moreover, BANKSY is unique in its ability to solve both algorithmic problems - the other 8 methods are specialized for one of the two tasks. Another major advantage of BANKSY is its stability across biological replicates and random seed settings - the other methods showed 3x greater variability in performance. Lastly, BANKSY is at least 8x more scalable than existing methods. It is also at least 10x faster on conventional CPUs, thus obviating the need for GPU computing.

In addition to the above, BANKSY's feature augmentation strategy is, to the best of our knowledge, both novel and required for the above-described generality and performance advantages. We now demonstrate one more conceptual advance, which involves the use of BANKSY's spatial embedding for data integration. We quantitatively show that BANKSY outperforms both non-spatial and BayesSpace data integration on the 12 DLPFC datasets. This result suggests that there is scope for developing new spatial data integration algorithms that use BANKSY's spatially-informed embedding instead of conventional non-spatial embeddings. More generally, we believe that BANKSY's embedding could support the development of multiple new algorithms for spatial omics data, including methods that infer cell-cell signaling or differentiation trajectories. For these reasons, we believe that BANKSY will be a transformative tool for spatial omics data analysis that will be widely used to generate novel biological insights.

Major points

1) One major result of the original MERFISH hypothalamus publication by Moffitt et al (<https://doi.org/10.1126/science.aau5324>) was that they "identified ~70 neuronal populations characterized by distinct neuromodulatory signatures and spatial organizations". The authors of Banksy do not provide any major comparison of the cell typing and spatial localization for these neuronal types, which were previously identified using non-spatial methods. Looking at the

UMAP presented in Figure 3b suggests that the authors were not able to capture this. While comparing all of the types presented by Moffitt et al, the authors should aim to characterize 5-10 examples of spatially localized neuronal subtypes as presented in the supplementary material of Moffitt et al.

We would like to clarify that Moffitt et al. did not identify the 70 neuronal subtypes in their initial clustering analysis of their dataset. In that paper, the authors performed iterative clustering, which is a manual procedure in which one first clusters all cells to identify the major cell-types, and then extracts each major cell-type for further sub-clustering at higher resolution. Since we ran BANKSY and non-spatial clustering on the entire Moffitt et al. dataset (all cells), we compared our results to the equivalent clustering result in the original paper, i.e. their first-round result that identified only the major cell-types.

In response to this comment, we have now followed the two-step iterative clustering approach from Moffitt et al. by separately sub-clustering excitatory and inhibitory neurons. In the second iteration, BANKSY did indeed identify 70 neuronal clusters (**Supp. Fig. 15b**). In particular, BANKSY fully recapitulated all of the spatially patterned neuronal subtypes illustrated in Moffitt et al.'s **Fig. 5a** (**Reviewer Fig. 1**, corresponding to **Supp. Fig. 15a**).

Reviewer Fig. 1 (corresponding to **Supp. Fig. 15a**): Top two rows: Fig 5a in the Moffitt. et al. paper, which shows neuronal sub-clusters localized to specific hypothalamic nuclei. Bottom two rows: corresponding BANKSY subclusters show close correspondence to Moffitt et al.'s neuronal subclusters.

One more general point to note is that the number of clusters is not necessarily an indicator of algorithm accuracy. Indeed, the most widely used algorithms for clustering scRNA-seq data use graph-based approaches (Louvain, Leiden) that can, in principle, identify any number of clusters simply by tuning the resolution parameter. This resolution parameter is typically tuned manually by the user. For example, if Moffitt et al. had used a higher resolution setting, they could have identified even more than 70 neuronal subclusters.

2) The original Moffitt study identified 6 Mature OD clusters in Moffitt et al Figure S22, using a non-

spatial cell typing method. This is more than the Banksy results. The authors should compare their results to Moffitt et al, or identify other examples where Banksy performs better than non-spatial methods for cell-typing including improved performance over the original analysis of the data.

We would like to clarify that the 6 Mature OD clusters in Moffitt et al.'s **Fig. S22** were identified by clustering scRNA-Seq data. In their MERFISH data, Moffitt et al. identified only 4 Mature OD clusters, of which one had very few cells (Mature OD 3, ~0.19% of all cells). We were unable to detect this rare cluster when we attempted to reproduce their (non-spatial) clustering analysis, perhaps due to differences in library versions. We therefore focused on the remaining 3 Mature OD clusters from their paper.

When we increased the Leiden clustering resolution parameter in BANKSY, we successfully identified these 3 clusters (**Supp. Fig. 13i** and **13o**). In addition, BANKSY was able to identify spatially patterned subpopulations of Mature OD 1 and 2. In both cases, one of the two subpopulations uniquely identified by BANKSY was localized to white matter in the anterior commissure, while the other was localized to the remaining grey matter (circled subpopulations in **Supp. Fig. 13o**). This split is also illustrated in **Fig. 3b-e** of our manuscript, and it is characterized by differential expression of multiple genes. Note that, even at high clustering resolution, non-spatial clustering was unable to identify these spatially segregated Mature OD subpopulations identified by BANKSY.

Supp. Fig. 13: Mature oligodendrocytes shown on both non-spatial and BANKSY UMAP coordinates. (a-d) Non-spatial UMAP coordinates, all cells. (a) Moffitt et al.'s labels. (b-d) Our non-spatial clustering ($\lambda = 0$) labels at increasing clustering resolutions. (e-h) BANKSY embedding UMAP coordinates, all cells. (e) Moffitt et al.'s labels. (f-h) BANKSY ($\lambda = 0.2$) labels at increasing clustering resolutions. (i-l): Boxed region in panel a, with Moffitt et al.'s labels (panel i), and our non-spatial labels at increasing clustering resolutions. (m-p) Boxed region in panel e, with Moffitt et al.'s labels (panel m), and

BANKSY's labels at increasing clustering resolutions. In particular, panels m and n show that the Moffitt et al. non-spatial labeling is along a complementary axes relative to the BANKSY split, and panel o shows that at higher resolutions, BANKSY is able to split the cells along *both* axes. (q-s) Non-spatial labels on the BANKSY embedding UMAPs show that non-spatial clustering at higher resolutions cannot capture the BANKSY split.

3) In Suppl Fig 12 (comparing non-spatial and Banksy UMAPs in the MERFISH hypothalamus), visual evaluation of UMAPs to evaluate clustering performance is controversial. To really demonstrate the superiority of Banksy over non-spatial clustering, the authors should show that using non-spatial clustering that the Mature OD cluster cannot be split across a range of reasonable parameters that result in up to e.g. 30 clusters from the non-spatial algorithm.

As suggested, we have now increased the resolution of non-spatial clustering to identify as many as 81 clusters ($res=5$). Even at this high resolution, non-spatial clustering failed to identify the spatially patterned Mature OD subtypes identified by BANKSY (see **Supp. Fig. 13d** above).

4) Is the identification of bad-quality regions a feature or a bug? The authors demonstrate that data quality affects Banksy's clustering, identifying the necrotic core of brain organoids. However, in many imaging-based spatial profiling methods, the quality of each tile is as good as neighboring tiles. Could this be why Banksy incorrectly splits layer L2/3 in the STARmap data? The authors should investigate the tiling in the STARmap image, and evaluate whether the splitting of L2/3 is due to it.

We have now performed additional analyses to investigate the possibility that BANKSY split L2/3 in the STARmap data because of differential data quality between tiles in the upper and lower halves of the imaged region. We note in passing that GraphST (Long et al., 2023) also generated the same split - this is not unique to BANKSY.

First, we examined the spatial distribution of the number of detected genes (NODG) in each cell, a common QC metric for data quality. We did not observe any clear tiling pattern in NODG (**Reviewer Fig. 2**). Rather, shifts in NODG appeared to map to cortical layer boundaries (e.g. lower NODG in L6). Thus, visual inspection did not indicate any correspondence between data quality and the up-down split in L2/3.

Next, we statistically compared NODG between the upper and lower clusters identified by BANKSY within L2/3 using the Wilcoxon rank-sum test. We performed this comparison on each of the 7 cell-types with at least 10 cells represented. None of the cell-types showed significant differentiation in data quality (FDR q -value > 0.1 ; Benjamini-Hochberg adjustment for multiple testing; **Reviewer Table 1**).

To further explore the possibility of differential data quality, we extended the above analysis to all layers. None of the 16 cell-types showed a significant difference in data quality between the upper and lower halves of the imaged region (FDR q -value > 0.2 ; Benjamini-Hochberg adjustment for multiple testing; **Reviewer Table 1**). Thus, we did not find evidence that data quality differences drove the split in Layer 2/3.

Reviewer Fig. 2: Spatial plot of number of detected genes (NODG) across the STARmap mouse visual cortex sample.

Cell-Type	Upper and lower clusters in L2/3, FDR Q-values (Benjamini-Hochberg)	Upper half vs lower half of all layers, FDR Q-values (Benjamini-Hochberg)
Astro-1	NA	0.399
Astro-2	0.709	0.399
Endo	0.877	0.724
HPC	NA	0.724
Micro	0.191	0.204
Oligo	NA	0.399
PVALB	NA	0.724
Reln	NA	0.958
SST	NA	0.638
Smc	NA	0.399
VIP	0.534	0.399
eL2/3	0.127	0.399
eL4	0.534	0.821
eL5	NA	0.680
eL6-1	0.541	0.638
eL6-2	NA	0.399

Reviewer Table 1: FDR q-values for each cell-type with at least 10 cells, between the upper and lower halves of the dataset, or between upper and lower clusters within L2/3 identified by BANKSY.

Having failed to detect significant evidence of data quality as a driver, we asked if the split in Layer 2/3 could be attributed to specific differentially expressed genes. Indeed, we were able to identify 4 genes with clear upregulation in the lower subcluster and 2 genes with clear upregulation in the upper subcluster (**Reviewer Fig. 4**; FDR q-value (BH) <0.04), which may explain the splitting of this tissue domain into two spatially distinct clusters by BANKSY and GraphST.

Reviewer Fig. 4: Violin plots showing genes that were significantly differentially expressed (q-value in parentheses) between the two clusters identified by BANKSY within the L2/3 region.

Nevertheless, for quantitative benchmarking purposes, we had treated the split in Layer 2/3 as an erroneous prediction by BANKSY. Despite this error, BANKSY remains the highest performer in terms of ARI on this dataset. Importantly, BANKSY segments all other domains accurately, labels spatially contiguous domains with contiguous labels (Fig. 5d), and is also

the best performer by 2 alternative metrics of label concordance: MCC and NMI. Below, we provide (**Supp. Fig. 30b**) as an alternate visualization of algorithm accuracy.

Supp. Fig. 30b. STARmap benchmark: alluvial plot showing correspondence between cluster labels assigned by spatial algorithms (left) and manual domain annotations (right). Correspondence is quantified using ARI, MCC and NMI.

5) For the colorectal cancer MERFISH data analysis I feel that comparing 22 Banksy clusters to 20 non-spatial clusters is an unfair comparison. The authors should instead demonstrate that increasing the number of non-spatial clusters to a higher number of clusters (e.g. 30) does not identify those clusters unique to Banksy.

As suggested, we have now increased the number of non-spatial clusters to 30. Even at this higher resolution, non-spatial clustering failed to identify the cycling cell population identified by BANKSY (Supp. Fig. 17).

Supp. Fig. 17: Comparison of BANKSY and non-spatial clustering on MERSCOPE colorectal tumor data. a) BANKSY UMAP, with a unique cycling cell population indicated by a red arrow. b) Non-spatial UMAP. c) Spatial plots illustrating non-spatial clusters at increasing resolutions, zoomed in to a single gland to visualize individual cells.

6) In the domain segmentation comparisons, the authors compare to a range of widely known domain segmentation algorithms, however, they miss some of the newer algorithms based on graph-based modeling methods, e.g. STAGATE (<https://doi.org/10.1038/s41467-022-29439-6>), GraphST (<https://doi.org/10.1038/s41467-023-36796-3>), which also demonstrates very good

performance (with self-reported ARI's higher than Banksy). I would suggest the authors add at least GraphST and STAGATE to their comparisons.

We thank the reviewer for pointing us to GraphST (Long et al., 2023) and STAGATE (Dong et al., 2022), two recently published domain segmentation methods. We have now added a

quantitative comparison to these two methods, on three benchmark datasets: (1) simulated STARmap, (2) real STARmap, (3) Visium 10x DLPFC.

Simulated STARmap. As in the original manuscript, we benchmarked the domain segmentation methods, now including STAGATE and GraphST, on the simulated STARmap dataset. Across all gene set sizes (400-1,020), BANKSY consistently outperformed all other methods by a substantial margin (revised **Supp. Fig. 3**, see below).

Supp. Fig. 3a-c (revised): Comparison of methods for segmenting domains in simulated STARmap data. a) Median ARI (across three variants of the simulated dataset) for each method, as a function of the number of genes in the dataset. b) Ground truth domain labels.

c) Clustering results from 7 methods on the full 1,020-gene simulated dataset.

Real STARmap. As has been previously noted in computer vision (Picard, D., 2023), we found that the performance of published deep learning methods on the real STARmap dataset (Fig. 5c) was substantially influenced by choice of random seed (Reviewer Fig. 6).

To robustly estimate performance, we therefore calculated the median ARI across 11 seeds. These included all seeds used in the published domain segmentation algorithms, as well as additional seeds commonly used in machine learning. For consistency, we also used these 11 seeds in BANKSY's Leiden clustering step. In terms of median ARI, BANKSY was again the highest performer on this dataset (**Reviewer Fig. 6; Fig. 5c**; see below). Moreover, BANKSY's performance was highly stable across random seeds.

Reviewer Fig. 6: Domain segmentation comparison (boxplot) on the real STARmap 1,020-gene mouse visual cortex dataset. GraphST, STAGATE and BANKSY were run with 11 different random seeds and ARI relative to the reference domain annotation was calculated in each case.

Fig. 5c: Domain segmentation comparison on the real STARmap 1,020-gene mouse visual cortex dataset. Median ARI across 11 random seeds is indicated in parentheses for each algorithm. L1-6: six neocortical layers; cc: corpus callosum; HPC: hippocampus.

Visium DLPFC datasets. In benchmarking the 7 domain segmentation methods (including GraphST and STAGATE) on the 12 Visium DLPFC datasets, we harmonized the label smoothing step to smooth across the 6 nearest neighbors, as originally implemented in SpaGCN. We did not adopt GraphST's approach of smoothing across 50 nearest neighbors, since this resulted in over-smoothing and loss of resolution in detecting the thinner layers, namely Layers 2 and 4 (data not shown). As above, we

again calculated the median performance of each deep learning algorithm across 11 random seeds, and then calculated the median of these medians across the 12 datasets. GraphST, BANKSY and STAGATE were the top performers on this dataset, with no significant difference in performance between the three (Fig. 5b, see below). BANKSY was again the most robust method, showing the lowest variation in ARI across the 12 datasets (lowest interquartile range).

Fig 5b: Boxplot of domain segmentation accuracy on the 12 DLPFC datasets, quantified using adjusted Rand index (ARI). For deep learning methods and BANKSY, the sample ARIs shown correspond to the seed run which achieved the median ARI across 11 random seeds. Center line: median of the 12 data points, height of box: interquartile range (IQR), whiskers:

$1.5 \times \text{IQR}$, *: $p < 0.05$, ***: $p < 0.001$ (paired one-sided Wilcoxon signed rank test).

Summarizing across the three benchmarks described above, BANKSY was the top performer overall at domain segmentation, and also consistently the most robust to variation in gene set size, random number seed and dataset.

Scalability Benchmark. We have now added the CPU computation times of GraphST

and STAGATE to the revised Fig. 6 (see below). In addition, to address a comment from Reviewer 2, we have included a new supplementary figure (Supp. Fig. 35) showing scalability benchmarking on GPUs for the deep learning methods GraphST, SpaGCN and STAGATE. For the latter, we used the Nvidia Tesla T4 (16 GB RAM, CUDA version 12.0) and followed the same procedure as in Fig. 6. As can be seen from these two figures, BANKSY's CPU run time was comparable to the GPU run times of GraphST, STAGATE and SpaGCN. Thus, BANKSY can be adopted by researchers without access to specialized GPU servers.

In addition to the fact that access to GPU servers is limited, one other drawback is that GPU memory is typically lower than CPU memory. Most single GPU units are designed with 8-32 GB RAM, whereas CPU-based machines can access system RAM (typically 64-128 GBs).

Consequently, GPU computing reduces the scalability of spatial clustering methods. Thus, BANKSY on a CPU machine is at least 8X more scalable than any of the deep learning methods run on GPUs (STAGATE was out of memory at 250k cells; Supp. Fig. 35).

In summary, BANKSY provides a powerful combination of accuracy, robustness, speed, scalability and accessibility, and thus represents a significant advance over spatial clustering methods.

Fig. 6 (revised): CPU run times of BayesSpace, FICT, Giotto HMRP, GraphST, MERINGUE, SpaGCN, SpiceMix, STAGATE, non-spatial clustering and BANKSY as a function of cell number. All methods were benchmarked on a 16 CPU 128 GB machine, with a cut-off of 16 hours. Runtimes are shown up to the maximum cell number accommodated by each method.

Supp. Fig. 33: GPU runtimes of GraphST, SpaGCN, SpiceMix and STAGATE as a function of cell number. Runtimes are shown up to the largest cell number accommodated by each method.

Minor

- I find the example presented in Suppl Fig 1 to be very crude. As far as I am aware, there are little to no commonly used tools for spatial clustering that are based on that simple model. I suggest both the figure and associated text be removed, at the author's discretion.

We agree with the reviewer that the spatial clustering approach illustrated in **Supp. Fig. 1** may not be ideal, since it makes no distinction between spatial coordinates and genes, except for a scale factor. Nevertheless, this clustering approach has been used in Fig. 7f of (Maynard et al., 2021), perhaps because it is the most obvious strategy for

combining two sets of features. Interestingly, its performance in their analysis was superior to that of non-spatial clustering. We therefore propose to retain this supplementary figure as a starting point for discussing how expression and spatial features could be combined for cell-typing and domain segmentation.

- Given the incorrect splitting of L2/3 (despite the higher ARI), the authors should clearly state that subjective evaluation of L2/3 shows that Banksy performed worse than some of the other algorithms such as BayesSpace, Spice Mix etc despite the higher metrics, suggesting a need for use of newer metric to evaluate domain contiguity etc.

We respectfully disagree that the other methods outperformed BANKSY in terms of subjective clustering quality. In fact, the lower ARI, NMI and MCC of the other algorithms (Supp. Fig. 30b) is consistent with multiple qualitative flaws evident in their results (Fig. 5c):

1. **Domain contiguity.** A fundamental requirement of domain segmentation algorithms is spatial contiguity of domain labels. BANKSY fulfills this requirement: all domain segments identified by BANKSY are spatially contiguous (Fig. 5c). In contrast, all other methods, to varying degrees, generate cell labels that are spatially discontinuous. For instance, BayesSpace, SpaGCN and SpiceMix populated almost all layers (except for cc) with intermixed cells from multiple clusters. GraphST intermixed HPC, L2/3 and L5 cells within L5, and HPC and cc cells within cc. STAGATE mixed labels for L5, HPC and L1 within L5 and L1. These discrepancies in domain contiguity are reflected in the lower ARI, NMI and MCC scores of the corresponding methods.
2. **Mislocalization and duplication of domains.** With the exception of L2/3, BANKSY places each domain in a single location, with accurate boundaries. In contrast, Giotto segments the dataset almost arbitrarily. GraphST creates a second, artifactual L5 domain within HPC, and also splits L2/3 very similarly to BANKSY. STAGATE places HPC cells in HPC, and also in two additional well separated locations: L5 and the boundary of L1 with L2/3. Again, these errors contribute to the lower accuracy scores of the methods in question.

In contrast to the above, the error in BANKSY's output is visually prominent, but restricted to a single domain (orange-green split in L2/3). Moreover, we emphasize again that BANKSY consistently outperforms the other algorithms by 3 distinct, widely-used

metrics: ARI, NMI and MCC. Below, we provide Supp. Fig. 30b as an alternate visualization of algorithm accuracy.

Fig. 5c: Domain segmentation comparison on the STARmap 1020-gene mouse visual cortex dataset. L1-6: six neocortical layers; cc: corpus callosum; HPC: hippocampus. Numbers in parentheses represent ARIs computed against implied domain annotations.

Supp. Fig. 30b. STARmap benchmark, 1,207 cells (same dataset as Fig. 5c): alluvial plot showing correspondence between cluster labels assigned by spatial algorithms (left) and manual domain annotations (right). Correspondence is quantified using ARI, MCC and NMI.

- The authors do not describe in detail how the algorithm would deal with rare cell types.

We thank the reviewer for raising this question. A key point to note here is that the majority of existing spatial clustering algorithms were designed for tissue domain segmentation and rely on spatial smoothing of cellular expression traits, either explicitly (for example, using a convolutional neural net) or implicitly (HMRF: Potts model). Consequently, they have limited ability to detect rare, interspersed cell-types. In contrast, BANKSY tackles both domain segmentation and cell-typing. In cell-typing mode, BANKSY applies a higher weight to the cell's own transcriptome than to the smoothed neighborhood transcriptome. Consequently, BANKSY retains the ability to detect rare cell-types that are surrounded by dissimilar cells.

As we now note in the Results section (with more details in Supp. Section 3), BANKSY is able to detect less-common cell-types in complex tissues. Below, we list 5 such examples, covering spatially localized as well as dispersed cell-types (Reviewer Fig. 7 below, corresponding to the new Supp. Figs. 16, 18 and 20c, d):

1. In the CosMx dataset, BANKSY detects a macrophage subpopulation that constitutes only 0.65% of cells in the dataset (Supp. Fig. 18). This population

specifically expresses multiple markers of M2 macrophages.

2. In the MERFISH dataset, ependymal cells (2.41% of cells, **Reviewer Fig. 7a**) are spatially contiguous, and microglia (2.12% of cells, **Reviewer Fig. 7b**) are dispersed. These cell-types were defined by Moffitt et al. on the basis of known markers, and they match the corresponding BANKSY clusters (ARI: 0.958 and 0.807, respectively).
3. In our VeraFISH data, oligodendrocyte precursors (2.28% of cells, **Reviewer Fig. 7d**) and microglia (2.72% of cells, **Reviewer Fig. 7e**) are spatially dispersed in the lateral thalamic nuclei, the fornix regions, and the general hippocampal area (labeled HPF in the schematic in **Reviewer Fig. 7c** below). These were defined on the basis of known marker genes (Gpr17 and Pdgfra for OPCs, and Cx3cr1 and Csf1r for microglia).

Reviewer Fig. 7 (corresponding to **Supp. Figs. 16** and **20c, d**). Rare cell-types in the mouse hypothalamus (MERFISH; a, b) and hippocampus (VeraFISH; c-e) datasets. (a) Spatially localized ependymal cells in the mouse hypothalamus data, as labeled by Moffitt et al. (top) and BANKSY (bottom). (b) Similar to (a), but for the spatially dispersed microglial population. (c) Allen Brain Atlas schematic of the hippocampal formation (HPF) and surrounding thalamic and fornix regions. This schematic is laterally inverted with respect to panels d, e. (d, e) Oligodendrocyte precursor cells and microglia distributed throughout the HPF, thalamic nuclei and fornix regions.

- SpiceMix is now published, so the bioRxiv reference should be updated

We have now updated the SpiceMix reference to cite the Nature Genetics publication.

- Fig S19 could be arranged as a matrix with all rows showing comparable regions.

We agree, and have now aligned all rows in **Supp. Fig. 23** and **Supp. Fig. 24** (reproduced below) to show comparable regions across each row.

Supp. Fig. 23 (left): Comparing BANKSY and non-spatial clustering results on the VeraFISH dataset to MERINGUE, Giotto (HMRF) and BayesSpace (HMRF).

Supp. Fig. 24 (right): Comparing BANKSY and non-spatial clustering results to SpiceMix (HMRF), SpaGCN and FICT (HMRF). For full captions, see supplementary figures.

- Fig S29, 31,33 panels do not look aligned

We have now aligned all rows in Supp. Figs. 29, 31, 33 (now labeled **Supp. Figs. 38, 40** and **42**) so that each row shows comparable regions.

Reviewer #2:

Remarks to the Author:

The authors developed Building Aggregates with a Neighborhood Kernel and Spatial Yardstick (BANKSY) that unifies clustering and domain segmentation analysis. BANKSY uses two spatial kernels to encode the transcriptomic signatures of a microenvironment around a cell, including a weighted mean expression (a Gaussian weighting envelope based on the radial physical distance between neighbouring cells) and an azimuthal Gabor Filter (AGF, for analysing the spatial texture of neighbourhood gene expression). By concatenating the non-spatial gene expression matrix with the average neighbourhood expression matrix and an AGF matrix, BANKSY constructs a neighbour-augmented matrix. AGF, an effective method for extracting frequency and spatial location information in a polar coordinate system, is an innovative approach that the authors introduced to analyse spatial transcriptomics data. The speed and scalability gain derives from the fast and linear construction of the neighbour augmented matrix. BANKSY is faster and more scalable to millions of cells, as opposed to other tools, especially those dependent on probabilistic optimisation of parameters like BayesSpace, Giotto HMRF and SpiceMix or those models using neural network models like SpaGCN.

The manuscript is well written. The code is available in R and Python, and the documentation is detailed and helpful, although it can be polished further (minor comments below). The authors have performed comprehensive benchmarking and parameter sweep analyses and convincingly show that this approach improves the domain mapping task.

We greatly appreciate the reviewer's feedback on our manuscript and on the BANKSY software package, which has contributed significantly to the refinement and advancement of our work. We are particularly appreciative of the reviewer's comments regarding the novelty of the azimuthal Gabor filter (AGF), and on the speed and scalability of BANKSY.

My main suggestions are below:

1. BANKSY is capable of processing million-cell datasets, making it suitable for the latest versions of the new generation of spatial single-cell resolution technologies. It would be useful to provide examples of how BANKSY works with common data in the field, like Xenium and CosMX. Would BANKSY work for high plex protein data like Akoya FusionCycler data?

As described below, we have now updated the manuscript to include analysis of CosMx (RNA) and Akoya PhenoCycler (a.k.a. CODEX, protein) spatial omics data. Due to the versatility and broad applicability of BANKSY's feature augmentation mechanism, we were able to retain BANKSY's default parameter settings in these analyses. Xenium: a recent preprint from Mats Nilsson's group performed a comprehensive benchmarking of spatial domain segmentation algorithms for Xenium data, and found that BANKSY yielded the most accurate clusters (Salas et al., 2023). They therefore recommended that Xenium users adopt BANKSY for spatial clustering. We now cite this preprint in the revised manuscript.

On a CosMx healthy colon dataset (Garrido-Trigo et al., 2023), BANKSY in cell-typing mode uniquely identified a subpopulation of macrophages (Supp. Fig. 18a, black arrow) that was spatially enriched in the colorectal submucosal layer (Supp. Fig. 18b-c). Based on up-regulation of M2 macrophage markers (Murray et al., 2014; Kosaric et al., 2020), we annotated this population as M2 macrophages.

Next, we examined a CODEX multiplexed protein imaging dataset from human intestine (Hickey et al., 2023). In this study, the authors proposed a hierarchical spatial model of intestinal structure, in which cell-types formed neighborhoods, which were organized into communities, and communities in turn formed tissue units. BANKSY in domain segmentation mode at low resolution accurately identified major tissue domains which matched the published tissue unit annotations (Supp. Fig. 32a-c). At a higher setting of the Leiden clustering resolution parameter, BANKSY identified communities that better correlated with antibody expression than the authors' annotations (Supp. Fig. 32d-e).

Lastly, BANKSY's unprecedented generality is demonstrated by the fact that it has shown

efficacy on datasets completely unrelated to molecular profiling, generated using multilayer brain MRI scans (Shamir et al. 2022).

a. Non-spatial

Supp. Fig. 32: BANKSY cell-typing on a CosMX healthy human colon dataset. a) Non-spatial and BANKSY UMAPs colored by their respective cluster labels, with a cell-type annotation legend at the right for BANKSY clusters. b) Spatial cell-type plots for the full dataset (left) and a single ROI (right), colored by BANKSY cluster labels. c) Left: spatial distribution of the two macrophage clusters detected by BANKSY, with the Mac.2 cluster

enriched in the submucosa. Right: violin plot of DE genes between the two macrophage subpopulations.

a. Tissue units

b. BANKSY

c. Non-spatial

d. Community

e. CD66

Supp. Fig. 32. BANKSY tissue domain segmentation and community identification on CODEX multiplexed protein imaging data from human intestine. a) BANKSY UMAP (left) and spatial plots (right) coloured by the published (Hickey et al.) tissue unit annotations. b) Same plots, colored by BANKSY clusters. c) Non-spatial UMAP and spatial cell maps colored by non-spatial cluster labels. d) Comparison of BANKSY and non-spatial clusters (right) to published cell community annotations (left). e) Left: spatial map of CD66 protein expression and box plots of CD66 expression in purple (CD66+ Mature Epithelial) and adjacent red (Plasma Cell Enriched) cell clusters defined by Hickey et al. vs BANKSY. Right: similar plots for the smooth muscle subpopulation (pink) uniquely detected by BANKSY, relative to the remaining smooth muscle cells (grey).

2. The spatial information used in BANKSY is for cells in one tissue section. How can BANKSY be applied to integrate data from multiple tissue sections or multiple regions of interest (ROIs) from the same sample?

We apologize for the lack of clarity. BANKSY does indeed have a multi-sample mode, where multiple samples are combined for processing by simply concatenating the neighbor-augmented matrices. For example, as described in the Methods section, the brain organoid domain segmentation which identified necrotic regions was performed by applying BANKSY to concatenated cells from 2 samples (Fig. 5e). In this case, we reduced batch effects by z-transforming (zero centering and scaling to unit variance) each gene on a per-sample basis. We now indicate this in the Results as well.

For benchmarking on DLPFC Visium data from Maynard et al. (2021), we ran BANKSY one sample at a time as is the norm in the field (Hu et al. 2021, Zhao et al. 2021, Dong & Zhang, 2022, Long et al. 2023). However, BANKSY can also be run in multi-sample mode to jointly cluster cells or spots from multiple datasets, as illustrated in the following vignette: <https://prabhakarlab.github.io/Banksy/articles/dlpfc-analysis.html>.

While simple concatenation of datasets, with or without z-transformation, may be sufficient in some cases, it may not always succeed in overcoming batch effects. Thus, multi-sample analysis could require the use of explicit batch-correction techniques. For example, in Supp. Fig. 9 of Huuki-Myers et al. (2023), the authors used non-spatial batch correction (Harmony) to integrate 12 Visium DLPFC datasets, and then fed the resulting integrated object to Bayesspace for spatial clustering (Harmony->BayesSpace). The

limitation of this approach is that spatial information is ignored in batch correction. To explore a more spatially informed strategy, we applied Harmony to the PC scores computed using BANKSY's spatial neighbor-augmented embedding of Visium spots from the 12 datasets, and then clustered the spots in the resulting integrated space (BANKSY->Harmony). Note that this did not require any change in the bioconductor software for BANKSY - we merely inserted one function call to Harmony after the BANKSY embedding step. Joint BANKSY clustering of the 12 datasets in this manner outperformed non-spatial clustering, BayesSpace without batch correction, and also BayesSpace with Harmony batch correction (Supp. Fig. 34). Thus, BANKSY straightforwardly facilitates spatially informed batch correction.

Supp. Fig. 34. Benchmarking performance of non-spatial clustering, BayesSpace and BANKSY with and without batch correction on joint clustering of spots from 12 DLPFC Visium datasets. ARI: adjusted Rand index. NMI: normalized mutual information. MCC: Matthew's correlation coefficient. Black lines: median score. Red dotted lines: mean score.

3. In the section 4.1 BANKSY algorithm, the authors describe clearly how the Neighbor

augmented matrix can be computed. However, it is recommended to add more information and discussion on how this matrix is used for downstream analysis. For example, PCA is applied to this matrix, followed by Leiden community detection. Is this matrix used for other types of analyses and whether multiple matrices are generated for cell type identification and domain mapping?

We apologize for the lack of clarity. We have now given a more detailed description in Methods Sections 4.1 and 4.1.4 of how the BANKSY matrix is constructed and used for cell-typing and domain segmentation. Briefly, the same neighbor-augmented data matrix is clustered for both for cell-typing and domain segmentation, except that the three components of the matrix are scaled differently depending on the value of lambda.

4. Using the augmented matrix described above, the final results highly depend on lambda that determines the relative weights/importance of the original gene-by-cell matrix vs. the weighted mean matrix vs. the AGF matrix (equation 2, justified in Supp Note 2). The authors have assessed the effect of varying lambda values from 0 to 0.3 (Supp Fig 31). With a lambda at 0.3, the weights for matrix C, M and G are 0.83, 0.45, and 0.32, respectively, giving more weight to the original gene expression matrix. It would be helpful also to see the effect of lambda values ≥ 0.8 where BANKSY prioritises more spatial domain information. For example, with a lambda at 0.8, the corresponding weights for C, M, and G are 0.44, 0.73 and 0.52.

We thank the reviewer for the suggestion to provide a sweep for values of lambda ≥ 0.8 (i.e., for domain segmentation). We chose to use the mouse hypothalamus MERFISH data to demonstrate a thorough sweep of lambda values, as the domains identified could be qualitatively checked against reference annotations from the Moffitt et al. study and Allen Brain Atlas (as was also done in Fig. 5c). We now provide Supp. Fig. 37b, which shows the result of a lambda sweep for values of lambda from 0.75 to 1. Tissue domains identified by BANKSY showed only limited variation across this parameter sweep, and corresponded closely to domain annotations in Moffitt et al. in all cases. This result further supports the robustness of BANKSY to parameter variation.

Supp. Fig. 37b (new): Lambda parameter sweep on the mouse hypothalamus dataset. For reference, Moffitt et al. annotations are indicated in the second row, with corresponding Allen Brain Atlas annotations in parentheses when their acronyms differ from those of Moffitt et al. PVA (PVT): paraventricular thalamic nucleus; Fx: fornix; PaAP (PVH): paraventricular hypothalamic nucleus; BNST (BST): bed nucleus of the stria terminalis; 3V (V3): third ventricle; MPN: medial preoptic nucleus; MPA (MPO): medial preoptic area.

5. Could the authors add more justification on why matrix G is assigned a lower weight than matrix M?

As shown below in **Reviewer Fig. 8**, the first 10 principal components explain a smaller

fraction of variance in the G matrix (AGF output) relative to the M matrix (neighborhood mean). In other words, the degree of correlated variation is greater in the M matrix, indicating that the latter is likely to contain a greater degree of information for discriminating cell-types and states. We therefore assign the G matrix a smaller weight. For simplicity, we set the weight of the G matrix such that its total variance would be half that of the M matrix.

Reviewer Fig. 8: VeraFISH Hippocampus dataset: scree plot of variance explained by each of the top 10 principal components of the M and G matrices.

6. It is well demonstrated that the BANKSY algorithm improves tissue domain mapping, but could the authors explain how BANKSY improves cell type identification?

As shown schematically in **Fig. 1c** (simplified 2D plot, reproduced below), the addition of neighborhood expression features provides novel dimensions to the embedding that improve BANKSY's ability to discriminate between closely related cell-types. This improvement relies on the intuition that cell types tend to have characteristic spatial neighborhoods.

We verified this intuition in our analysis of the spatially-restricted mature oligodendrocyte (MOD) subtypes, where we used self (x-axis) vs neighbor (y-axis) metagene expression to generate the same 2D visualization. As seen in **Fig. 2e** reproduced below, we indeed see a clear separation in neighbor gene-expression space

(y-axis) and overlapping gene-expression signatures from the cells themselves (x-axis), which matched our intuition in Fig. 1c.

C Cells lifted to neighbour-augmented space

Left (Fig. 1c): Simplified schematic of two distinct cell-types in the neighbor-augmented space. The neighbor expression features, representing the local microenvironment, help to separate two clusters that would be difficult to separate based on the cells' own expression alone. For simplicity, we show 'pure' microenvironments containing only a single cell-type (Cell-Type 1 in Zone A and Cell-Type 2 in Zone B), though BANKSY is equally applicable to heterogeneous microenvironments containing mixtures of cell-types (Supp. Fig. 2).

Right (Fig. 3d, e): Heatmap of genes differentially expressed (DE) between the two MOD subtypes, showing MERFISH expression values z-scaled across the mature oligodendrocytes. (e) Average expression MOD DE genes from (d), plotted in the first two components of BANKSY's neighbor-augmented product space (X-axis: own expression; Y-axis: mean neighborhood expression)

Minor comments:

- Fig 1b not cited in the main text

We apologize for the omission. We have now cited **Figs. 1a-c** as follows:

“BANKSY uses the mean neighborhood expression and the AGF (**Fig. 1a, b**) to compute a representation of the transcriptomic microenvironment around each cell. Importantly, the AGF (**Fig. 1b**), which can be thought of as an approximate measure of the magnitude of the gradient of gene expression in each cell’s neighborhood, is invariant to sample rotation.

Next, these values are used as additional features to embed cells in a neighbor-augmented product space, as described in Methods Section 4.1 (**Fig. 1c**).”

- In Fig 3h, are the number of clusters from non-spatial and BANKSY the same for comparing the cell type annotation?

For this dataset, we ran non-spatial and BANKSY clustering at the same resolution (0.7), which effectively identified major cell-types. This analysis identified 22 BANKSY clusters and 18 non-spatial clusters. As requested by Reviewer 1, we tested if increasing the number of non-spatial clusters could reproduce the cycling cell population identified by BANKSY. However, increasing the number of non-spatial clusters to match BANKSY (22 clusters), and even as high as 30, did not yield the cycling cell population uniquely identified by BANKSY (**Supp. Fig. 17c**).

Supp. Fig. 17: Comparison of BANKSY and non-spatial clustering on MERSCOPE colorectal tumor data. a) BANKSY UMAP, with a unique cycling cell population indicated by a red arrow. b) Non-spatial UMAP. c) Spatial plots illustrating non-spatial clusters at increasing resolution, zoomed in to a single gland to visualize individual cells.

- For deep learning based methods like SpaGCN, could the authors make use of GPUs as this is often needed for these methods.

We thank the reviewer for this suggestion. We have now added **Supp. Fig. 35**, showing scalability benchmarking on a GPU for the deep learning methods SpaGCN, GraphST and STAGATE. We also tested SpiceMix, since it is GPU-compatible. We used the Nvidia

Tesla T4 (16 GB RAM, CUDA Version 12.0) and followed the same procedure as in Fig. 6.

BANKSY's CPU run time (3.25 min for 100k cells) was comparable to the GPU run times of GraphST, STAGATE and SpaGCN. Thus, BANKSY can be adopted by researchers who don't have access to specialized GPU servers.

In addition to the fact that access to GPU servers is limited, one other drawback is that GPU memory is typically lower than CPU memory. Most single GPU units are designed with 8-32 GB RAM, whereas CPU-based machines can access system RAM (typically 64-128 GBs).

Consequently, GPU computing reduces the scalability of spatial clustering methods. Thus, BANKSY on a CPU machine is at least 8X more scalable than the three deep learning methods run on a GPU (STAGATE ran out of memory when tested at 250k cells; Supp. Fig. 35).

Supp. Fig. 35: GPU runtimes of GraphST, SpaGCN, SpiceMix and STAGATE as a function of cell number. Runtimes are shown up to the largest tested cell number accommodated by each method.

- For the Python version, would be helpful for installation to make it available through Anaconda or PyPI.

We agree, and have now provided detailed instructions on how to set up the user's Conda environment and clone the BANKSY repository from our GitHub page. We confirmed that we were able to replicate the environment and run BANKSY on a separate workstation. We plan to submit the updated version of the code to PyPi to improve the ease of installation for users of the Python version of BANKSY.

- The R installation went very smoothly (Macs users might need to install the additional bitops package)

Thank you! We will add that dependency in the installation instructions explicitly.

The default R tutorials are well written and the workflow is understandable, but some commands wouldn't work as they are in the current tutorial:

<https://prabhakarlabor.github.io/Banksy/> (for example the SubsetBanksy as an error: nCount not found; a very small error that can be fixed easily)

Our apologies, it seems that some commands were missed when we updated our code base between versions. We will fix all such issues in the coming R version (0.2.5), which uses the BioConductor / SingleCellExperiment data classes and methods, per Reviewer 3's suggestion.

Reviewer #3:

Remarks to the Author:

The authors of this manuscript present a novel and very interesting method for clustering called BANKSY. It has a key parameter, lambda, that controls whether to cluster cells (or spots for spot-based spatially-resolved transcriptomics methods such as Visium) with information borrowed from neighbor spatial cells or not. The degree to which you borrow information from neighbor cells will influence the resulting spatial shape of the resulting clusters. The authors showed with basically as many mainstream technologies you can think of how their method compares to other competitor methods, and if there was some orthogonal measurement ("ground truth"), then they

evaluated their performance against it. BANKSY looks very promising based on these comparisons as well as in terms of scalability. It is available as both an R package as well as a Python software, though I question further below whether this will be the case moving forward. The R package is very well documented and has multiple vignettes at <https://prabhakarlab.github.io/Banksy> showing how to use the software with multiple data types. The R package has been tested in multiple operating systems and in theory users can install it (I did so with R 4.3.1 – aka the latest release version – without any issues). Some of my major comments below are mostly focused on the software rather than the manuscript, since I think that the software can be improved a little bit to have a much larger impact on users. For example, by increasing the interoperability of Banksy with other common data containers. More happy users will lead to more citations ;). In other words, I do think that this extra work will be worth doing as I think that many people will want to use BANKSY to analyze their spatially-resolved data. I think the manuscript is very well written (with some missing method citations, which can be fixed) and the major points are well sustained. I do think that readers will benefit from improving the explanation behind Figure 1b as this novel feature role is easily overlooked. As a future BANKSY user, I'm excited to read this work and hope that your software will be maintained for years to come. Though I understand that maintaining open source software is challenging in this funding climate.

We are deeply grateful for the reviewer's positive evaluation of our manuscript, and recognition of BANKSY's promise for broad adoption and impact in the field of spatial omics. We are also grateful for the detailed suggestions for improving software quality and usability. These suggestions, in particular the Bioconductor submission and the use of SingleCellExperiment and SpatialExperiment containers, have enabled us to substantially improve our software and will allow users to better integrate BANKSY into their workflows.

We are fully committed to continuing developing of BANKSY, since we are using it in our own spatial omics datasets (colorectal cancer, brain organoids) in multiple ongoing projects. We have also been approached by multiple researchers who adopted BANKSY after testing it on their own data - we are strongly motivated to support our growing user base by maintaining and upgrading both the R and Python versions.

Major comments

1. To me, Supplementary Figure 1 is too important to be included in the supplement. I recommend adding it after Figure 1C, maybe in a vertical format like | Sup Fig 1A | Sup Fig 1B | etc, or whichever way you think it fits best. I am aware that given a large influence to spatial coordinates does result in something similar to what we see in Supplementary Figure 1D. BANKSY's contributions lie in the land of $\lambda > 0$ and < 1 , which is why I anticipate that many people will use it. Yes, the rest of the main figures are important to convince users across different technologies and applications of the validity of the results as well as the performance of BANKSY. But this figure is the one that best exemplifies the extremes in the values of λ as well as what happens in the interesting part of $\lambda > 0$ and < 1 .

We apologize for the lack of clarity in describing **Supp. Fig. 1**. We have now modified the legend for this figure to explicitly state that cells were clustered using the method in Fig. 7f of Maynard et al. (2021), and described formally in **Supp. Section 1**. In other words, the results in this figure were not generated using BANKSY (they are unrelated to BANKSY's λ parameter). Rather, this figure illustrates the consequences of simply concatenating gene expression levels and x-y coordinates for each cell. In contrast, BANKSY concatenates the gene expression levels of a cell with those of its neighbors.

We agree with the reviewer that it is useful to examine the full range of λ values. Indeed, Reviewer 2 has also raised a similar point (Major Comment 4). In the revised manuscript, we now provide **Supp. Fig. 37**, containing comprehensive λ sweeps on mouse hypothalamus MERFISH data (Moffitt et al., 2021), both for values of λ ranging

from 0 to 1 in steps of 0.2 (**Supp. Fig. 37a**), and a more detailed sweep for values of λ between 0.75 and 1 in steps of 0.05 (**Supp. Fig. 37b**).

Supp. Fig. 37 (new): Lambda sweeps on the MERFISH data. (a) A sweep of lambda from 0 to 1 in steps of 0.2. (b) A more detailed sweep of the high lambda regime showing that the spatial domains found by BANKSY are in broad agreement with the annotation provided by Moffitt et al. and the Allen Brain Atlas (in parentheses).

2. Deposit your code on Zenodo or some more permanent location than GitHub. Otherwise, you could decide to delete your GitHub repository at any moment. See

<https://docs.github.com/en/repositories/archiving-a-github-repository/referencing-and-citing-g->

content for more details. Then on lines 1529 and 1530 you can cite the DOI for your two GitHub repositories.

We would like to assure the reviewer that we are committed to maintaining both the R and Python codebases on Github and will not delete the repositories. Nevertheless, we appreciate the reviewer's suggestion of providing a more permanent and citable location for our package, and thus are in the process of depositing our codebases (and analysis scripts) on Zenodo.

3. I think that authors should include in their discussion the fact that BANKSY was run one sample at a time (like in the case of the Maynard et al, 2021 Visium data). This can be a major limitation if the resulting clusters across samples are not guaranteed to be interpreted as the same unit. That is, cluster 1 from sample 1 might not mean the same thing as cluster 1 from sample 2. See https://edward130603.github.io/BayesSpace/articles/joint_clustering.html for an example of a competitor method that does perform joint clustering. I would also highly recommend discussing how batch effects impact BANKSY when analyzing multiple samples. See also <https://doi.org/10.1101/2023.02.15.528722> supplementary Figure 9 for the effect of batch correction on joint clustering with BayesSpace.

We thank the reviewer for raising this point - multisample analysis is indeed important for BANKSY and related methods. For benchmarking on DLPFC Visium data from Maynard et al. (2021), we ran BANKSY one sample at a time as this is the norm in the field for benchmarking on this dataset (Hu et al. 2021, Zhao et al. 2021, Dong & Zhang, 2022, Long et al. 2023). However, BANKSY can also be run in multi-sample mode to jointly cluster cells or spots from multiple datasets, as illustrated on 4 DLPFC datasets from the same patient in the following vignette: <https://prabhakarlab.github.io/Banksy/articles/dlpfc-analysis.html> and in **Supp. Fig. 33**.

We fully agree also that the effect of batch correction on joint clustering is an important issue. While simple concatenation of datasets, with or without z-transformation (as shown in the vignette), may be sufficient when batch effects are small or involve simple affine transformations, multi-sample analysis could in many cases require the use of formal batch-correction algorithms. In the example pointed out by the reviewer (Supp. Fig. 9 of Huuki-Myers et al. (2023)), the authors used non-spatial batch correction (Harmony) to integrate all 12 Visium DLPFC datasets, and then fed the resulting integrated object to BayesSpace for spatial clustering (Harmony->BayesSpace). The

limitation of this approach is that spatial information is ignored in batch correction.

To explore a more spatially informed strategy, we applied Harmony to BANKSY’s neighbor-augmented spatial embedding of Visium spots from the 12 datasets. In other words, we performed batch correction in “BANKSY space” (BANKSY->Harmony) and then clustered the batch-corrected Visium spots. Note that this did not require any change in the bioconductor software for BANKSY - we merely inserted a function call to Harmony after the BANKSY embedding step. Joint BANKSY clustering of the 12 datasets in this manner outperformed non-spatial clustering, BayesSpace without batch correction, and also BayesSpace with Harmony batch correction (Supp. Fig. 34). Thus, BANKSY straightforwardly facilitates spatially informed batch correction.

Supp. Fig. 34: Benchmarking performance of non-spatial clustering, BayesSpace and BANKSY with and without batch correction on joint clustering of spots from 12 DLPFC Visium datasets. ARI: adjusted Rand index. NMI: normalized mutual information. MCC: Matthew’s correlation coefficient. Black lines: median score. Red dotted lines: mean

score.

4. Where is the code for reproducing your analyses? Please include it on section 5.4 Code availability. Also deposit it on Zenodo or some similar host (Figshare, etc).

We are in the process of uploading the R and Python scripts to Github for reproducing our analyses, and have indicated their availability in section 5.4.

The code to reproduce our analysis on Slide-seq, VeraFISH, MERFISH, MERSCOPE, DLPFC, STARmap (simulated and real), CosMX, CODEX will be available on Zenodo.

5. I think that you will have many people interested in using your software. As an R user, I mostly looked at <https://github.com/prabhakarlab/Banksy>. I think that it would be best to submit your R package to Bioconductor <http://bioconductor.org/> with the Software and Spatial biocView terms http://bioconductor.org/packages/release/BiocViews.html#__Spatial. This will help you and your users by making sure that your software is tested on Windows, Linux, and macOS daily. Yes, I see that you did test it on these OSes at GitHub Actions <https://github.com/prabhakarlab/Banksy/blob/6d76280cad5169be77af8c4b055bcf7a2399c3d3/.github/workflows/check-standard.yml#L21-L23>, but that's only at the time of pushing a commit. A cron job on GitHub Actions could help, but I do think that submitting your software to Bioconductor would be the best.

Following the reviewer's suggestion, we are now in the process of submitting our package to Bioconductor (see our response to the next point for more details).

6. Related to your R package, I'm not sure why you decided to make new container classes instead of re-using classes like SpatialExperiment or SingleCellExperiment. Different classes mean that users will have to re-cast their data and as package authors, you'll have to keep a close eye to make sure users can easily transform their data from those other common containers into your objects. That is, you'll need to keep your asBanksyObject() function that I see at <https://prabhakarlab.github.io/Banksy/articles/single-cell-exp.html#running-banksy> updated. Overall, if you submit your R package to Bioconductor, the reviewers from Bioconductor might have more specific suggested changes with users in mind.

We thank the reviewer for this suggestion. We have now modified our R package for compatibility with the SingleCellExperiment and SpatialExperiment classes, and have updated the documentation to show how users can use BANKSY within the Bioconductor ecosystem (<https://github.com/prabhakarlab/Banksy/tree/bioc>).

7. In contrast to the R version, the Python version has very sparse documentation at https://github.com/prabhakarlab/Banksy_py. It also only has 3 commits versus 203 on the R version. Do you intend to maintain both versions? If you keep only the R version, Python users could use <https://bioconductor.org/packages/zellkonverter/> to transform the AnnData objects into SpatialExperiment ones, then run Banksy in R, and export the clustering results into a format that they can read in Python. Also, if you are not aware, basilisk at <https://www.bioconductor.org/packages/basilisk/> can be used to incorporate Python dependencies on R packages. For example, if there's some computation that for some reason you prefer to run through Python rather than R. So this could be a way to merge your two projects into one.

We agree with the reviewer that the R version is currently more well documented and has had more software commits. This is mainly because the R version was developed earlier and was therefore the primary focus of our software development efforts. We have now updated the Python code and documentation, and uploaded it to the BANKSY GitHub page (https://github.com/prabhakarlab/Banksy_py). We plan to upload the updated code to PyPi concurrently with the submission of the revised manuscript. While the R version is likely to remain our primary focus, we are committed to also maintaining the Python version.

Minor comments

* Page 3, Lines 110 to 116 would benefit from specific citations.

We have now added citations for each of the examples listed in the text, including for cerebral hemispheres, immune cells and brain organoid neuroepithelial buds.

* Page 3, Lines 117 to 119. It would be best to cite the examples on mammalian brain you are

referring to here.

We have now cited Maynard et al. (2021) and the STARmap paper (Wang et al. 2018), which generated human or mouse brain datasets that benefit from cortical layer segmentation by spatial algorithms.

*Page 3 lines 131 to 133. <https://doi.org/10.1101/2023.02.15.528722> did use BayesSpace on 113,927 Visium spots and <https://doi.org/10.1038/s41586-022-05060-x> has 91,517 spots (though they didn't use a spatial clustering method). Of course, it's not millions, but it is larger than some of those initial Visium studies. Also, it feels weird to highlight millions on spots as a limitation of BayesSpace since you ran BANKSY one sample at a time, and did not perform joint clustering like it can be done with BayesSpace https://edward130603.github.io/BayesSpace/articles/joint_clustering.html.

We have now changed the following text:

“Furthermore, as with cell-type clustering, these methods have largely been demonstrated on **small datasets**, and their performance on larger datasets has not been explored.”

to:

“Furthermore, as with cell-type clustering, these methods have mostly been demonstrated on small datasets . **With a few exceptions [28]**, their performance on larger datasets has not been explored.”

The reviewer also raises an interesting point regarding the number of cells per dataset. High resolution spatial omics data from technologies such as MERSCOPE, Xenium and CosMX can include 500k-1M cells from a single tissue section (see, for example, **Fig 3g-j**). The Stereo-seq technology can profile even larger tissue regions, exceeding 1M cells in a single dataset (Chen et al., 2023). Scalability is thus essential for spatial clustering algorithms.

We also appreciate the reviewer’s suggestion regarding integrative analysis of multiple spatial datasets by BANKSY, which we address in our response to Major Comment 3 (see

above).

* Page 4 lines 158 to 160. Please cite Seurat, SingleCellExperiment, Scanpy and other methods.

We have now added citations for Seurat, SingleCellExperiment and Scanpy.

* Figure 1 is quoted on the main text overall, without quoting Fig 1a, Fig 1b, etc. Given that, the caption for Figure 1b is very sparse and doesn't really explain the "real component" versus the "imaginary component" shown, nor the -1 to 1 values shown. Parts a and c are much more explained both in the figures themselves as well as the caption.

We have now provided direct references to the sub-figures of Figure 1, as Reviewer 2 also requested:

“BANKSY uses the mean neighborhood expression and the AGF (Fig. 1a,b) to compute a representation of the transcriptomic microenvironment around each cell. Importantly, the AGF (Fig. 1b), which can be thought of as an approximate measure of the magnitude of the gradient of gene expression in each cell’s neighborhood, is invariant to sample rotation.

Next, these values are used as additional features to embed cells in a neighborhood-augmented product space, as described in Methods Section 4.1 (Fig. 1c).”

We have also added a more detailed explanation in the caption for Figure 1b:

“Heatmap of the real and imaginary components of a gradient-sensitive AGF kernel. Plots show an unnormalized AGF kernel $e^{i\varphi}G(r)$, where $G(r)$ is a radially symmetric Gaussian kernel that decays from magnitude 1 at $r = 0$. The $e^{i\varphi}$ term confers gradient sensitivity: the real part ($\cos(\varphi)$) senses the gradient along the x-axis and the imaginary part ($\sin(\varphi)$) senses the gradient along the y-axis.”

* Figure 2a: by "non-spatial", do you mean BANKSY with $\lambda = 0$? Or some other non-spatial clustering method? Ideally, it would be $\lambda = 0$ to make sure that we are not comparing Leiden vs some other type of non-spatial clustering method (say Louvain). If you are indeed comparing $\lambda = 0$ vs some other λ value, I recommend

specifying this on the caption of figure 2a.

We have now explicitly stated in the legend of **Fig. 2a** that we performed non-spatial clustering using BANKSY with $\lambda = 0$.

* Page 22 line 1007: corresopnding -> corresponding

We have now corrected this typo in the text.

* Page 23, lines 1022 to 1025. Please formally cite the methods you mentioned. For example, with R you can check how to cite a package by using `citation("RcppHungarian")`. A lot of work is put into software development (just like you have done with Banksy) and citations is how such work is formally credited. Related to this, `citation("Banksy")` doesn't link to the pre-print version of Banksy <https://www.biorxiv.org/content/10.1101/2022.04.14.488259v1>. This can be done by adding an inst/CITATION file at <https://github.com/prabhakarlab/Banksy>.

We thank the reviewer for pointing this out. We have now formally cited SciPy for the Python version of the function, and have added a citation for the Rcpp Hungarian package.

We have also now added a citation file to our package (<https://github.com/prabhakarlab/Banksy/blob/bioc/inst/CITATION>).

* Page 23 line 1057. Related to one of my earlier points, I think that it would be best if you changed phrases like "We then performed BANKSY and non-spatial clustering" to something like "We then performed BANKSY ($\lambda = 0.2$) and non-spatial clustering ($\lambda = 0$)" if indeed you used BANKSY with $\lambda = 0$ for the non-spatial clustering. In any case, being explicit about the λ values is best, despite your common λ values being stated in lines 1034 to 1036. I see that you did explicitly the λ values sometimes, like in line 1239. I encourage you to do this all the time.

We fully agree. We now state in the third paragraph of the results section (Page 5) that we ran BANKSY at default settings for all analyses, and have reworded this sentence as: "We then performed BANKSY and non-spatial clustering ($\lambda = 0$)."

We have added a similar clarification wherever we referred to our own non-spatial clustering.

* I'm not familiar with VeraFISH data, but on page 24 line 1093 it caught my attention that you are normalizing cell counts to a single common value (100 in this case). Is this commonly done when analyzing this type of data? If so, you could cite some external source where these processing steps have been used.

Single cell analysis typically packages normalize cell counts to 10000 (Seurat default), or to the median total count across all cells (Scanpy default). We used an approach that borrowed from both: we normalized to a simple round number (Seurat), but chose a number (100) that was similar to the median transcript count per cell (Scanpy). We note also that the actual value of the normalized total count does not matter here, since BANKSY does not log-transform the data (Moffitt et al., 2018). If we had normalized each cell to 10,000 counts instead of 100, this would merely have scaled the entire gene-cell matrix 100-fold. Since BANKSY zero-centers and scales each gene to unit variance after count normalization, it is insensitive to this change.

* Page 24 line 1102. Also line 1205. Please cite scran (it is spelled with a lower case s). Check the output of citation("scran") in R.

We have now corrected the spelling error and added a citation for scran.

* Line 1111. Cite Seurat.

We have now cited Seurat v4 here.

* Line 1120: why are you now using 2% instead of 1% as in line 1091? You also used 2% at line 1218.

Thank you for noticing this oversight. We had indeed used 1% as a threshold for the VeraFISH data, while using 2% as a threshold for the corresponding scRNA-seq validation data. We have now aligned both thresholds to 1% and re-run the analysis. Our

conclusions remain the same (Fig. 4 and Supp. Fig. 20).

* Sometimes you spell out the word "version" like in line 1111 and sometimes you use "v" like in line 1142. I suggest that you use one option to be consistent.

We have now aligned all references to software versions to be spelled out (i.e. "Version x.x").

* Lines 1143 to 1148 (and others) are confusingly written. You make it sound like you are setting a lot of non-default parameters. I strongly recommend being explicit to highlight which are the default values and which are not. For example, the 50,000 MCMC iterations and the burn-in period of 1,000 iterations are the defaults at

<https://github.com/edward130603/BayesSpace/blob/8e9af8f2fa8e93518cf9ecee1ded9ab93e88fffd/R/spatialCluster.R#L123> (gamma is set to NULL but defaults to 3 when platform is equal to "Visium" from the documentation at

<https://edward130603.github.io/BayesSpace/reference/spatialCluster.html> and

<https://github.com/edward130603/BayesSpace/blob/8e9af8f2fa8e93518cf9ecee1ded9ab93e88fffd/R/spatialCluster.R#L162-L164>) whereas k (or N as you used in your manuscript)

= 16 to 20 is not the default. As a reader, I have a hard time distinguishing default parameters vs non-default parameters that you used for other software that I'm not as familiar with. Later on, like in lines 1221 to 1222 you are clearer by saying which function you used and which non-default parameters you used (if any).

We apologize for the missing information. We have now modified the text to explicitly state whether the parameters used are defaults or otherwise.

* Lines 1157 to 1158, cite the smfishHmrf package.

We have now cited the smfishHmrf package here.

* Line 1163. The link is not hyperlinked. Also, hm... I don't think that FICT has a formal citation from what I see at <https://github.com/haotianteng/FICT-SAMPLE>. But please check. You could ask the authors at <https://github.com/haotianteng/FICT-SAMPLE/issues> about how to cite them formally. They could simply deposit their code to Zenodo <https://docs.github.com/en/repositories/archiving-a-github-repository/referencing-and-citing-content>, which would generate a DOI that you could then

formally cite.

We appreciate the feedback. FICT has now been published, and we have updated the citation details for FICT and added a citation here. We have also added a hyperlink to the github address.

* Lines 1169 to 1171, cite SpiceMix and dbscan. Yes, I know that you later cite SpiceMix in line 1175, but since you mentioned it at line 1169, it's best to cite it there too.

We now cite SpiceMix where it first appears in the paragraph, and also added the citation for dbscan.

* Line 1175: I really like how you mentioned the specific Git commit ID!

Thank you!

* Line 1178: cite Leiden. See the output of `"?igraph::cluster_leiden"` which points to "Traag, V. A., Waltman, L., & van Eck, N. J. (2019). From Louvain to Leiden: guaranteeing well-connected communities. *Scientific reports*, 9(1), 5233. doi: 10.1038/s41598-019-41695-z, arXiv:1810.08473v3 [cs.SI]"

We have now added the citation for Leiden clustering.

* Line 1179: cite igraph.

We have now cited the igraph package.

* Line 1210 to 1211 doesn't make sense to me. Maybe there's a typo? Like a gene is either from the mitochondria or it isn't. Did you mean "cells" instead of "genes"?

We thank the reviewer for catching this. Yes, it is indeed a typo and we have corrected the text to read "cells" instead of "genes".

* Line 1230. If there is no explicit version number, specify the date you downloaded the data, in case the name is kept the same ("colon cancer 1"), but the underlying data changes at some future point.

We have now added the date of download.

* Lines 1267. Cite spatialLIBD; check citation("spatialLIBD"). Same in line 1522. We

have now cited spatialLIBD at both locations in the text.

* Line 1272 "normalizing the data to the median library size for each sample" could be a little clearer. I think here you meant to say that for each sample, you normalized the counts per spot to the median library size across all spots of the sample.

We thank the reviewer for pointing this out, and have changed the wording as suggested: "...normalizing the counts per spot to the median library size (total number of transcripts) across all spots of the sample"

* Related to an earlier point, lines 1315 and 1316 say "Default values were used for all other parameters."

We apologize for the confusion. As in the response to the earlier point, we now explicitly state that the parameters described in these lines are defaults:

"Clustering was performed at default settings for Visium data (t-distributed error model, with 50,000 MCMC iterations, a burn-in period of 1000 iterations, and a gamma smoothing parameter of 3)."

* Line 1409, change "We" to "we".

Done.

* Cite scDesign2.

We have now cited scDesign2 in the "Simulated Data" section of Methods.

* Supplementary Figure 28. Can you highlight with some box color annotation the values you used for the main figures? Do a similar thing for Supplementary Figure 29 and other related

figures. It can be hard to remember all the parameters mentioned in the main text when looking at these supplementary figures.

We agree - this is helpful for making Supp. Fig. 28 (now labeled Supp. Fig. 36) clearer. We have now also added red boxes to Supp. Figs. 36-51 to highlight the default parameter settings.

Supp. Fig. 36: Parameter sweep on the 10X Visium DLPFC dataset, varying the number of features used (highly variable genes), number of principal components (PCs) and k_{expr} . (a) k_{expr} held constant. (b) Number of highly variable genes held constant. (c) Number of principal components (PCs) held constant. Red boxes show default values.

* Fix the citation in supplementary page 45. It says citation 9, but it's 7 on the main paper. Actually, I see that you duplicated references in the supplementary material. Hm... I'd check with the editors if

that's ok if they prefer to have unique main text references vs supplementary text references (so no overlap between them).

We thank the reviewer for raising this issue. We will work with the editorial team to ensure that the citations are arranged according to their policies.

* Some of your supplementary citations are missing information. I recommend including the URL for some of them like reference 1, 4, 6, and 7.

We have now added URLs for these references.

* * I'm aware that `spatialLIBD::fetch_data()` can be used to download the Maynard et al 2021 data. So at <https://prabhakarlab.github.io/Banksy/articles/dlpfc-analysis.html> I was a bit disappointed to not see how you transformed the `SpatialExperiment` objects from that study into a `BanksyObject`. This information is not present at <https://prabhakarlab.github.io/Banksy/reference/dlpfc151673.html> or similar files either.

We apologize for this omission, and now provide a vignette showing end-to-end analysis of BANKSY on the data from Maynard et al. (2021), starting with the data download from `spatialLIBD` (<https://github.com/prabhakarlab/Banksy/blob/bioc/vignettes/multi-sample.Rmd>) .

* Hopefully the timing of "Lin et al., manuscript in preparation" works in such a way that you will at least be able to cite the pre-print version of it by the time you are doing the proofs of this manuscript.

We hope we will be able to cite Lin et al. at least in this form, as suggested by the reviewer.

* I liked Supplementary Figures 19 and 20 in terms of your overall layout to show results across multiple `N` values.

Thank you!

* I appreciated the detailed response to previous reviewer comments related to SpiceMix. It reminded me a bit of <https://twitter.com/helucro/status/1597579271945715717>

Indeed, the output and performance of some clustering algorithms appears to be highly variable, as we demonstrated in the case of SpiceMix. In contrast, BANKSY has relatively stable performance across datasets and across choices of random number seed.

References

- Chen, A. et al. Single-cell spatial transcriptome reveals cell-type organization in the macaque cortex. *Cell* **186**, 3726–3743.e24. <https://doi.org/10.1016/j.cell.2023.06.009> (Aug. 2023).
- Chidester, B., Zhou, T., Alam, S. & Ma, J. SpiceMix enables integrative single-cell spatial modeling of cell identity. *Nature Genetics* **55** (1), 78–88 (2023).
- Dong, K. & Zhang, S. Deciphering spatial domains from spatially resolved transcriptomics with an adaptive graph attention auto-encoder. *Nature Communications* **13**. <https://doi.org/10.1038/s41467-022-29439-6> (Apr. 2022).
- Garrido-Trigo, A. et al. Macrophage and neutrophil heterogeneity at single-cell spatial resolution in human inflammatory bowel disease. *Nature Communications* **14**. <https://doi.org/10.1038/s41467-023-40156-6> (July 2023).
- Hickey, J. W. et al. Organization of the human intestine at single-cell resolution. *Nature* **619**, 572–584. <https://doi.org/10.1038/s41586-023-05915-x> (July 2023).
- Hu, J. et al. SpaGCN: Integrating gene expression, spatial location and histology to identify spatial domains and spatially variable genes by graph convolutional network. *Nature Methods* **18**, 1342–1351. <https://doi.org/10.1038/s41592-021-01255-8> (Oct. 2021).
- Huuki-Myers, L. et al. Integrated single cell and unsupervised spatial transcriptomic analysis defines molecular anatomy of the human dorsolateral prefrontal cortex. *bioRxiv*. <https://doi.org/10.1101/2023.02.15.528722> (Feb. 2023).
- Kosaric, N. et al. Macrophage Subpopulation Dynamics Shift following Intravenous Infusion of Mesenchymal Stromal Cells. *Molecular Therapy* **28**, 2007–2022.

<https://doi.org/10.1016/j.ymthe.2020.05.022> (Sept. 2020).

Long, Y. et al. Spatially informed clustering, integration, and deconvolution of spatial transcriptomics with GraphST. *Nature Communications* **14**.
<https://doi.org/10.1038/s41467-023-36796-3> (Mar. 2023).

Maynard, K. R. et al. Transcriptome-scale spatial gene expression in the human dorsolateral prefrontal cortex. *Nature Neuroscience* **24**, 425-436.
<https://doi.org/10.1038/s41593-020-00787-0> (Feb. 2021).

Moffitt, J. R. et al. Molecular, spatial, and functional single-cell profiling of the hypothalamic preoptic region. *Science* **362**. <https://doi.org/10.1126/science.aau5324> (Nov. 2018).

Murray, P. J. et al. Macrophage Activation and Polarization: Nomenclature and Experimental Guidelines. *Immunity* **41**, 14-20.
<https://doi.org/10.1016/j.immuni.2014.06.008> (July 2014).

Picard, D. Torch.manual_seed(3407) is all you need: On the influence of random seeds in deep learning architectures for computer vision. arXiv: 2109.08203 [cs.CV].

Salas, S. M. et al. Optimizing Xenium In Situ data utility by quality assessment and best practice analysis workflows. *bioRxiv*. <https://doi.org/10.1101/2023.02.13.528102> (Feb. 2023).

Shamir, I., Assaf, Y. & Shamir, R. Clustering laminar cytoarchitecture: in vivo parcellation based on cortical granularity. *bioRxiv*.
<https://doi.org/10.1101/2022.10.29.514347> (Oct. 2022).

Zhang, M. et al. Spatially resolved cell atlas of the mouse primary motor cortex by MERFISH. *Nature* **598**, 137-143. <https://doi.org/10.1038/s41586-021-03705-x> (Oct. 2021).

Zhao, E. et al. Spatial transcriptomics at subspot resolution with BayesSpace. *Nature Biotechnology* **39**, 1375-1384. <https://doi.org/10.1038/s41587-021-00935-2> (June 2021).

Decision Letter, second revision:

12th Oct 2023

Dear Shyam,

Thank you for submitting your revised manuscript "BANKSY: A Spatial Clustering Algorithm that Unifies Cell Typing and Tissue Domain Segmentation" (NG-A62310R2). It has now been seen by the original referees and their comments are below. The reviewers find that the paper has improved in revision, and therefore we'll be happy in principle to publish it in Nature Genetics, pending minor revisions to satisfy the referees' final requests and to comply with our editorial and formatting guidelines.

Sincerely,

Michael Fletcher, PhD
Senior Editor, Nature Genetics

ORCID: 0000-0003-1589-7087

Reviewer #1 (Remarks to the Author):

I have reviewed the resubmitted work by Vipul Singhal, Nigel Chou, and colleagues. Their revised work addresses my major concerns well, and now I feel that the work more convincingly demonstrates the advantages of Banksy over other non-spatial and other spatial clustering approaches. However, there are still two easily addressable issue I would like the authors to consider. The first relate to the installation instructions of the tool (points 1-6), and the second relates including additional text in the manuscript that corresponds to reviewer/editor only discussion (points 7,8).

Despite these issues, I am convinced that work presented is indeed worthy for consideration for a top tier journal. Well done!

Installation of Banksy R

1) The version of R is not defined. Perhaps the authors should encourage creating a fresh conda environment for Banksy, only defining the version of R, after which the remote installation can take place.

2) I encountered linux dependencies for installation via R. One is listed on the Banksy GitHub, but two were not. Perhaps the authors could add these dependencies to a potential conda setup for install a base version of R, and these additional dependencies. I found version of all three packages below on conda-forge.

2a) I encountered the leidenag issue in that it required, requires igraph. These issues were resolved via instructions on the leidenag GitHub, but authors could consider mentioning how to remedy the issue on the Banksy GitHub. In my case it was solved by "sudo apt-get update" and "sudo apt-get install libxml2-dev libgmp-dev libgmp-dev" on my personal machine. I do not know how easily this can be solved when users do not have root/sudo access to a machine, as is common with university HPC systems.

2b) I also had to install cairo ("sudo apt-get install libcairo2-dev"). This issue is not stated on the Banksy GitHub.

2c) I also had to install curl ("libcurl4-openssl-dev"). This issue is not stated on the Banksy GitHub.

3) Remark: apart from the above issues, I could complete the basic tutorial.

Installation of Banksy_py

4) Banksy_py, GitHub section on "Installation in new Anaconda environment (recommended)"

4a) After cloning the repo, the "cd Banksy_py" step is missed

4b) If the "python utils" library is already installed via pip, the "banksy Utils" is not accessible. I could get around this by uninstalling python utils via pip, but this is not an ideal solution.

4c) Remark: apart from the above issues, I could run tutorial for slideseq v1 to end on a 8Gb laptop in very little time

5) Banksy_py, GitHub section on "Quick installation (via pip) Using python 3.11.5 (>3.8)"

5a) The requirements.txt file cannot be open by "less" (reports it as a binary file), but "head" can see the contents. Please check the file.

5b) Format of requirements.txt is incorrect for pip. It looks more like a conda yml recipe, rather than a pip requirements.

5c) After I manually corrected the format of requirements.txt to "tool==version", the pip installation fails at backcall "ERROR: Could not find a version that satisfies the requirement backports==1.0 (from versions: none). ERROR: No matching distribution found for backports==1.0".

6) Banksy_py, GitHub section on "Quick installation via environment.yml file (via Anaconda)"

6a) The installation order seems incorrect. I think it should be git clone, move to directory, conda create, conda activate (right now it is git clone, conda activate, then conda create)

6b) I encountered dependency errors when trying to setup a conda environment based on the conda yml recipe in the GitHub. It was a "Solving environment: failed, ResolvePackageNotFound:" error, with a list of 115 dependencies that could not be found.

7) The revised work by the authors is substantial, but perhaps they authors could consider mentioning

the reasoning behind the split of the L2/3 region in the STARmap data (my question 4, reviewer fig 4). I am sure others will spot this, and perhaps the authors should provide an answer for this somewhere in the final version, at the authors discretion.

8) I think that the idea of using different starting seeds giving different results is not a surprise for most people. However, I was shocked at the wildly different performance of various tools based on their seed. Perhaps this warrants an additional sentence in the discussion as a reminder to the audience of stochasticity in the performance of tools. Again, at the authors discretion.

Minor

9) The authors should mention the coordinates for the displayed region in the MERSCOPE colorectal cancer patient 1 dataset.

Remarks with no consequence

10) In response to the authors answer to my previous comment 1 (last paragraph of the authors response). I was not eluding to that "more cluster are better", but rather in the special case of Moffit et al, they identified more clusters that were spatially restricted, without considering spatial information during the cell-typing process - the large number of spatially restricted neuronal populations is at least worth some consideration. At the very least, I would expect that a spatially aware cell-typing approach recapitulate those previous results (... which you show convincingly).

Reviewer #2 (Remarks to the Author):

The authors have addressed my comments. The integration of spatial data from multiple datasets by applying Harmony to the PC scores calculated from spatial neighbour-augmented embedding makes good sense. The harmony batch correction approach computes a correction factor for each dataset and when applied to spatial-adjusted PC scores, the factor would account for datasets with more or fewer cells with denser or more sparse cells. The improvement from the integration results looks convincing. New analyses, especially the application of new datatypes are interesting and add value. I believe the package will be applied broadly.

Reviewer #3 (Remarks to the Author):

The authors have satisfactorily addressed all major and minor comments I made (reviewer 3). I understand that some suggestions I made will take time to fully complete, such as getting BANSKY's R version submitted to Bioconductor. Though I see at <https://github.com/prabhakarlab/Banksy/commits/bioc> that the authors are actively working on this. I think this will greatly benefit BANSKY's usability as well as users of the software (such as myself!). Below I include a few notes I made along the way.

Notes from response to reviewer 2:

* "For benchmarking on DLPFC Visium data from Maynard et al. (2021), we ran BANKSY one sample

at a time as is the norm in the field (Hu et al. 2021, Zhao et al. 2021, Dong & Zhang, 2022, Long et al. 2023)." Multi-sample processing is challenging and yes, many people avoid it. Our group only uses tools that can process data across multiple samples, otherwise you end up with clusters across samples that are not related to each other and can be hard to match. So claiming that others typically avoid multi-sample processing as a justification seems flawed to me. I see that you address this again in the response to reviewer 3 (me) and do agree that joint clustering is important to address.

* At <https://prabhakarlab.github.io/Banksy/articles/dlpfc-analysis.html> it looks like BANKSY did not recover the blue manually-annotated layer (L4 as labeled in the UMAP plot). This document is also very sparse in text and explanations. This seems to be a limitation of using ARI, as the ARI scores for BANKSY are indeed on the higher end (most of them are above 0.6).

* Also at <https://prabhakarlab.github.io/Banksy/articles/dlpfc-analysis.html>, even with the extra information I have access to as a reviewer, I can't tell if this vignette is using z-transformations, BANKSY->Harmony, processing each sample independently, or something else. EDIT: I see that on the response to reviewer 3 you did clarify that this vignette is a simple concatenation of the samples. In that case, I would encourage you to actually show a vignette of the BANKSY->HARMONY option that you found performed the best as shown in Supplementary Figure 34.

Notes from response to reviewer 3:

* Thanks for clarifying my misunderstanding of Supplementary Figure 1 and adding Supplementary Figure 37.

* I wish you best of luck with maintaining both the R and Python versions of BANKSY. Your users will greatly benefit from you doing so.

* Related to 5.4, I look forward to learning from your code once you deposit it on Zenodo. Users can sometimes learn from it information that is not part of vignettes, the official documentation, methods sections on papers, etc.

* Since the R version is going to be your primary one, you could include on it a vignette for documenting the Python version. Then at https://github.com/prabhakarlab/Banksy_py/blob/main/README.md point users to this vignette. That way you can avoid having to keep READMEs in two different places updated for information that is common across both implementations of BANKSY. Of course, you might decide that it's easier for you to keep the documentation for R and Python separate, particularly if that's a requirement for your PyPi submission.

* I agree that BANKSY's ability to handle 500k-1M cells per MERSCOPE / Xenium / CosMX sample is going to be really important, and thus that scalability is so important. You might want to directly mention this fact on the paper, not only on your reviewers response.

* Thanks for adding <https://github.com/prabhakarlab/Banksy/blob/bioc/vignettes/multi-sample.Rmd>!

* I appreciate all the work behind the scenes to make BANKSY compatible with Bioconductor infrastructure packages and for planning to submit it to Bioconductor (I don't see it yet with <https://github.com/Bioconductor/Contributions/issues?q=banksy>). Thank you sincerely for doing all this work!

Author Rebuttal, second revision:

Reviewer #1 (Remarks to the Author):

I have reviewed the resubmitted work by Vipul Singhal, Nigel Chou, and colleagues. Their revised work addresses my major concerns well, and now I feel that the work more convincingly demonstrates the advantages of Banksy over other non-spatial and other spatial clustering approaches. However, there are still two easily addressable issues I would like the authors to consider. The first relates to the installation instructions of the tool (points 1-6), and the second relates to including additional text in the manuscript that corresponds to reviewer/editor only discussion (points 7,8).

Despite these issues, I am convinced that work presented is indeed worthy for consideration for a top tier journal. Well done!

We thank the reviewer for their kind words, for their careful checking of the installation process on an independent machine, and also for their recommendations on parts of the rebuttal that might be included in the manuscript.

Installation of Banksy R

- 1) The version of R is not defined. Perhaps the authors should encourage creating a fresh conda environment for Banksy, only defining the version of R, after which the remote installation can take place.

We thank the reviewer for this suggestion. It is indeed important to define the version of R, on which the installation of the remaining libraries can take place. This is specified in the description file of the R package (<https://github.com/prabhakarlab/Banksy/blob/1a15f0c1a4f498d04764eb3a319d6507caadd64e/DESCRIPTION#L22>). We now also indicate the R version in the Code Availability section.

The installation has now been tested in a virtual machine, and should proceed smoothly.

- 2) I encountered linux dependencies for installation via R. One is listed on the Banksy GitHub, but two were not. Perhaps the authors could add these dependencies to a potential conda setup for install a base version of R, and these additional dependencies. I found version of all three packages below on conda-forge.

2a) I encountered the leidenag issue in that it required, requires igraph. These issues were resolved via instructions on the leidenag GitHub, but authors could consider mentioning how to remedy the issue on the Banksy GitHub. In my case it was solved by “sudo apt-get update” and “sudo apt-get install libxml2-dev libgmp-dev libglpk-dev” on my personal machine. I do not know how easily this can be solved when users do not have root/sudo access to a machine, as is common with university HPC systems.

2b) I also had to install cairo (“sudo apt-get install libcairo2-dev”). This issue is not stated on the Banksy GitHub.

2c) I also had to install curl (“libcurl4-openssl-dev”). This issue is not stated on the Banksy GitHub.

We apologize for letting dependencies 2 and 3 escape our notice, and thank you for your careful checks and suggestions. These two dependencies are only present for the older version of the code, which used the custom BanksyObject framework (v0.1.5), and are not present in the new BioConductor release. We have documented the need for these packages in the readme file of the old release.

As mentioned above, we have now tested the installation of the new version on a VM, and documented the full process for a clean installation

(<https://github.com/prabhakarlab/Banksy/wiki/Installation-on-Linux-from-scratch>).

There should not be any more issues.

3) Remark: apart from the above issues, I could complete the basic tutorial. Installation of Banksy_py

We thank the reviewer for their careful checking of the Banksy_py installation process, and are glad to hear that the reviewer was able to install and run the slideseq v1 tutorial. We have now resolved the installation issues the reviewer pointed out (detailed below) and have updated and improved the installation instructions.

4) Banksy_py, GitHub section on “Installation in new Anaconda environment (recommended)”

4a) After cloning the repo, the “cd Banksy_py” step is missed

We thank the reviewer for pointing this out. We have now edited our README.md to include clearer instructions, including the “cd Banksy_py” step, as an example code section:

Installation via Anaconda (recommended)

To use `Banksy_py`, we recommend setting up a `conda` environment and installing the prerequisite packages, then cloning this repository.

```
(base) $ conda create --name banksy
(base) $ conda activate banksy
(banksy) $ conda install -c conda-forge scanpy python-igraph leidenalg
(banksy) $ git clone https://github.com/prabhakarlab/Banksy_py.git
(banksy) $ cd Banksy_py
```

To run the examples presented in `jupyter` notebooks, install the extensions for `jupyter`.

```
(banksy) $ conda install -c conda-forge jupyter
```

4b) If the “python utils” library is already installed via pip, the “banksy Utils” is not accessible. I could get around this by uninstalling python utils via pip, but this is not an ideal solution.

We thank the reviewer for pointing out this namespace conflict. We have now refactored the `Banksy_py` package by renaming the “utils” folder to “banksy_utils” to avoid conflict with the python utils library. We tested the code after refactoring, and the package is working as intended.

4c) Remark: apart from the above issues, I could run tutorial for slideseq v1 to end on a 8Gb laptop in very little time

We are glad to hear that the tutorial was quick to implement.

5) `Banksy_py`, GitHub section on “Quick installation (via pip) Using python 3.11.5 (>3.8)”

5a) The requirements.txt file cannot be open by “less” (reports it as a binary file), but “head”

can see the contents. Please check the file.

We have now replaced the requirements.txt file with the correct format (see response to 5b,c below) and ensured that it uses the “UTF-8” character encoding that can be opened on both Windows and Unix-based systems. We checked that it can be opened with “less”, “more” and “head” on a linux system, as well as with “Get-Content”, “type” and “more” (Windows Powershell) on a Windows machine. Hence most users should not encounter such an issue now.

Regarding the error reading the original requirements.txt file, we have traced the problem to its root cause, and explain the troubleshooting process in detail below.

First, we realized that the file could be read by most tools in Windows (Notepad, Notepad++, “Get-Content”, “type” and “more” commands on Windows Powershell), which narrowed down the issue to Unix-based systems. After checking the character encoding of the file using Notepad++, we realized that the file was written with the “UTF-16 LE” encoding, commonly used on Windows systems. Conversely, Linux uses the “UTF-8” encoding. When we converted the “UTF-16 LE” encoding to “UTF-8” using Notepad++ (Encoding -> Convert to UTF-8), the file could be read by “less” with no errors. Alternatively, using the “iconv” command (Linux command line) to convert from “UTF-16 LE” to “UTF-8”:

```
iconv -f UTF-16LE -t UTF-8 requirements.txt -o requirements_utf8.txt
```

also allowed the file to be read. Hence, it was this encoding issue that caused the errors encountered with the Linux commands “less” and “more”. We now ensure that all text files (e.g. the new requirements.txt) use “UTF-8” encoding which can be read by both Windows and Unix-based platforms.

5b) Format of requirements.txt is incorrect for pip. It looks more like a conda yml recipe, rather than a pip requirements.

We have checked the previous requirements.txt file and believe that the issue is caused by the file being created using conda with the command:

```
conda list --export > requirements.txt
```

This is why the file is not in the format that pip recognizes.

We have now generated a new requirements.txt file with the correct format (“tool==version”) using

```
pip freeze > requirements.txt
```

and have tested that we can successfully perform the installation via pip on a Windows laptop and a Linux environment. Note that for this installation method, the Python version has to be <3.12, which we now indicate in the readme. We have updated the requirements.txt file in our github page (https://github.com/prabhakarlab/Banksy_py) and added the instructions for installation via pip in README.md:

Installation using pip from requirements.txt file

Users who have python = 3.8-3.11 and pip can also install our environment from requirements.txt here after cloning in this repository:

```
$ git clone https://github.com/prabhakarlab/Banksy_py.git
$ cd Banksy_py
$ pip install -r requirements.txt
```

5c) After I manually corrected the format of requirements.txt to “tool==version”, the pip installation fails at backcall “ERROR: Could not find a version that satisfies the requirement backports==1.0 (from versions: none). ERROR: No matching distribution found for backports==1.0”.

We apologize for the errors caused by the previously misformatted requirement.txt. As described in response to comment 5b, we have now updated the requirements.txt file to the correct format, and have tested that we can recreate the environment using this file.

6) Banksy_py, GitHub section on “Quick installation via environment.yml file (via Anaconda)”
6a) The installation order seems incorrect. I think it should be git clone, move to directory, conda create, conda activate (right now it is git clone, conda activate, then conda create)

We thank the reviewer for pointing out this mistake. That is indeed the correct order, and we have corrected the installation order in the readme:

Installation from `environment.yml` file

Users can directly install the prerequisite packages (which replicates our Anaconda environment) from `environment.yml` here after cloning in this repository:

```
(base) $ git clone https://github.com/prabhakarlab/Banksy_py.git
(base) $ cd Banksy_py
(base) $ conda env create --name banksy --file=environment.yml
(base) $ conda activate banksy
```

6b) I encountered dependency errors when trying to setup a conda environment based on the conda yml recipe in the GitHub. It was a “Solving environment: failed, ResolvePackageNotFound:” error, with a list of 115 dependencies that could not be found.

We apologize for this issue, and have now regenerated the conda yml file using:

```
conda env export --no-builds | grep -v "prefix" > environment.yml
```

We tested that we were able to set up this environment using the new `environment.yml` file on a Windows laptop and Linux environment.

7) The revised work by the authors is substantial, but perhaps they authors could consider mentioning the reasoning behind the split of the L2/3 region in the STARmap data (my question 4, reviewer fig 4). I am sure others will spot this, and perhaps the authors should provide an answer for this somewhere in the final version, at the authors discretion.

We agree that the reasoning behind the L2/3 split deserves mention in the manuscript. We have now added a discussion in Section 8.11 (STARmap Data):

We note that both BANKSY and GraphST split the L2/3 region vertically. We performed additional analyses to investigate the possibility that this split was because of differential data quality between tiles in the upper and lower halves of the imaged region. First, we examined the spatial distribution of the number of detected genes (NODG) in each cell, a common QC metric for data quality, but did not observe a tiling pattern in NODG that could explain the L2/3 split. We also did not find any statistically significant differences in NODG between upper and lower clusters identified by BANKSY for any of the cell types (FDR q-value > 0.2; Benjamini-Hochberg adjustment for multiple testing; NODG for each cell type with at least 10 cells represented was compared using the two-tailed Wilcoxon rank-sum test). Thus, we did not find evidence that data quality differences drove the split in L2/3. Next, we asked if the split in L2/3 could be attributed to specific differentially expressed genes. We identified 4 genes with significant upregulation in the lower subcluster (Trim32, Nr4a1, Nrgn and 2900055J20Rik; FDR q-value (BH) < 0.02) and 2 genes with upregulation in the upper subcluster (Hlf, Bcl6; FDR q-value (BH) < 0.04), which may explain the splitting of this tissue domain into two spatially distinct clusters by BANKSY and GraphST. Nevertheless, for quantitative benchmarking purposes, we treated the split in L2/3 as an erroneous prediction by BANKSY and GraphST, and it was therefore reflected in the ARI metric.

8) I think that the idea of using different starting seeds giving different results is not a surprise for most people. However, I was shocked at the wildly different performance of various tools based on their seed. Perhaps this warrants an additional sentence in the discussion as a reminder to the audience of stochasticity in the performance of tools. Again, at the authors discretion.

We found the extent of the variation surprising as well. We have now added the following text in the Results section (page 13):

The performance of deep-learning methods can be sensitive to the choice of initial random seed (Picard, D., 2021), and we found that this was indeed the case for the published deep-learning methods we tested (SpaGCN, STAGATE, and GraphST, Extended Data Fig. 7b). To robustly estimate their performance, we therefore calculated the median ARI across 11 commonly used seeds in both the DLPFC and STARmap analyses above (Methods Sections 4.10 and 4.11). For consistency, we also used these 11 seeds in BANKSY's Leiden clustering step.

The figure cited above (*Extended Data Fig. 7b*) was previously Reviewer Fig. 6. We have added this extended data figure to address this comment.

Extended Data Fig. 7b: Domain segmentation comparison (boxplot) on the STARmap 1,020-gene mouse visual cortex dataset, showing performance variability across random seeds. GraphST, STAGATE, SpaGCN and BANKSY were run with 11 different random seeds and ARI relative to the reference domain annotation was calculated in each case.

Finally, we have added the following line mentioning the stochastic nature of the algorithms to the Methods Section 8.10 (highlighted in brown, lines 1482-1483):

“Due to the stochasticity inherent in deep-learning methods, their performance can be sensitive to the choice of random seeds (Picard, D., 2021).”

Minor

- 9) The authors should mention the coordinates for the displayed region in the

MERSCOPE colorectal cancer patient 1 dataset.

We have now included the coordinates for the displayed region in the figure.

Remarks with no consequence

10) In response to the authors answer to my previous comment 1 (last paragraph of the authors response). I was not eluding to that "more cluster are better", but rather in the special case of Moffit et al, they identified more clusters that were spatially restricted,

without considering spatial information during the cell-typing process - the large number of spatially restricted neuronal populations is at least worth some consideration. At the very least, I would expect that a spatially aware cell-typing approach recapitulate those previous results (... which you show convincingly).

We appreciate the reviewer's comment, and completely agree about the importance of identifying these spatially restricted neuronal subpopulations.

Reviewer #2 (Remarks to the Author):

The authors have addressed my comments. The integration of spatial data from multiple datasets by applying Harmony to the PC scores calculated from spatial neighbour-augmented embedding makes good sense. The harmony batch correction approach computes a correction factor for each dataset and when applied to spatial-adjusted PC scores, the factor would account for datasets with more or fewer cells with denser or more sparse cells. The improvement from the integration results looks convincing. New analyses, especially the application of new datatypes are interesting and add value. I believe the package will be applied broadly.

We appreciate the positive feedback on our integration approach, and are glad that our new analyses are seen as valuable. We indeed share the sentiment that the BANKSY package will be adopted and applied broadly.

Reviewer #3 (Remarks to the Author):

The authors have satisfactorily addressed all major and minor comments I made (reviewer 3). I understand that some suggestions I made will take time to fully complete, such as getting BANSKY's R version submitted to Bioconductor. Though I see at <https://github.com/prabhakarlab/Banksy/commits/bioc> that the authors are actively working on this. I think this will greatly benefit BANSKY's usability as well as users of the software (such as myself!). Below I include a few notes I made along the way.

Notes from response to reviewer 2:

* "For benchmarking on DLPFC Visium data from Maynard et al. (2021), we ran BANSKY one sample at a time as is the norm in the field (Hu et al. 2021, Zhao et al. 2021, Dong & Zhang, 2022, Long et al. 2023)." Multi-sample processing is challenging and yes, many people avoid it. Our group only uses tools that can process data across multiple samples, otherwise you end up with clusters across samples that are not related to each other and can be hard to match. So claiming that others typically avoid multi-sample processing as a justification seems flawed to me. I see that you address this again in the response to reviewer 3 (me) and do agree that joint clustering is important to address.

We appreciate the reviewer's comment and apologize for the lack of clarity in our previous reply. We ran DLPFC samples one at a time to provide a quantitative benchmarking result (i.e. the median ARI value) that is directly comparable against numerous previous publications that also used this benchmarking procedure. If we had reported the results using BANSKY->Harmony integration (which yields an even higher ARI) in Fig. 6, this would not be an apples-to-apples comparison and it would be unfair to compare this ARI value to existing literature benchmark ARIs.

However, if the question at hand is not benchmarking but best practices for analyzing multiple related samples, we fully agree with the reviewer that data integration (e.g. BANSKY->Harmony) would be ideal as this approach yielded superior results compared to running samples one at a time on this dataset.

We have now clarified in the results section that analysis of one sample at a time was to match the procedure used in previous benchmarking analyses.

* At <https://prabhakarlab.github.io/Banksy/articles/dlpfc-analysis.html> it looks like BANSKY did not recover the blue manually-annotated layer (L4 as labeled in the UMAP plot). This document is also very sparse in text and explanations. This seems to be a limitation of using ARI, as the ARI scores for BANSKY are indeed on the higher end (most of them are above 0.6).

We have substantially revamped the vignette (<https://prabhakarlab.github.io/Banksy/articles/multi-sample.html>) in preparation for Bioconductor submission, and have added more text to describe the code. We agree with the reviewer that the ARI has its limitations. Additional metrics such as the NMI and MCC should also be used in conjunction with the ARI to judge the quality of the clustering algorithm (Extended Data Figs. 6, 7, 9 and Supp. Fig. 22). As seen in these supplementary figures and in the vignette, BANSKY's performs just as well by the NMI and MCC metrics.

* Also at <https://prabhakarlab.github.io/Banksy/articles/dlpfc-analysis.html>, even with the extra information I have access to as a reviewer, I can't tell if this vignette is using z-transformations, BANSKY->Harmony, processing each sample independently, or something else. EDIT: I see that on the response to reviewer 3 you did clarify that this vignette is a simple concatenation of the samples. In that case, I would encourage you to actually show a vignette of the BANSKY->HARMONY option that you found performed the best as shown in Supplementary Figure 34.

We have now included the best performing BANSKY->HARMONY option (Extended Data Fig 9) as a vignette (<https://prabhakarlab.github.io/Banksy/articles/batch-correction.html>). As requested, we have also added text to the original vignette (<https://prabhakarlab.github.io/Banksy/articles/multi-sample.html>) explaining that samples were z-transformed separately and simply concatenated.

Notes from response to reviewer 3:

* Thanks for clarifying my misunderstanding of Supplementary Figure 1 and adding Supplementary Figure 37.

* I wish you best of luck with maintaining both the R and Python versions of BANSKY.

Your users will greatly benefit from you doing so.

We thank the reviewer for the encouraging words.

* Related to 5.4, I look forward to learning from your code once you deposit it on Zenodo. Users can sometimes learn from it information that is not part of vignettes, the official documentation, methods sections on papers, etc.

* Since the R version is going to be your primary one, you could include on it a vignette for documenting the Python version. Then at https://github.com/prabhakarlab/Banksy_py/blob/main/README.md point users to this vignette. That way you can avoid having to keep READMEs in two different places updated for information that is common across both implementations of BANKSY. Of course, you might decide that it's easier for you to keep the documentation for R and Python separate, particularly if that's a requirement for your PyPi submission.

We thank the reviewer for this comment. We have provided hyperlinks pointing users to the other implementation within each readme (R -> Python and Python -> R).

* I agree that BANKSY's ability to handle 500k-1M cells per MERSCOPE / Xenium / CosMX sample is going to be really important, and thus that scalability is so important. You might want to directly mention this fact on the paper, not only on your reviewers response.

We agree that scalability is so important that it should be included in the manuscript. We discuss this in the manuscript on lines 472-474 of the discussion section:

"...speed and scalability are essential features of any modern spatial clustering tool, now that technologies are available to profile over a million cells [53] in a single dataset."

* Thanks for adding <https://github.com/prabhakarlab/Banksy/blob/bioc/vignettes/multi-sample.Rmd>!

* I appreciate all the work behind the scenes to make BANSKY compatible with Bioconductor infrastructure packages and for planning to submit it to Bioconductor (I don't see it yet with <https://github.com/Bioconductor/Contributions/issues?q=banksy>).

Thank you sincerely for doing all this work!

We thank the reviewer for this comment, and note that we have now initiated the Bioconductor submission process:

<https://github.com/Bioconductor/Contributions/issues/3201>

Final Decision Letter:

16th Jan 2024

Dear Shyam,

I am delighted to say that your manuscript "BANKSY unifies cell typing and tissue domain segmentation for scalable spatial omics data analysis" has been accepted for publication in an upcoming issue of Nature Genetics.

Your paper will be published online after we receive your corrections and will appear in print in the next available issue. You can find out your date of online publication by contacting the Nature Press Office (press@nature.com) after sending your e-proof corrections.

Before your paper is published online, we shall be distributing a press release to news organizations

worldwide, which may very well include details of your work. We are happy for your institution or funding agency to prepare its own press release, but it must mention the embargo date and Nature Genetics. Our Press Office may contact you closer to the time of publication, but if you or your Press Office have any enquiries in the meantime, please contact press@nature.com.

Please note that *Nature Genetics* is a Transformative Journal (TJ). Authors may publish their research with us through the traditional subscription access route or make their paper immediately open access through payment of an article-processing charge (APC). Authors will not be required to make a final decision about access to their article until it has been accepted. [Find out more about Transformative Journals](https://www.springernature.com/gp/open-research/transformative-journals)

Authors may need to take specific actions to achieve [compliance](https://www.springernature.com/gp/open-research/funding/policy-compliance-faqs) with funder and institutional open access mandates. If your research is supported by a funder that requires immediate open access (e.g. according to [Plan S principles](https://www.springernature.com/gp/open-research/plan-s-compliance)) then you should select the gold OA route, and we will direct you to the compliant route where possible. For authors selecting the subscription publication route, the journal's standard licensing terms will need to be accepted, including [self-archiving-and-license-to-publish](https://www.nature.com/nature-portfolio/editorial-policies/self-archiving-and-license-to-publish). Those licensing terms will supersede any other terms that the author or any third party may assert apply to any version of the manuscript.

An online order form for reprints of your paper is available at <https://www.nature.com/reprints/author->

reprints.html"><https://www.nature.com/reprints/author-reprints.html>. Please let your coauthors and your institutions' public affairs office know that they are also welcome to order reprints by this method.

If you have not already done so, we invite you to upload the step-by-step protocols used in this manuscript to the Protocols Exchange, part of our on-line web resource, natureprotocols.com. If you complete the upload by the time you receive your manuscript proofs, we can insert links in your article that lead directly to the protocol details. Your protocol will be made freely available upon publication of your paper. By participating in natureprotocols.com, you are enabling researchers to more readily reproduce or adapt the methodology you use. [Natureprotocols.com](http://natureprotocols.com) is fully searchable, providing your protocols and paper with increased utility and visibility. Please submit your protocol to <https://protocolexchange.researchsquare.com/>. After entering your nature.com username and password you will need to enter your manuscript number (NG-A62310R3). Further information can be found at <https://www.nature.com/nature-portfolio/editorial-policies/reporting-standards#protocols>

Sincerely,

Michael Fletcher, PhD
Senior Editor, Nature Genetics

ORCID: 0000-0003-1589-7087